

# Boreal forest fire CO and CH$_4$ emission factors derived from tower observations in Alaska during the extreme fire season of 2015

Elizabeth B. Wiggins[1*], Arlyn Andrews[2], Colm Sweeney[2], John B. Miller[2], Charles E. Miller[3], Sander Veraverbeke[4], Roisin Commane[5], Steven Wofsy[6], John M. Henderson[7], and James T. Randerson[1]

[1]Department of Earth System Science, University of California, Irvine, California, USA, [2]National Oceanic and Atmospheric Administration, Boulder, Colorado, USA, [3]Jet Propulsion Laboratory, California Institute of Technology, Pasadena, California, USA, [4]Vrije University Amsterdam, Netherlands, [5]Department of Earth and Environmental Sciences, Columbia University, Palisades, New York, USA, [6]School of Engineering and Applied Sciences, Harvard, Cambridge, Massachusetts, USA, [7]Atmospheric and Environmental Research, Inc., Lexington, Massachusetts, USA,*Now at NASA Langley Research Center, Hampton, VA, USA

*Correspondence to:* Elizabeth B. Wiggins (Elizabeth.b.wiggins@nasa.gov)

**Abstract.** Recent increases in boreal forest burned area, which have been linked with climate warming, highlight the need to better understand the composition of wildfire emissions and their atmospheric impacts. Here we quantified emission factors for CO and CH$_4$ from a massive regional fire complex in interior Alaska during the summer of 2015 using continuous high-resolution trace gas observations from the Carbon in Arctic Reservoirs Vulnerability Experiment (CRV) tower in Fox, Alaska. Averaged over the 2015 fire season, the mean CO/CO$_2$ emission ratio was $0.142 \pm 0.051$ and the mean CO emission factor was $127 \pm 40$ g kg$^{-1}$ dry biomass burned. The CO/CO$_2$ emission ratio was about 39% higher than the mean of previous estimates derived from aircraft sampling of wildfires from boreal North America. The mean CH$_4$/CO$_2$ emission ratio was $0.010 \pm 0.004$ and the CH$_4$ emission factor was $5.3 \pm 1.8$ g kg$^{-1}$ dry biomass burned, which are consistent with the mean of previous reports. CO and CH$_4$ emission ratios varied in synchrony, with higher CH$_4$ emission factors observed during periods with lower modified combustion efficiency (MCE). By coupling a fire emissions inventory with an atmospheric model, we identified at least 34 individual fires that contributed to trace gas variations measured at the CRV tower, representing a sample size that is nearly the same as the total number of boreal fires measured in all previous field campaigns. The model also indicated that typical mean transit times between trace gas emission within a fire perimeter and tower measurement were 1-3 days, indicating that the time series sampled combustion across day and night burning phases. The high CO emission ratio estimates reported here provide evidence for a prominent role of smoldering combustion, and illustrate the importance of continuously sampling fires across time-varying environmental conditions that are representative of a fire season.

## 1 Introduction

Boreal forest fires influence the global carbon cycle and climate system through a variety of pathways. These fires initiate succession, influence landscape patterns of carbon accumulation, and directly release carbon dioxide and other trace gases and aerosols into the atmosphere [*Johnson*, 1996]. One of the largest reservoirs of global terrestrial carbon resides in organic soils underlying boreal forests [*Apps et al.*, 1993; *Rapalee et al.,* 1998; *Tarnocai et al.,* 2009], and fires in the boreal forest can consume significant amounts of aboveground and belowground biomass [*Harden et al.*, 2000; *French et al.*, 2004; *Boby et al.*, 2010; *Walker et al.*, 2018]. Many boreal forest fires are stand replacing and high energy [*Johnstone et al.*, 2011; *Rogers et al.*, 2015], with enough

convective power to inject aerosols into the upper troposphere and lower stratosphere where they can be widely dispersed across the Northern Hemisphere [*Fromm et al.*, 2000; *Forster et al.*, 2001; *Turquety et al.*, 2007; *Peterson et al.*, 2018].

Emissions from boreal fires are known to considerably influence atmospheric composition in downwind areas. Fire plumes from regional fire complexes in Alaska and western Canada, for example, have been shown to influence air quality over Nova Scotia [*Duck et al.*, 2007], the south-central United States [*Wotawa et al.*, 2001; *Kasischke et al.*, 2005] and Europe [*Forster et al.*, 2001]. Similarly, emissions from boreal forest fires in Russia have caused unhealthy air quality in Moscow [*Konovalov et al.*, 2011] and have affected ozone and other trace gases concentrations across the western United States [*Jaffe et al.*, 2004]. Over the past few decades, annual burned area in several regions in boreal North America has increased [*Gillett et al.*, 2004; *Kasischke and Turetsky*, 2006; *Veraverbeke et al.*, 2017], and projections suggest further increases may occur in response to changes in fire weather and a lengthening of the fire season [*Flannigan et al.*, 2001; *de Groot et al.*, 2013; *Young et al.*, 2017]. As a consequence, fires are likely to play an increasingly important role in regulating air quality and climate during the remainder of the 21$^{st}$ century.

Emission factors provide a straightforward way to convert fire consumption of dry biomass into emissions of specific trace gas species, such as CO, $CH_4$, and $CO_2$. This technique is commonly used to model emissions of select species in fire inventories, allowing for comparison of atmospheric model simulations with in-situ or remotely sensed mole fraction or concentration observations. The most frequently used boreal forest fire emission factors are derived from meta-analyses that average together information from individual field campaigns [*Andreae and Merlet*, 2001; *Akagi et al.*, 2011; *Andreae*, 2019]. These syntheses often include in situ airborne and ground based measurements along with laboratory measurements of combusted fuels. There is no consensus on how to combine information from different studies, and in past work individual studies have sometimes been given equal weight when estimating biome-level means, even when the number of fires and duration of sampling has varied considerably from one field campaign to another.

In past work, the most common approach for measuring emission factors from boreal fires is to fly aircraft through smoke plumes, measuring trace gases using gas analyzers mounted in the aircraft or by collecting flasks of air that are measured later in the laboratory. Over a period of more than 25 years, a total of at least 42 boreal fires have been sampled by aircraft, including 19 wildfires and 14 prescribed land management fires from boreal North America and 9 prescribed fires in Siberia (Table 1). Aircraft sampling is a highly effective approach for sampling large and remote wildfires, especially for characterizing non-conserved trace gas and particulate emissions that have lifetimes of hours to days. It also important to recognize potential limits associated with sampling fires in this way. Aircraft observations are mostly confined to periods with good visibility, often sampling well-developed fire plumes during mid-day and during periods with relatively low cloud cover. These conditions represent a subset of the environmental variability that a large wildland fire may experience in boreal forest ecosystems as it burns over a period of weeks to months. An alternative approach for measuring in situ emission factors involves using a fixed surface site that continuously samples trace gas concentrations in an area downwind of a fire. This approach has been used to estimate CO emission ratios during a moderate fire season in Alaska [*Wiggins et al.*, 2016] and to estimate emission factors in other biomes [*Collier et al.*, 2016; *Benedict et al.*, 2017; *Selimovic et al.*, 2019; *Selimovic et al.*, 2020]. Surface sampling near or within fire perimeters may have an advantage with respect to providing measurements during intervals when aircraft are unable to fly, but are also more likely to under sample emissions injected above the boundary layer by fire plumes and pyro-cumulus clouds [*Selimovic et al.*, 2019].

Environmental conditions, including weather, vegetation, and edaphic conditions are known to influence the composition of emissions, in part by regulating the prevalence of flaming and smoldering combustion processes [*Ward and Radke*, 1993; *Yokelson et al.*, 1997; *Akagi et al.*, 2011; *Urbanski*, 2014]. The relative amounts of smoldering and flaming combustion are difficult to measure, but can be estimated using the modified combustion efficiency (MCE), defined as $\Delta CO_2/(\Delta CO_2 + \Delta CO)$, where the $\Delta$ notation denotes the fire-associated dry air mole fraction of a sample gas after background levels have been removed. Fire

emissions dominated by flaming combustion have an MCE from 0.92 – 1.0, while emissions dominated by smoldering combustion have an MCE that often ranges between 0.65 and 0.85 [*Akagi et al.*, 2011; *Urbanski*, 2014]. MCE can be used to understand the relative contribution of flaming and smoldering combustion processes to the composition of trace gases and aerosols in air measured downwind of a fire. Smoldering combustion converts solid biomass to gases and aerosols, while flaming oxidizes some emissions [*Yokelson et al.*, 1996, 1997]. As a consequence, smoldering combustion produces more CO, $CH_4$, and organic carbon aerosol relative to $CO_2$ [*Ward and Radke*, 1993; *Urbanski et al.*, 2008]. Flaming combustion requires the presence of organic material that burns efficiently [*Ryan et al.,* 2002], and often occurs in boreal forests when fires consume dry aboveground fuels, including vegetation components with low moisture content, litter, and fine woody debris [*French et al.*, 2002]. Smoldering, in contrast, is a dominant combustion process for burning of belowground biomass and larger coarse woody debris. Residual smoldering combustion in boreal forests can continue to occur for weeks after a flaming fire front has passed through, especially in peatland areas with carbon-rich organic soils [*Harden et al.*, 2000; *Bertschi et al.*, 2003]. Over the lifetime of a large fire, smoldering combustion is more likely to occur during periods with lower temperatures and higher atmospheric humidity that increase the moisture content of fine fuels [*Stocks et al.*, 2001; *Ryan*, 2002].

Here we used trace gas observations of CO, $CH_4$, and $CO_2$ from the CRV tower to estimate emission factors from boreal forest fires that burned during the near-record high Alaska fire season of 2015. The summer of 2015 was the second largest fire season in terms of burned area since records began in 1940 with about 2.1 million hectares burned [*Hayasaka et al.*, 2016; *Partain et al.*, 2016]. An unseasonably warm spring and early snowmelt allowed fuels to dry early in the season [*Partain et al.*, 2016]. In mid-June, thunderstorms caused an unprecedented number of lightning strikes (over 65,000) that ignited over 270 individual fires on anomalously dry fuel beds over the course of a week [*Hayasaka et al.*, 2016; *Veraverbeke et al.*, 2017]. Fires expanded rapidly during several hot and dry periods through mid-July, and then slowed down as multiple precipitation events and cool, damp weather minimized fire growth for the rest of the summer fire season.

The CRV tower captured an integrated signal of trace gas emissions from multiple fires across interior Alaska during the 2015 fire season [*Karion et al.*, 2016]. The data stream was comprised of continuous sampling for about 47 minutes out of every hour from June 9 – August 13, yielding more than 59,800 individual measurements, each with a 30 s duration. We identified intervals when fire emissions had a dominant influence on trace gas variability at CRV tower, and used these intervals to derive emission ratios. Analysis of these data indicate that smoldering processes may have a higher contribution to total wildfire emissions from North American boreal forests than previous estimates derived from aircraft sampling. To quantify the spatial and temporal variability of individual fires and their influence on CO, $CH_4$, and $CO_2$ at the CRV tower, we coupled a fire emissions inventory, the Alaska Fire Emissions Database (AKFED) [*Veraverbeke et al.*, 2015] with an atmospheric transport model, the Polar Weather Research and Forecasting Stochastic Time Integrated Lagrangian Transport (PWRF-STILT) model [*Henderson et al.*, 2015]. This modeling analysis indicated that the number of 2015 wildfires sampled in our study is comparable to the total number of North American boreal forest fires sampled in past work.

## 2 Methods

### 2.1 CARVE (CRV) Tower Observations

Atmospheric CO, $CH_4$, and $CO_2$ mole fractions were measured using a cavity ring-down spectrometer (CRDS, Picarro models 2401 and 2401m) [*Karion et al.*, 2016] at the CRV tower in Fox, Alaska (64.986°N, 147.598°W, ground elevation 611m above sea level). The tower is located about 20 km northeast of Fairbanks Alaska on top of a hill in hilly terrain (Figure 1), and within the interior lowland and upland forested ecoregion in interior Alaska [*Cooper et al.*, 2006]. There are three separate inlets

on the CRV tower at different heights above ground level from which the spectrometer draws air for sampling. The spectrometer samples air from the highest level for about 50 minutes out of every hour, and then draws air from the other two levels for 5 minutes at each level [*Karion et al.,* 2016]. Standard reference gases are sampled every 8 hours for 5 minutes, and measurements are removed for a time equivalent to three flushing volumes of the line, approximately 3 minutes, after a level change or switch to or from a calibration tank. All raw 30 s average measurements were calibrated according to *Karion et al.* [2016].

We used observations from air drawn from the top intake height at a height of 32 m above ground level in our analysis because this level had the highest measurement density and the smallest sensitivity to local ecosystem $CO_2$ fluxes near the tower [*Karion et al.*, 2016]. We used gaps in this time series, created when the spectrometer cycled to the lower inlets and following calibration, to separate the time series into discrete time intervals for the calculation of emission ratios. Each 30 s average measurement within a 47-minute sampling interval served as an individual point in our calculation of an emission ratio described below (Table 2).

**2.2 Emission Ratios, Emission Factors, and Modified Combustion Efficiency**

We isolated intervals when fire had a dominant influence on trace gas variability observed at CRV to calculate emission ratios. An interval with dominant fire influence was defined as a continuous 47-minute measurement period that had: 1) a minimum of at least 30 trace gas measurements (with each measurement representing a mean over 30 seconds), 2) a mean CO over the entire interval exceeding 0.5 ppm, and 3) significant correlations between CO and $CO_2$, and between $CH_4$ and $CO_2$, with $r^2$ values for both relationships exceeding 0.80.

For each interval, we required a sample size of at least 30 individual 30 s measurements. For each interval meeting this criterion, we calculated the mean CO mole fraction and discarded intervals that had a mean CO less than 0.5 ppm. For each of the intervals with mean CO that exceeded the 0.5 ppm threshold, we then extracted the 30 s measurement time series of CO, $CH_4$, and $CO_2$ mole fractions and calculated correlation coefficients between the trace gas time series. Only intervals with high and significant correlations between CO and $CO_2$ and between $CH_4$ and $CO_2$ ($r^2 > 0.80$; $p < 0.01$, $n > 30$) were retained, because covariance among these co-emitted species is a typical signature of combustion [*Urbanski*, 2014]. Data from each of the intervals that met the three criteria described above were used to compute emission ratios, emission factors, and MCE. These intervals are reported in chronological order in Table 2.

We calculated background mole fractions of CO and $CH_4$ by taking an average of observations prior to any major fire activity in interior Alaska during day of year (DOY) 160 – 162.5. This yielded a CO background of 0.110 ppm and a $CH_4$ background of 1.900 ppm. We modeled hourly $CO_2$ background mole fractions to account for the influence of net ecosystem exchange (NEE) using a multi-variable linear regression model trained on CRV tower observations during 2012, a year with little to no fire influence on trace gas variability. The variables used in the $CO_2$ model include DOY and hourly observations of temperature, vapor pressure deficit, precipitation, latent heat flux, and hourly $CO_2$ observations from Barrow, AK (Figure 2). Meteorological variables were acquired from the National Climatic Data Center Automated Weather Observing System for Fairbanks International Airport (http://www7.ncdc.noaa.gov/CDO/cdopoemain.cmd). This location was chosen due to its proximity to the CRV tower. We obtained 3-hourly latent heat flux estimates from the NOAH2.7.1 GLDAS/NOAH experiment 001 for version 2 of the Global Land Data Assimilation System (GLDAS-2) [*Rodell et al.*, 2015]. Hourly in situ $CO_2$ observations from a clean air site at Barrow, AK were obtained from the Earth System Research Laboratory Global Monitoring Division [*Thoning et al.*, 2007]. Our model assumed negligible influence from fossil fuel combustion on background mole fraction variability. After training on data from the summer of 2012, the model was then run using 2015 input variables to calculate time

evolving $CO_2$ background mole fractions during our analysis period. In a final step, the hourly $CO_2$ model was linearly interpolated to have the same temporal resolution as the 30 s individual trace gas measurements.

We estimated an emission ratio ($ER_X$; equation 1) by calculating the slope from a type II linear regression of CO or $CH_4$ excess mole fractions ($\Delta X$) relative to the $CO_2$ excess mole fraction ($\Delta CO_2$) using all of the 30 s observations available within a single 47-minute sampling interval when fire had a dominant influence on tower trace gas variability (up to 95 pairs of measurements). Uncertainty estimates for each interval were estimated as the standard deviation of the slope of the regression. To estimate excess mole fractions (denoted with a $\Delta$), we first removed background mole fractions (described above) before performing the regression analysis and obtaining the slope. The assumed background levels for CO and $CH_4$ did not influence this emission ratio estimate, because they were assumed to remain constant throughout the duration of each 47-minute interval (i.e., they influenced the intercept but not the slope of the regression line). In a sensitivity analysis we found that the removal of the $CO_2$ background, which did evolve within each 47-minute interval, had only a negligible effect, because the $CO_2$ background did not change rapidly over time. Since multiple fires were often burning simultaneously during the 2015 fire season, the emission ratios we report in Table 2 for each interval likely represent a composite of emissions from several fires.

$$ER_X = \frac{\Delta X}{\Delta CO_2} = \frac{X_{Fire} - X_{Background}}{CO_{2\,Fire} - CO_{2\,Background}} \tag{1}$$

Emission factors ($EF_X$) were calculated using equation 2, where $MM_x$ is the molar mass of CO or $CH_4$, $MM_C$ is the molar mass of carbon, $F_C$ is the mass fraction of carbon in dry biomass, 1000 is a factor to convert kg to g, $ER_X$ is the emission ratio, and $C_T$ is given by equation 3. The units for an emission factor are grams of compound emitted per kg dry biomass burned. In equation 3, $n$ is the number of carbon containing species measured, $N_i$ is the number of carbon atoms in species $i$, and $\Delta X_i$ is the excess mole fraction of species $i$ [*Yokelson et al.*, 1999; *Akagi et al.*, 2011]. Here we computed $C_T$ by allowing $i$ in equation 3 to cycle over $CO_2$, CO, and $CH_4$ ($n = 3$). We assumed the fraction of carbon in combusted fuels, $F_C$, was 0.45 [*Santin et al.*, 2015], but note that $F_C$ can range from $0.45 - 0.55$ [*Akagi et al.*, 2011].

$$EF_X = \frac{MM_X}{MM_C} * F_C * 1000 * \frac{ER_X}{C_T} \tag{2}$$

$$C_T = \sum_{i=1}^{n} N_i * \frac{\Delta X_i}{\Delta CO_{2i}} \tag{3}$$

We also calculated the MCE for each fire-affected interval. Modified combustion efficiency is defined as the excess mole fraction of $CO_2$ divided by the sum of the excess mole fractions of CO and $CO_2$ [*Ward and Radke*, 1993]. MCE was used to separate intervals into three categories: smoldering, mixed, or flaming. These categories reflect the dominant combustion process contributing to trace gas anomalies at the CRV tower during the summer of 2015. Periods with an MCE less than 0.85 were considered to consist of mostly smoldering combustion, periods with a MCE of greater than or equal to 0.85 and less than 0.92 were classified as consisting of a mixture of smoldering and flaming combustion, and periods with an MCE greater than 0.92 were classified as flaming [*Urbanski*, 2014]. We performed this classification to allow for a visualization of how the sampled combustion processes varied from interval to interval (and day to day) during the 2015 fire season.

**2.3 Transport Modeling**

We coupled a fire emission model, the Alaskan Fire Emissions Database (AKFED) [*Veraverbeke et al.*, 2015] with an atmospheric transport model, the Polar Weather Research and Forecasting Stochastic Time Integrated Lagrangian Transport model

(PWRF-STILT) [*Henderson et al.*, 2015] to estimate fire contributions to trace gas variability at the CRV tower, following *Wiggins et al.* [2016]. For this application, STILT [*Lin et al.*, 2007] was used to estimate the adjoint of PWRF [*Skamarock et al.*, 2005; *Chang et al.*, 2014; *Henderson et al.*, 2015] during the summer of 2015 at the location of the CRV tower, to generate surface influence functions that relate surface ecosystem fluxes from Alaska to trace mole fractions at CRV. These gridded influence functions are known as footprints and have units of mole fraction per unit of surface flux (ppm/($\mu$mol m$^{-2}$ s$^{-1}$)). Here we emitted fire emissions into the surface influenced volume of PWRF-STILT, which extends from the surface to the top of the planetary boundary layer, with the assumption that fire emissions were equally distributed within the planetary boundary layer [*Turquety et al.*, 2007; *Kahn et al.*, 2008]. In a previous study using the same tower, a sensitivity study revealed plume injection height contributed only minimally to variability in remote fire CO predictions at CRV with PWRF-STILT [*Wiggins et al.,* 2016].

Daily burned area in AKFED was mapped using thermal imagery from the Moderate Resolution Imaging Spectroradiometer (MODIS) within fire perimeters from the Alaska Large Fire Database. Both above and belowground carbon consumption were modeled as a function of elevation, day of burning, pre-fire tree cover, and difference normalized burn ratio (dNBR) measurements derived from 500 m MODIS surface reflectance bands [*Veraverbeke et al.*, 2015]. AKFED predicted carbon emissions from fires with a temporal resolution of 1 day and a spatial resolution of 450 m. We regridded AKFED to the same spatial resolution as the atmospheric transport model (0.5°) for the model coupling. To account for diurnal variability in emissions, here we imposed a diurnal cycle on daily emissions following *Kaiser et al.* [2009], where the diurnal cycle was the sum of a constant and a Gaussian function that peaks in early afternoon with 90% of emissions occurring during the day (hours 0600 to 1800 local time) and 10% at night (hours 1800 to 0600 local time). Analysis of the sum of fire radiative power from all of the fire detections in the MODIS MCD14ML C6 product showed that 83% of detected fire activity occurred during the daytime overpasses (10:30am and 1:30pm) relative to the sum across both daytime and nighttime overpasses during the 2015 Alaskan wildfire season (data not shown). The satellite observations, although temporally sparse (with only 4 over passes per day), were broadly consistent with the diurnal cycle we prescribed for fire emissions in the model.

We convolved AKFED with the PWRF-STILT footprints to determine individual fire contributions to CO anomalies at the CRV tower. This was achieved by calculating the total CO contribution from each individual 0.5° grid cell from the AKFED × PWRF-STILT combined model and utilizing the fire perimeters from the Alaska Large Fire Database (data provided by Bureau of Land Management (BLM) Alaska Fire Service, on behalf of the Alaska Wildland Fire Coordinating Group (AWFCG) and Alaska Interagency Coordination Center (AICC)) to identify the location of individual fires. AKFED uses the same fire perimeter database for burned area and carbon emissions estimates [*Veraverbeke et al.,* 2015]. We determined an individual fire's contribution to CO at the CRV tower by setting all emissions in AKFED for a particular grid cell to zero and rerunning the model coupling with PWRF-STILT. The difference between the original model and the updated coupling that excluded emissions from an individual fire was equal to the individual fire's contribution to CO at CRV tower, when integrated over the 2015 fire season. Due to the 0.5° grid cell size used for model coupling, more than one fire perimeter existed in some of the individual grid cells. In these cases, the contribution for each fire was determined by weighting the total signal contribution by fire size.

We also used the footprints from PWRF-STILT to quantify the contribution of day and night emissions and mean transit times (Figure 3). The footprints are on a 0.5° latitude-longitude grid with a temporal resolution of 1 h during hours 0600 to 1800 (day) local time and 3 h during hours 1800 to 0600 local time (night). These functions provide an estimate of the impact of upwind surface fluxes at different times in the past on CRV tower trace gas mole fraction measurements at a given time. We analyzed the footprints for each interval in Table 2 to confirm CRV tower observations integrated emissions from multiple fires and captured variability in emissions across the diurnal fire cycle. Overall, we found that 73% of the summer fire CO anomaly at CRV originated from fire emissions that occurred during the day (0600 to 1800 local time) and 27% from emissions that occurred at night (1800 –

0600 local time). The footprints associated with each emission factor interval also were used to determine how much of the signal was coming from burning on previous days. We found that more than 99% of the fire emissions that influenced CO at CRV occurred within 3 days of an sampling interval used to derive an emission ratio, with 76% occurring within the first 24 hours, 21% during the next 24 hours, and 3% occurring three days prior to the sampling interval.

**2.4 Comparison with Previous CO Emission Ratio Studies**

To investigate the possible influence of sampling strategy and differences associated with sampling in different ecosystem types, we compiled available studies that report CO emission ratios for boreal forest fires and organized the studies into several categories with common characteristics, including aircraft sampling of North American boreal forest wildfires, aircraft sampling of North American boreal forest management or prescribed fires, combustion of North American boreal forest fuels measured in the laboratory, and sampling of Siberian boreal fires from both aircraft and surface platforms (Table 1). In our analysis we included original studies reported in Andreae (2019) and Akagi et al. (2011) and several others we found in a literature survey.

**3 Results**

**3.1 Emission Factors and Modified Combustion Efficiency**

During the 2015 Alaska fire season, we observed synchronized enhancements of CO, $CH_4$, and $CO_2$ well above background concentrations at CRV from DOY 173 – 196 (Figure 4). We identified 55 individual fire-affected intervals in the measurement time series (that each span about 47 minutes) and used these intervals to calculate emission ratios, emission factors, and MCE (Figure 5; Table 2). $CO/CO_2$ emission ratios ranged from 0.025 to 0.272 and $CH_4/CO_2$ emission ratios ranged from 0.002 to 0.020. MCE varied between 0.786 and 0.976 (Table 2). CO emission factors ranged from 25 to 223 g $kg^{-1}$ dry biomass burned, and $CH_4$ emission factors ranged from 1.2 to 10.7 g $kg^{-1}$ dry biomass burned.

The mean $CO/CO_2$ emission ratio was $0.141 \pm 0.051$, the mean CO emission factor was $127 \pm 40$ g $kg^{-1}$ dry biomass burned, and the mean MCE was $0.878 \pm 0.039$. Concurrently, the mean $CH_4/CO_2$ emission ratio was $0.010 \pm 0.004$ and the mean $CH_4$ emission factor was $5.32 \pm 1.82$ g $kg^{-1}$ dry biomass burned.

A strong linear relationship existed between the $CH_4$ emission factor and MCE across the different sampling intervals (Figure 6). Linear relationships between $CH_4$ emission factors and MCE have also been observed in previous studies [*Yokelson et al.*, 2007; *Burling et al.*, 2011; *Van Leeuwen and van der Werf*, 2011; *Yokelson et al.*, 2013; *Urbanski*, 2014; *Smith et al.*, 2014; *Strand et al.*, 2016, *Guerette et al.*, 2018]. The relationship shown in Figure 6 implies MCE can be used to estimate $CH_4$ emissions (and emissions of other closely related trace gases) from North American boreal forest wildfires when measurements of $CH_4$ are not available.

We classified each fire-affected sampling interval as being associated with smoldering, mixed, or flaming combustion processes using thresholds on MCE. This analysis revealed that intervals with different combustion phases were interspersed throughout the fire season, with no clear progression over time, or clustering of flaming or smoldering processes during periods with high or low levels of burning. We identified 12 smoldering intervals, 37 mixed intervals, and 6 flaming intervals throughout the fire season (Figure 5, with examples shown in Figure 7). Smoldering intervals had a mean $CO/CO_2$ ratio of $0.214 \pm 0.030$, a mean CO emission factor of $183 \pm 21$ g $kg^{-1}$ dry biomass burned, a mean $CH_4/CO_2$ ratio of $0.014 \pm 0.003$, a mean $CH_4$ emission factor of $6.89 \pm 1.18$ g $kg^{-1}$ dry biomass burned, and a mean MCE of $0.824 \pm 0.020$. Mixed intervals consisting of both smoldering and flaming combustion had a mean $CO/CO_2$ emission ratio of $0.131 \pm 0.024$, a mean CO emission factor of g $kg^{-1}$ dry biomass

burned, a mean $CH_4/CO_2$ emission ratio of $0.010 \pm 0.003$, a mean $CH_4$ emission factor of $5.28 \pm 1.51$ g kg$^{-1}$ dry biomass burned, and a mean MCE of $0.884 \pm 0.019$. Flaming intervals had a mean $CO/CO_2$ emission ratio of $0.060 \pm 0.020$, a mean CO emission factor of $59 \pm 19$ g kg$^{-1}$ dry biomass burned, a mean $CH_4/CO_2$ emission ratio of $0.004 \pm 0.001$, a mean $CH_4$ emission factor of $2.49 \pm 0.78$ g kg$^{-1}$ dry biomass burned, and a mean MCE of $0.944 \pm 0.018$ (Table 3).

In our primary analysis described above, each individual fire-influenced interval used to compute an emission ratio was weighted equally in computing a season-wide mean. As a sensitivity analysis, we computed the mean emission ratios weighting each interval according to its mean $\Delta CO$ mole fraction, and, alternately, according to its mean $\Delta CO_2$ mole fraction. Weighting by $\Delta CO$ caused the CO emission ratio to increase from 0.141 to 0.146 but did not change the $CH_4$ emission ratio. Weighting by $\Delta CO_2$ caused the emission ratios to slightly increase, yielding a CO emission ratio of 0.144 and, again, no change in the $CH_4$ emission ratio. Although the variation introduced from different weighting approaches was relatively small, the analysis highlights the challenge of combining information from different individual fires, and the importance of moving toward flux-weighted estimates in future work.

**3.2 The Influence of Individual Fires on Trace Gas Variability at the CRV Tower**

The forward model simulations combining AKFED fire emissions with PWRF-STILT confirmed that the elevated CO signals at the CRV tower can be attributed primarily to boreal forest fire emissions (Figure 8) and not to fossil fuels or other CO sources. The AKFED model had a Pearson's correlation coefficient of 0.61 with observed daily mean CO and had a low bias of approximately 7%. Differences between the model simulations and observations were likely caused by errors in the magnitude and timing of fire emissions within AKFED as well as the limited spatial resolution and incomplete representation of atmospheric transport within PWRF-STILT. Nevertheless, the broad agreement between the model and the observations, including the timing of the large burning interval between DOY 173 and 179, provides some confidence that our model can be used to explore the influence and contribution of individual fires.

We identified 34 individual fires that contributed to at least 1% of the CO mole fraction time series at CRV tower over the entire 2015 fire season (Figure 9; Figure 10; Table 3). The average distance of these fires from the CRV tower, weighted by their fractional contribution, was $259 \pm 134$ km. Most of the fires were located to the west of Fairbanks, in the direction of the prevailing summer surface winds. This analysis revealed that the CRV tower was sufficiently downwind to measure the integrated impact of multiple fires on regional trace gas concentration anomalies, sampling air masses that were mixed through the full planetary boundary layer and across several day-night cycles. The total CO emitted from these fires accounted for 75% of the excess CO mole fraction signal during DOY 160 – 200. The remaining CO signal originated from many smaller fires that were widely distributed across interior Alaska. The Tozitna fire was responsible for the greatest percentage of the total CO anomaly integrated over the 2015 fire season at the CRV tower (accounting for more than 10% of the integrated CO anomaly at CRV). The fires that contributed the most to the CO anomaly at CRV tower were not necessarily the closest fires to the tower or the largest fires of the 2015 fire season in terms of burned area. Combined, however, this set of 34 fires accounted for 0.97 Mha, or approximately 46% of the total burned area reported during the 2015 fire season [*Veraverbeke et al.*, 2017].

**3.3 Comparison of emission ratios between sampling strategies**

Previous studies sampled a total of 45 individual boreal forest fires for $\Delta CO/\Delta CO_2$ emission ratios or CO emission factors, and additional measurements have been made by combusting fuels in a laboratory setting. Solely considering emission ratio measurements from North American boreal forests (excluding boreal forests in Eurasia), the mean of aircraft sampling of wildfires

(0.102 ± 0.033, n=19) or management and prescribed fires (0.077 ± 0.022, n=14) were significantly lower than the mean derived

from tower measurements reported here along with earlier measurements from *Wiggins et al.* [2016] (0.141 ± 0.049, n=37) as

evaluated using a Student's t test. The mean emission ratio from Siberian boreal forest fires was 0.219 ± 0.048 (n=9), which was

significantly higher than the mean of emission ratios reported for boreal forest wildfires in North America (sampled either by

aircraft or tower).

**4 Discussion**

The most widely used emission factors for boreal forest fires are derived from syntheses that average together data from

individual field campaigns [*Andreae and Merlet*, 2001; *Akagi et al.*, 2011; *Andreae*, 2019]. Our mean emission factor for CO (127

± 40 g kg$^{-1}$ dry biomass burned) is similar to the mean reported in past syntheses for boreal forests, including estimates by *Andreae*

[2019] (121 ± 47 g kg$^{-1}$ dry biomass burned) and *Akagi et al.* [2011] (127 ± 45 g kg$^{-1}$ dry biomass burned). Emission factors for

$CH_4$ were also similar to the estimates reported in these syntheses. Considering boreal forests as a whole, our measurements provide

a partial validation of the approach taken in previous compilations, which have attempted to combine information from different

sampling strategies and boreal forest ecoregions. The broad level of agreement provides confidence in the estimates of emission

factors for non-conserved species that cannot be measured using a remote tower sampling approach.

The observations summarized in Table 1 also show there are several important differences in boreal forest emission ratios

that exist as a function sampling strategy and ecoregion. Within North American boreal forests, the CRV observations we analyzed

here provide evidence that smoldering combustion contributes more to CO emissions than what has been estimated from previous

aircraft studies. Specifically, our mean CO emission ratio from the CRV tower is 39% higher (and significantly different at a $p <$

0.01 level using a Student's t test) than the mean derived from aircraft based measurements of 19 North American boreal wildfires

(Table 1). Although differences in reported emission ratios are expected between aircraft and ground based sampling approaches

[*Christian et al.*, 2007; *Burling et al.*, 2011; *Akagi et al.*, 2014; *Collier et al.*, 2016; *Benedict et al.*, 2017; *Selimovic et al.*, 2019],

several features of the CRV tower sampling are conducive to providing a regionally-representative mean estimate of emission

ratios during the 2015 Alaska fire season. First, we note that the CRV tower was located at a higher elevation (611 m above sea

level) than the core fire complex located in western Alaska and several hundreds of kilometers downwind. Multi-angle Imaging

SpectroRadiometer (MISR) satellite observations from Alaskan wildfires indicate most fire plumes reside within the planetary

boundary layer, which is typically between 1 and 3 km during midday in summer [*val Martin et al.*, 2010; *Wiggins et al.*, 2016].

Combining this vertical length scale with the mean horizontal distance of the 34 fires that most influenced CO at CRV (259 km),

we obtain a factor of about 100 for a back-of-the-envelope ratio of horizontal to vertical mixing processes. This ratio, together with

the simulated time delay of 1-2 days between emission and detection of CO anomalies at CRV (Figure 3), imply that mesoscale

atmospheric circulation played an important role in averaging together trace gas emissions from multiple fires before the air masses

were sampled (Figure 10). As a result, observations from the CRV tower represent a temporal integration of fire emissions over

32 day-night burning cycles as well as a spatial integration across flaming combustion at active fire fronts along with residual

smoldering combustion in soils that often persists for days after a fire front moves through an area. Collectively, the fires sampled

at CRV appeared to experience time-varying environmental conditions that were less ideal for flaming combustion than the fire

plumes sampled in past work by aircraft. This finding is consistent with remote tower observations of the black carbon to CO ratio

measured for wildfires from temperate North America [*Selimovic et al.*, 2019].

In contrast with remote tower sampling, aircraft-based studies often sample fires that have a strong contribution from

flaming combustion, which releases enough energy to generate well-defined plumes at an altitude accessible by the aircraft. This

methodology provides an opportunity to comprehensively measure the vertical and horizontal distribution of emissions from an individual fire and their atmospheric evolution in a smoke plume. However, airborne sampling techniques are often limited to daytime periods with good visibility, making it difficult to comprehensively measure emissions over a diurnal cycle or over the full lifetime of a fire which may span several periods with inclement weather. Due to these sampling constraints, aircraft studies are less likely to measure emissions from less energetic smoldering combustion, since these emissions are more likely to remain near the surface [*Ward and Radke*, 1993; *Selimovic et al.,* 2019]. Emissions from smoldering boreal forest fires can sometimes be entrained in the convective columns of certain flaming fires and can be sampled by aircraft, but nighttime emissions or residual smoldering emissions from fires that have weak convective columns usually cannot be measured in this way [*Bertschi et al.,* 2003; *Burling et al.,* 2011]. While past studies have attempted to combine information from aircraft (more likely sampling flaming combustion phases) with laboratory observations of emissions from smoldering combustion [*Akagi et al.*, 2011], the balance of these processes is well known to be sensitive to environmental conditions that can rapidly change over the lifetime of a wildfire; this highlights the importance of designing sampling approaches that provide regionally-integrated estimates over the full duration of a wildfire event or a regional fire complex.

During the latter half of June and early July of 2015, weather in Alaska was very hot and dry, allowing for a record number of fires to rapidly expand in size, and yielding the second highest level of annual burned area in the observed record. The extreme fire weather conditions would be expected to reduce fuel moisture content, thus promoting crown fires and flaming combustion processes [e.g., *Sedano and Randerson*, 2014]. This raises the question of whether longer term monitoring of many normal and low fire years (which tend to co-occur in cooler and wetter conditions) would provide evidence for an even larger role of smoldering combustion compared to the estimates we report here for 2015. Another related question is whether even within a fire season, do day-to-day or week-to-week variations in fire weather influence variability in emission ratios? We explored this latter question with the datasets described here but were unable to uncover structural relationships between daily meteorological variables such as vapor pressure deficit and CO emission ratios. Together, these questions represent important directions for future research and emphasize the critical need of sustained long-term support for trace gas monitoring networks and field campaigns.

As a function of ecoregion, emission ratios from fires in boreal Eurasia tend to be higher than emission ratios from fires in boreal North America, and are significantly different than tower or aircraft observations from North America when compared using a Student's t test. Although more measurements are needed, higher CO emission ratios for Siberian fires appear consistent with past work showing that boreal fire behavior is considerably different between North American and Eurasian continents as a consequence of differences in tree species and their impacts on fire dynamics [*Goldammer and Furyaev*, 1996; *Cofer et al.,* 1998]. Notably, as consequence of the presence of black spruce in many boreal forests of North America, fires tend to burn with a higher fire radiative power and faster spread rate, traveling through the crowns of trees and inducing higher levels of tree mortality [*Rogers et al.*, 2015]. This occurs because black spruce is a well-known fire embracer, retaining dead branches that serve as ladder fuels and carry fire into the overstory. Black spruce trees are absent from Siberia, where many pine and larch tree species lack ladder fuels and are known to be fire resistors. In Siberian ecosystems ground fires are more common [*Korovin*, 1996; *Rogers et al.*, 2015], a finding that appears consistent with the higher CO emission ratios (and larger contribution of smoldering combustion) shown in Table 1. Although emission factors from the Siberian boreal forest are often grouped together with emission factors from North American boreal forest in biome-level syntheses [e.g., *Andreae*, 2019], both emission ratio and remote sensing observations of fire severity suggest there may be enough evidence to separate these two ecoregions in future syntheses.

In Table 1 we also separated aircraft-based studies that measured emissions from wildfires from those that measured emissions from prescribed slash and land management fires, where trees are bulldozed, dried and intentionally arranged to promote maximum fuel consumption [*Cofer et al.*, 1990; *Cofer et al.*, 1998]. Land management fires consume dried aboveground fuels with

a different fuel structure and moisture content than fuels consumed in a wildfire, where combustion from soil organic material layers is a dominant component of bulk emissions [Boby et al., 2010; *Dieleman et al.*, 2020]. Although the number of land management fires is relatively small, the mean from these studies suggest flaming processes are a more important contributor to this fire type than for wildfires, and some consideration of this difference should be factored into regional and global syntheses.

Several additional studies report emission ratios from laboratory combustion of fuels collected from North American boreal forests including biomass samples from black spruce, white spruce, and jack pine, as well as moss and surface organic material (duff). The laboratory studies have considerable variability that can be attributed to the type of fuel combusted and fuel moisture content. This work indicates duff consumption yields higher emission ratios for CO and $CH_4$ than combustion of black spruce or jack pine needles and other fine fuels [*Bertschi et al.*, 2003; *McMeeking et al.,* 2009; *Burling et al.*, 2011]. The fuels used in laboratory studies are usually dried and burned individually, although some studies have attempted to mimic natural fires by placing dried fine fuels on top of damp fuels that undergo residual smoldering combustion [*Bertschi et al.,* 2003]. The structure, composition, and moisture content of fuels are well known as key drivers of the composition and magnitude of emissions. Although these laboratory studies provide valuable information on emissions from individual fuel components, they are not able to capture the full complexity of a wildfire.

In the context of these comparisons among ecoregions and sampling strategies, it is important to recognize that tower-based sampling strategies, including the methodology presented in this study, have important limits. Ground-based sites may potentially miss some of the emissions injected above the planetary boundary layer. The fixed nature of this sampling technique also restricts the range of sampling, because towers can only monitor upwind fires. Although the tower-based sampling strategy allows for integration of emissions from fires across a range of environmental conditions and at different stages of fire life cycles, it may not allow for emission ratio measurements of non-conserved species, including particulate matter and many fire-emitted volatile organic compounds that have short lifetimes. The technique is also subject to higher uncertainty in the definition of background mole fractions for fire-affected trace gases, because of the dilution and mixing of fire emissions that occurs during transport. Thus, tower may not be a feasible or effective sampling methodology during years with low fire activity.

## 5 Conclusions

Using a remote tower downwind of a large regional fire complex in interior Alaska, we measured CO and $CH_4$ emission factors from about 34 individual fires during the summer of 2015. This is comparable to the number of individual wildfires sampled in North America in previous studies. Our results indicate smoldering combustion processes in North American boreal forest fires contribute to more trace gas emissions than previous estimates derived from aircraft sampling. Together, the two-month near continuous time series of $CO_2$, CO, and $CH_4$, along with the derived emission ratios reported here, may provide a means to test models that couple together fire processes, emissions, and regional atmospheric transport.

Comparison of emission ratios reported here with observations derived other sampling strategies and ecoregions in northern boreal forests provides directions for reducing future uncertainties. For boreal North America, our analysis of CRV tower observations indicate CO emission ratios are likely higher what would be inferred from previous studies, although questions remain regarding the representativeness of remote tower-based sampling. Given recent increases in data density for North America and improvements in our understanding of differences in tree species composition and fire dynamics between North America and Eurasia, it may be possible to reduce uncertainties in future syntheses by separately reporting emission factors for the two continents. More data, particularly for Siberian fires, however, is needed to assess whether the continental differences in emission ratios noted here are robust. Long-term monitoring from remote towers has the potential to provide new information about fire

complexes in other biomes, integrating across day-night variations in fire behavior, periods with different environmental conditions, and across multiple fires in different stages of growth and extinction. In this context, more work is needed to find ways to combine tower and aircraft sampling to attain accurate estimates of the total budget of fire-emitted trace gases and aerosols (i.e., estimating flux-weighted emission factors), given the large differences in data density and the different strengths and weaknesses of the two approaches. To make progress on this issue, a closer integration is needed in future field campaigns between measurements of pre-fire ecosystem state, fire behavior (temperature, fire radiative power, and spread rate), measurements of emissions composition, and post-fire sampling of fuel consumption and combustion completeness during times when fire dynamics are fundamentally different. This coordination across disciplines in both study design, data analysis, and modeling is rare and may provide a path toward creating the observations needed to dynamically model the temporal evolution of the chemical composition of wildland fire emissions over the lifetime of an individual fire and, within a region, during different phases of a fire season.

**Data availability**

The CRV tower observations and footprints used in our analysis are archived at the U.S. Oak Ridge National Laboratory Distributed Active Archive Center for Biogeochemical Dynamics (http://dx.doi.org/10.3334/ORNLDAAC/1316).

**Author contribution**

Conceptualization: EBW and JTR. Data curation: CEM, SV, JMH, and AA. Investigation and Visualization: EBW. Writing – original draft: EBW and JTR. Writing – review and editing: Everyone.

**Competing interest**

The authors declare that they have no conflict of interest.

**Acknowledgements**

E.B.W. thanks the U.S. National Science Foundation for a Graduate Research Fellowship (NSF 2013172241). The trace gas observations, fire emissions time series, and WRF-STILT model were created through funding support to NASA's CARVE field program led by C. Miller. JTR acknowledges additional NASA support from CMS (80NSSC18K0179) and IDS (80NSSC17K0416) programs. We thank the staff of the NOAA Fairbanks Command and Data Acquisition Station that hosts CRV, and especially Frank Holan and Marc Meindl, for their technical support. We also thank NOAA/ESRL/GMD staff, especially Phil Handley, Jon Kofler, and Tim Newberger for ongoing remote maintenance of CRV.

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

# 1  Figures

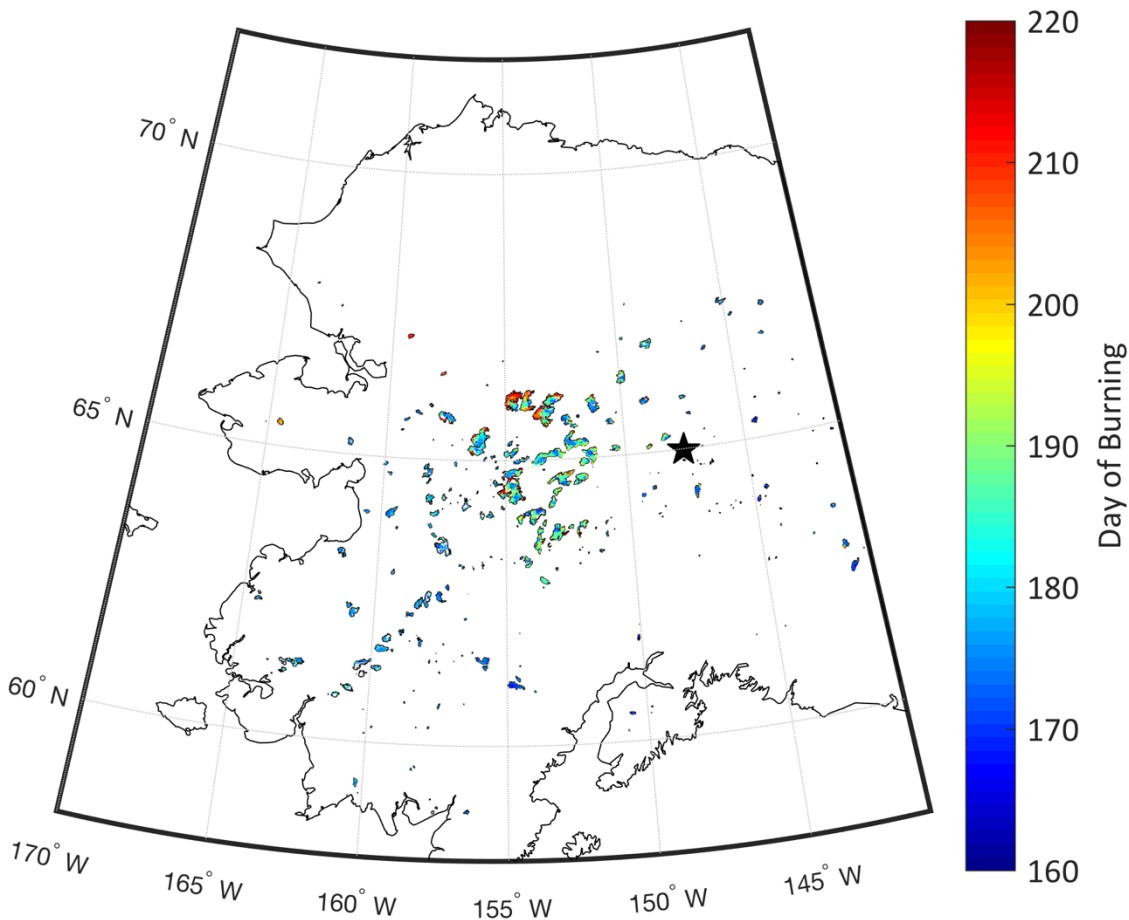

3  **Figure 1.** The location of wildfires in Alaska during 2015, with color representing the day of burning estimated from the Alaska
4  Fire Emissions Database (AKFED). The black star denotes the location of CRV tower.

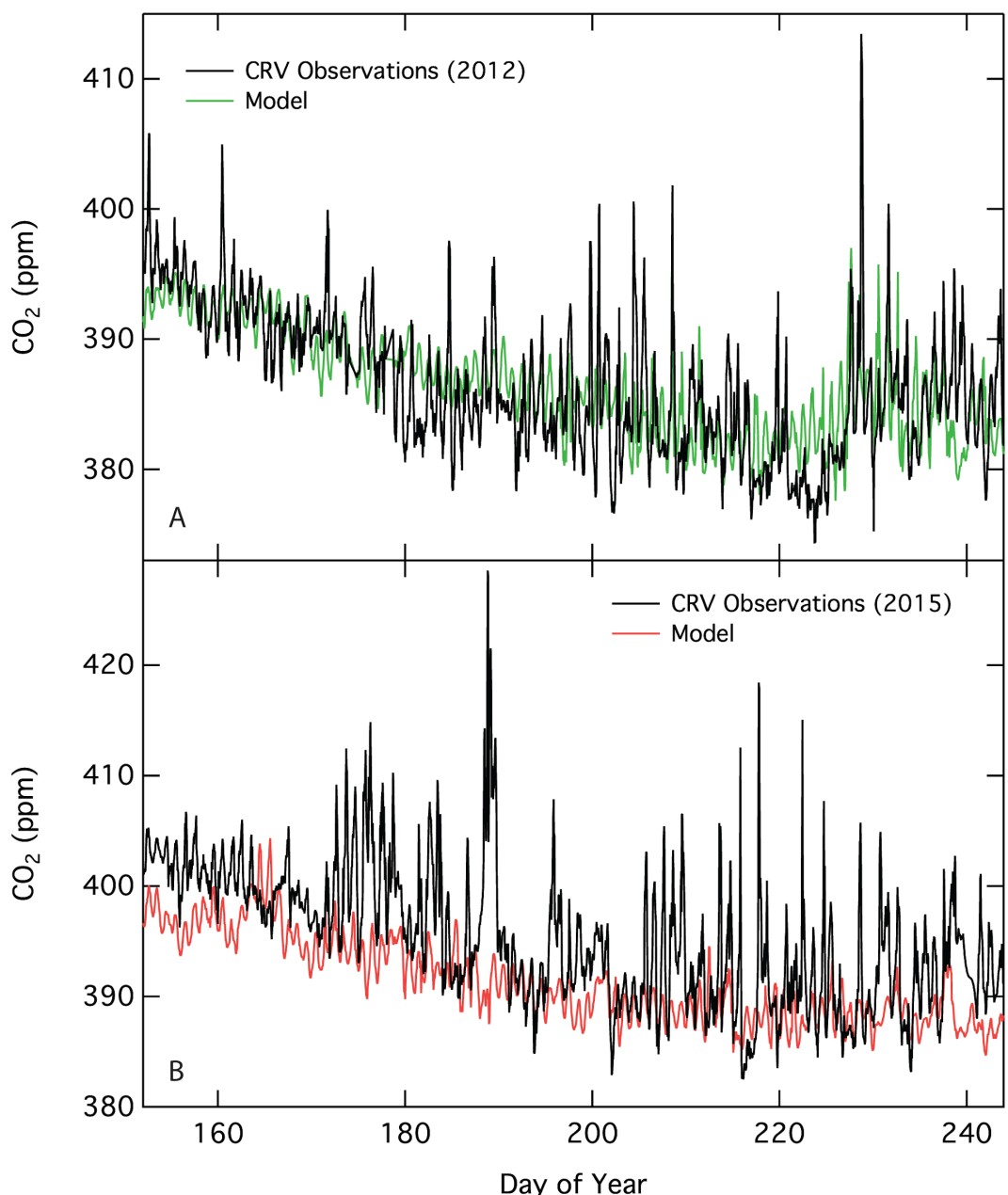

Figure 2. A) Observations of $CO_2$ mole fraction at the CRV tower in 2012 (black line) along with model estimates of the $CO_2$ background (green line) at CRV using the approach described in the main text. Very few fires occurred during 2012, and as a consequence most of the $CO_2$ variability in the observations and in the model are associated with terrestrial net ecosystem exchange. B) In 2015 wildfires in interior Alaska contributed significantly to $CO_2$ variability at the CRV tower, causing positive anomalies in the observations shown in black, particularly between days 170 and 190. The modeled background for 2015 is shown in red. The $CO_2$ mole fraction observations and model estimates have a 1 hour temporal resolution.

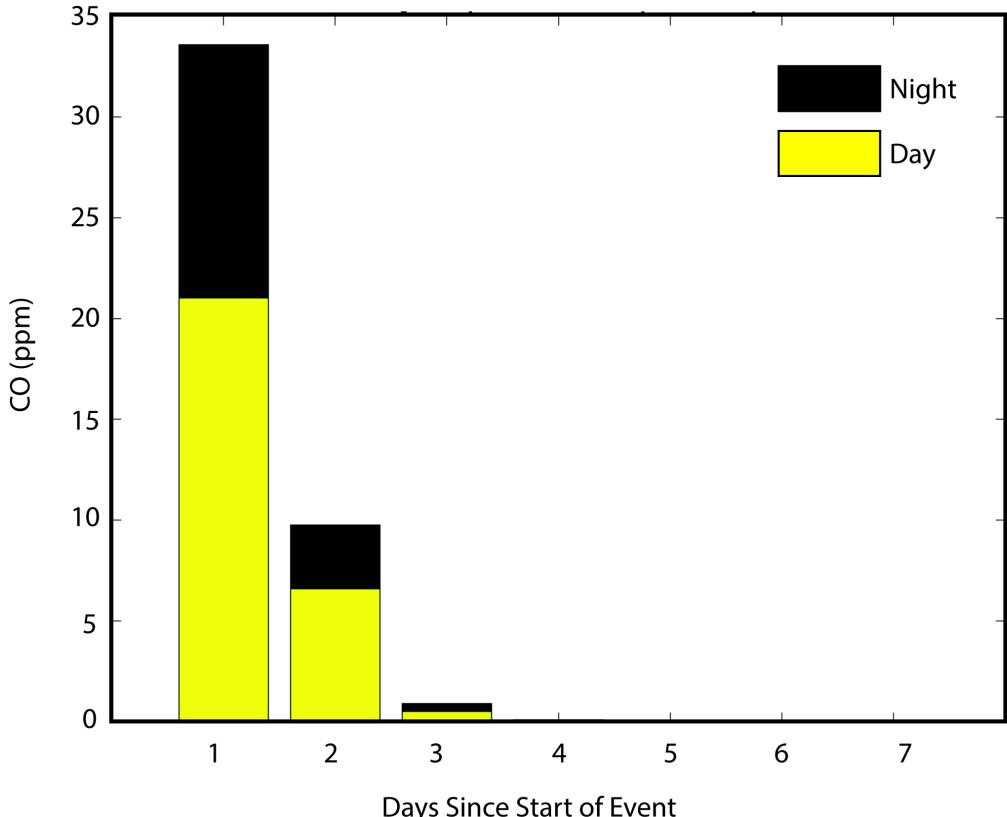

2 **Figure 3.** Distribution of transit times representing thedifference between the time when CO was emitted by a fire and the time the

3 CO anomaly reached the CRV tower, as estimated by multiplying footprints from PWRF-STILT with fire emissions from AKFED.

4 Only times when fire emission ratios were calculated were used in the analysis.

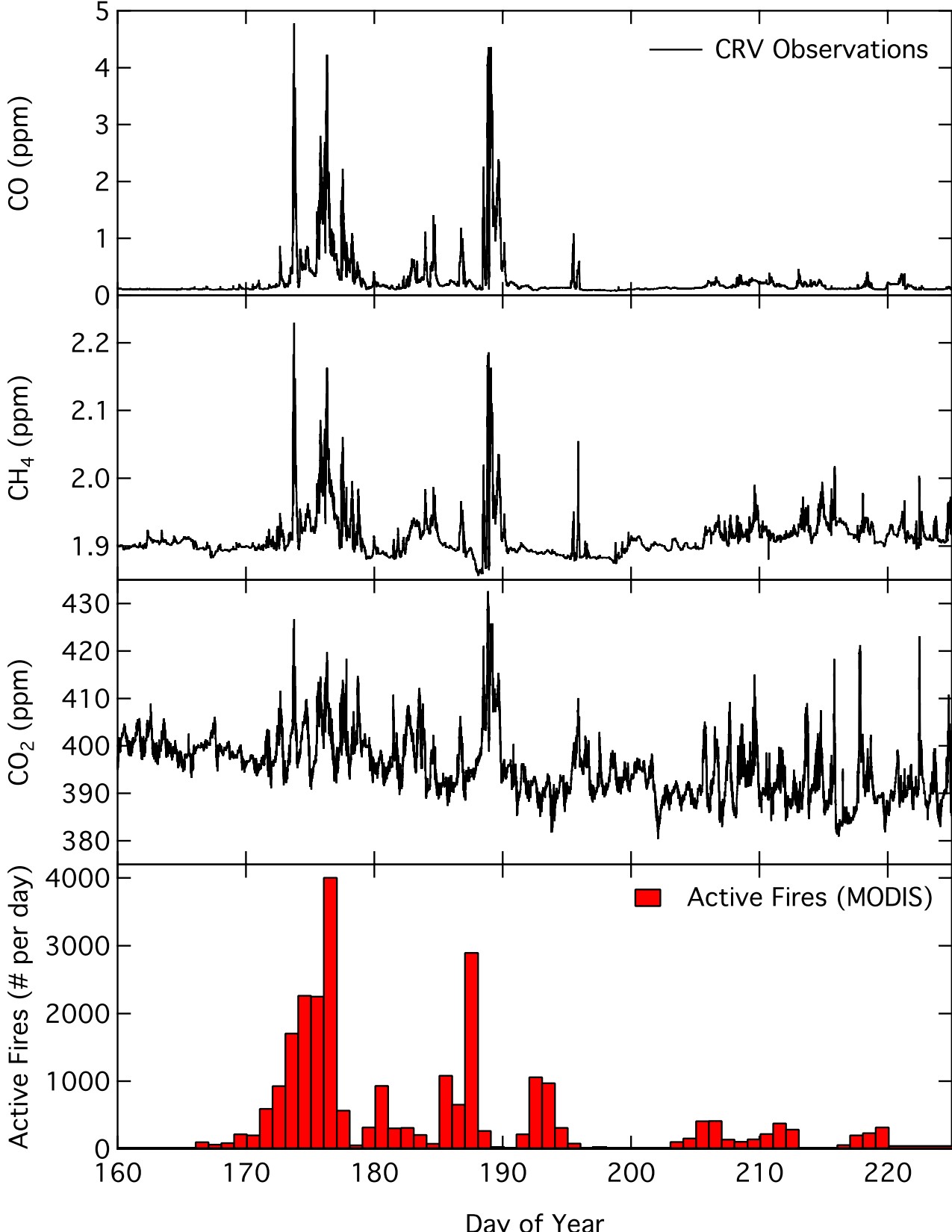

2  **Figure 4.** Trace gas observations at the CRV tower during the summer of 2015 for A) CO, B) $CH_4$, and C) $CO_2$ mole fractions The

3  trace gas observations are shown at a 30 s temporal resolution. Daily active fire detections derived from the MODIS sensors on

4  Terra and Aqua satellites (MCD14ML C6) are shown in panel D.

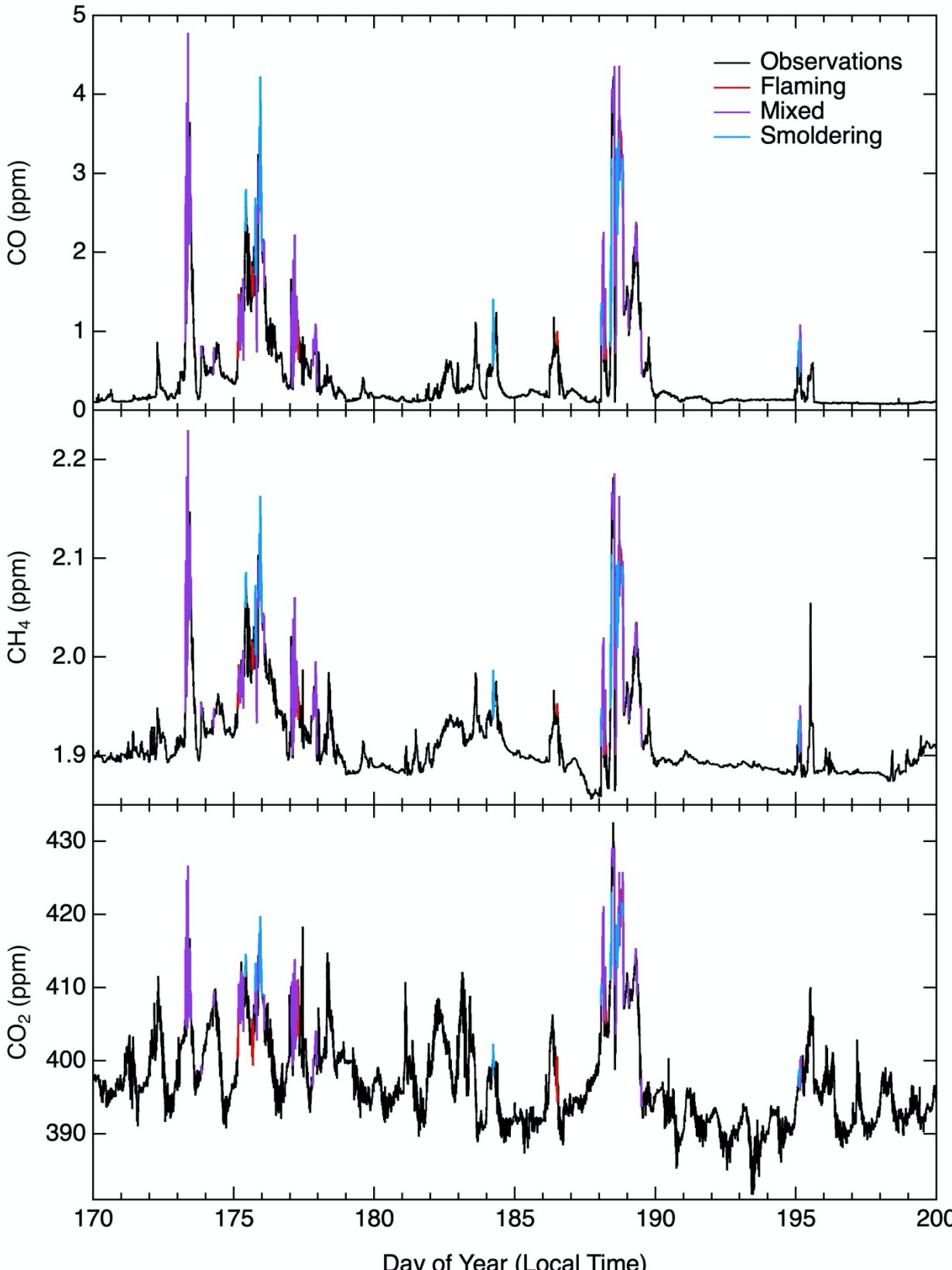

2  **Figure 5.** CRV tower observations of A) CO, B) CH$_4$, and C) CO$_2$ are shown along with intervals used to calculate emission ratios
3  (shown in color). The primary combustion process is noted with blue for smoldering, purple for mixed, and red for flaming. The
4  trace gas observations are shown at a 30 s temporal resolution.

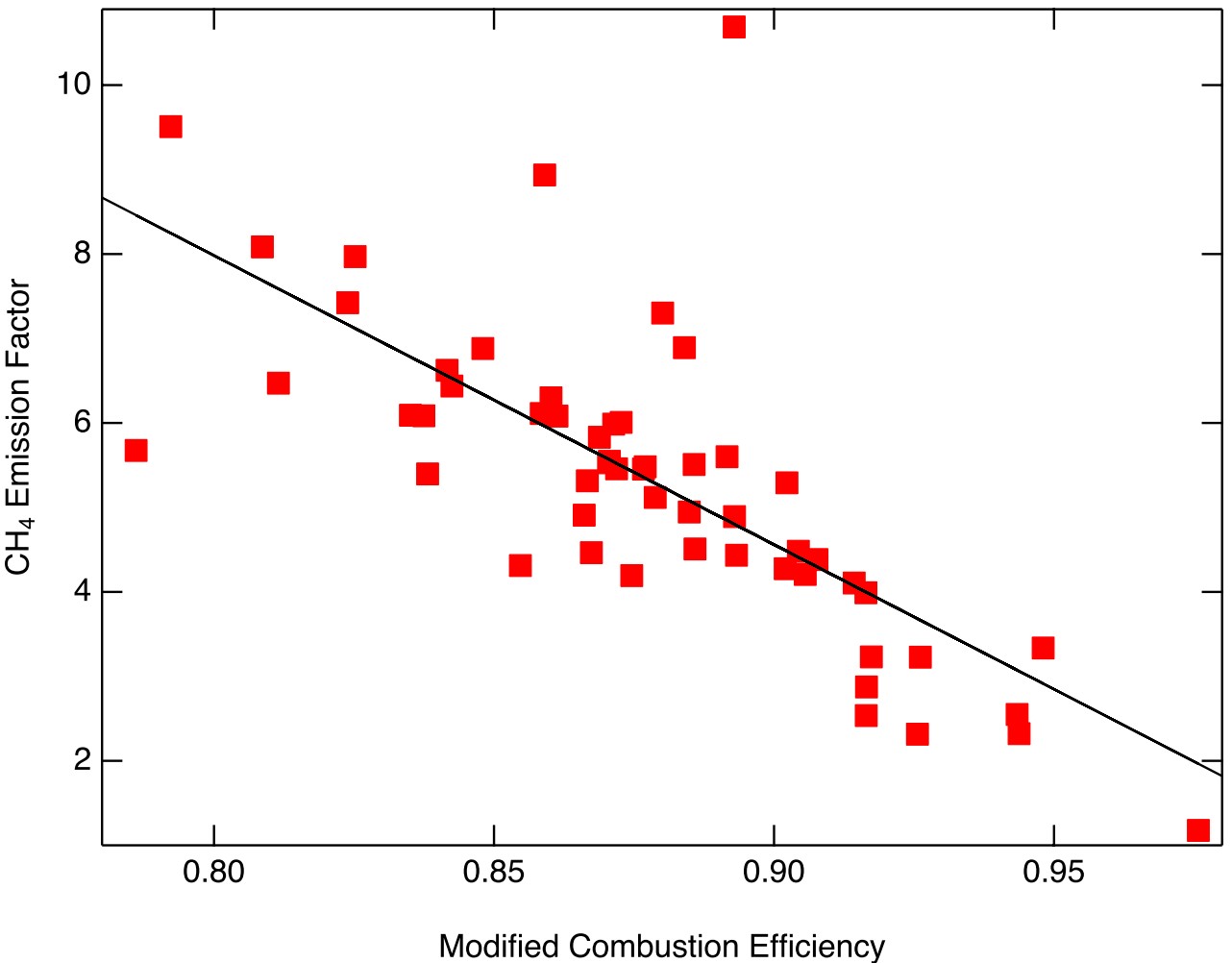

2 **Figure 6.** Relationship between $CH_4$ emission factor and modified combustion efficiency (MCE). The strong linear relationship

3 indicates that periods with more smoldering combustion (and a lower MCE) produce significantly higher levels of $CH_4$ emissions.

4 The relationship was defined by a slope of -46.77 ± 4.70, a Y intercept of 46.37 ± 4.13 g $kg^{-1}$ dry biomass burned, an $r^2$ of 0.54,

5 and a significance value of p <0.01.

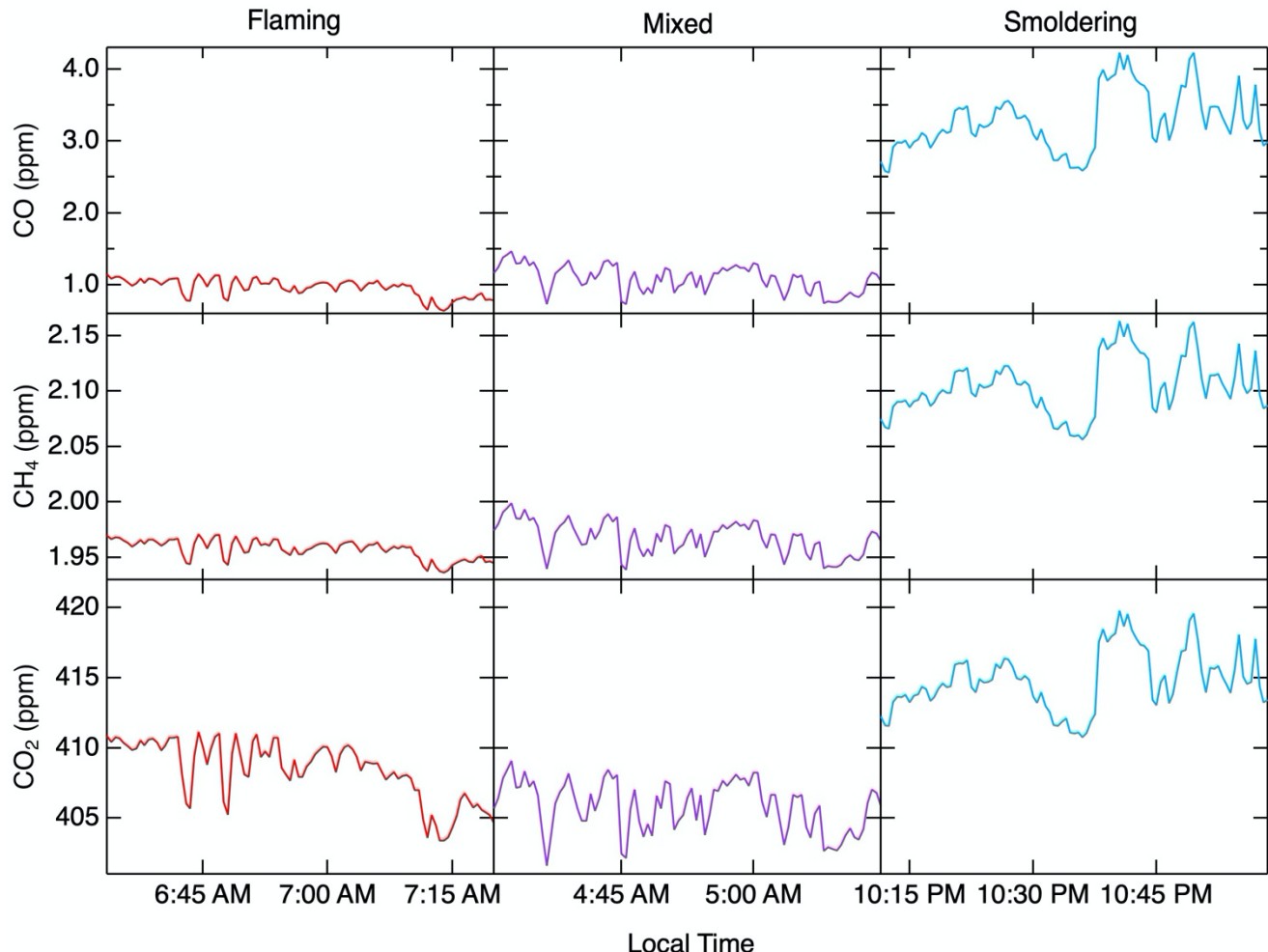

2 **Figure 7.** Examples of intervals used to calculate emission ratios. The flaming combustion example is from DOY 177, the mixed

3 example is from DOY 177, and smoldering example is from DOY 175. These intervals correspond to events 27, 25, and 19 in

4 Table 2. The trace gas measurements are shown at a 30 s temporal resolution.

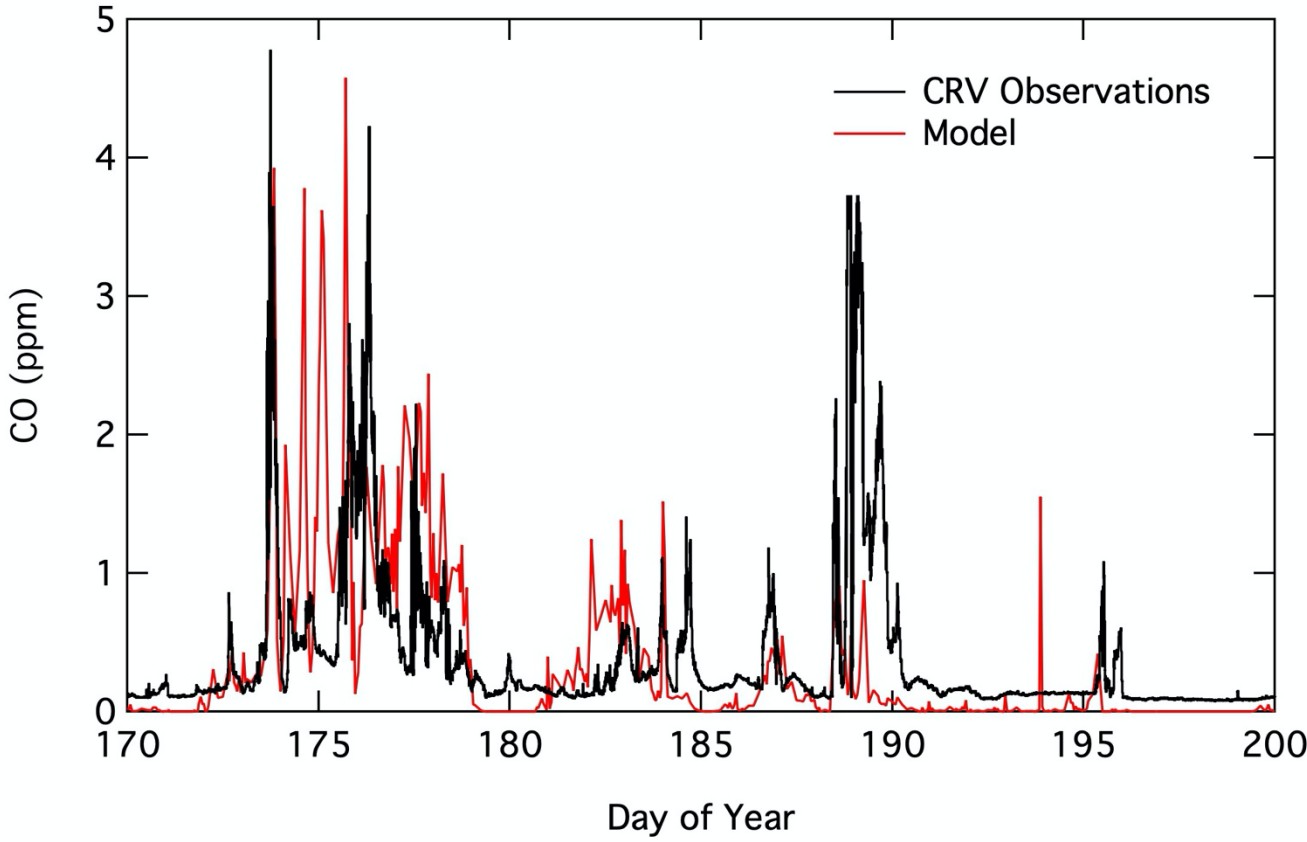

**Figure 8.** CRV observations of CO (black) compared with the modeled CO anomaly from fires (red) derived from the PWRF-
STILT atmospheric model driven by AKFED fire emissions. The trace gas observations and model predictions are shown at a 1
hour temporal resolution.

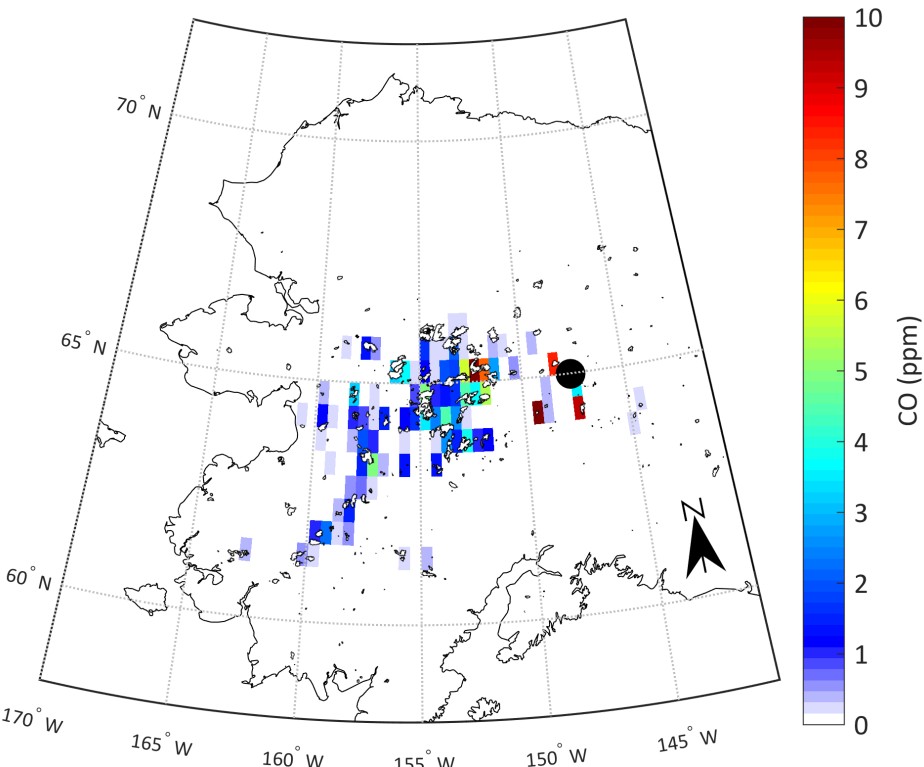

2 **Figure 9.** Individual fire contributions to the total fire season integral of CO anomalies measured at the CRV tower, as determined

3 by convolving footprints from PWRF-STILT with fire emissions from AKFED. The location of CRV is shown as a black circle.

4 Fire perimeters are shown in black.

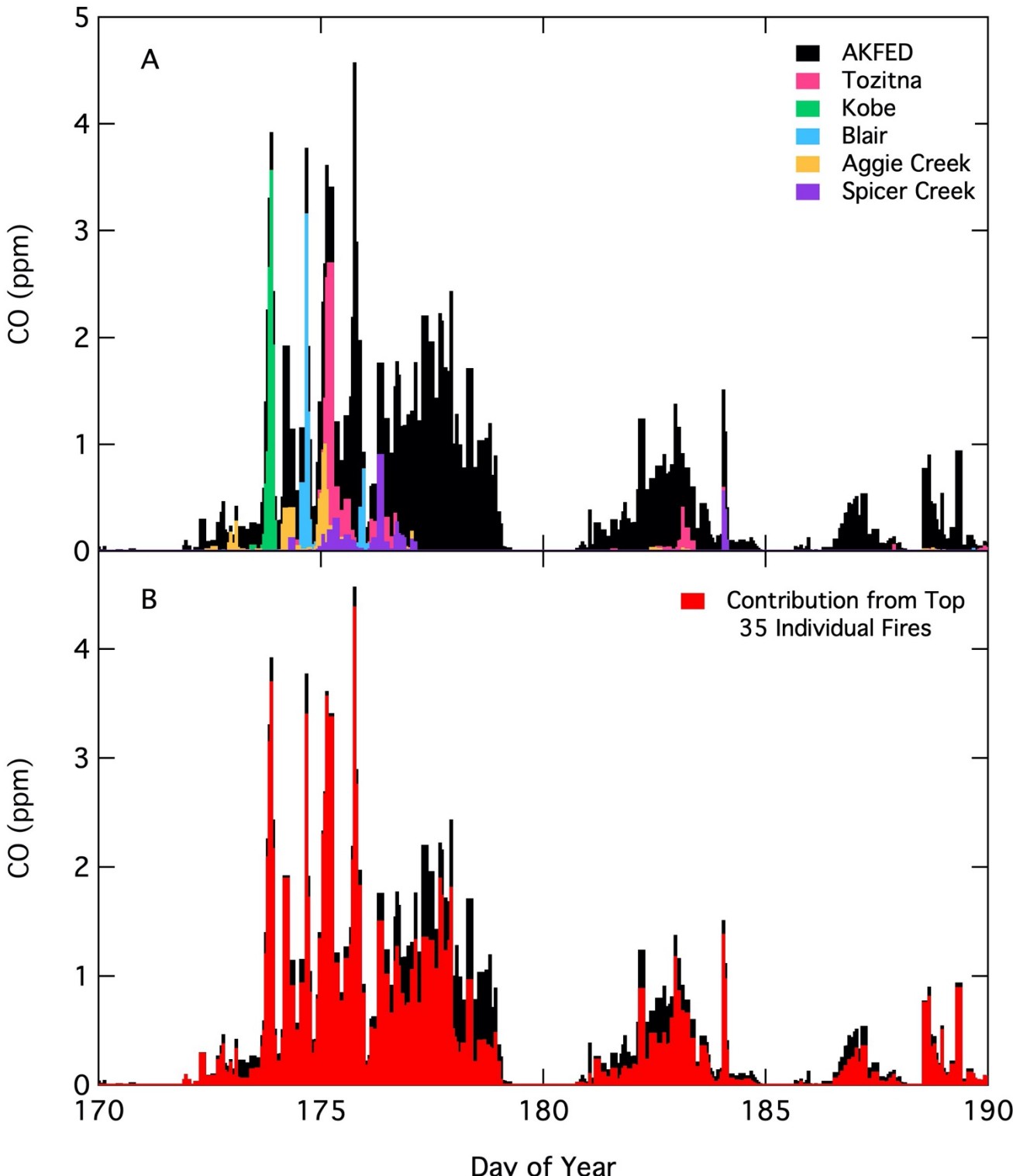

**Figure 10.** A) Top 5 individual fire contributions to the CO anomalies simulated at the CRV tower. The black line shows original

PWRF-STILT × AKFED model, pink denotes contributions from the Tozitna fire, green from the Kobe fire, blue from the Blair

fire, gold from the Aggie Creek fire, and purple from the Spicer Creek fire. B) The total CO anomaly from the 34 fires that

contributed to at least 1% of the modeled CO anomaly at CRV tower (red) compared to sum of all fire shown in black derived

from original PWRF-STILT × AKFED simulation (black).

# Tables

**Table 1**. Comparison of CO emission ratio and modified combustion efficiency (MCE) from previous studies that sampled emissions from boreal forest fires. The studies are organized according to wildfire domain (North America or Siberia), management practice (wildfire or management fire), and sampling approach (aircraft, laboratory, or surface tower). Siberian studies are indicated as aircraft studies (A), surface based studies (S), or a combination of the two (A & S). The CO emission ratio column has units of ppmv ppmv$^{-1}$ and uses $CO_2$ as the reference gas. MCE was calculated as $1/(1 + \Delta CO/\Delta CO_2)$ when not directly reported in the study. The weighted mean of emission ratios and MCE for all previous studies is shown in the row labeled fire-weighted mean, with each study weighted according to the number of fires sampled.

| Study | $\Delta CO/\Delta CO_2$ Emission Ratio | Modified Combustion Efficiency | Number of fires sampled |
|---|---|---|---|
| *North American wildfires sampled by aircraft* | | | |
| Cofer et al., 1989 | 0.069 ± 0.004 | 0.935 ± 0.004 | 1 |
| Cofer et al., 1998 | 0.140 ± 0.012 | 0.878 ± 0.009 | 1 |
| Friedli et al., 2003 | 0.100 ± 0.020 | 0.909 ± 0.017 | 1 |
| Goode et al., 2000 | 0.085 ± 0.008 | 0.922 ± 0.007 | 4 |
| Laursen et al., 1992 | 0.050 ± 0.007 | 0.953 ± 0.006 | 1 |
| Nance et al., 1993 | 0.078 ± 0.012 | 0.928 ± 0.011 | 1 |
| O'Shea et al., 2013 | 0.150 ± 0.024 | 0.871 ± 0.012 | 4 |
| Radke et al., 1991 | 0.116 ± 0.087 | 0.896 ± 0.075 | 1 |
| Simpson et al., 2011 | 0.110 ± 0.070 | 0.901 ± 0.061 | 5 |
| **Fire-weighted mean** | **0.102 ± 0.033** | **0.908 ± 0.027** | **19** |
| *North American management fires sampled by aircraft* | | | |
| Cofer et al., 1990 | 0.086 ± 0.008 | 0.921 ± 0.007 | 2 |
| Cofer et al., 1998 | 0.095 ± 0.016 | 0.913 ± 0.013 | 7 |
| Radke et al., 1991 | 0.047 ± 0.032 | 0.956 ± 0.030 | 4 |
| Susott et al., 1991 | 0.060 ± 0.061 | 0.943 ± 0.058 | 1 |
| **Fire-weighted mean** | **0.077 ± 0.022** | **0.929 ± 0.020** | **14** |
| *North American fuels sampled in the laboratory* | | | |
| Yokelson et al., 1997[a] | 0.208 ± 0.039 | 0.827 ± 0.083 | - |
| Yokelson et al., 1997[b] | 0.231 ± 0.068 | 0.813 ± 0.167 | - |
| Yokelson et al., 1997[c] | 0.162 | 0.860 | - |
| Bertschi et al., 2003[d] | 0.151 ± 0.040 | 0.870 ± 0.030 | - |
| Burling et al., 2010[e] | 0.209 | 0.827 | - |
| Mc**Meeking** et al., 2009[e] | 0.153 ± 0.032 | 0.867 ± 0.074 | - |
| McMeeking et al., 2009[f] | 0.045 ± 0.005 | 0.957 ± 0.012 | - |
| McMeeking et al., 2009[c] | 0.030 | 0.971 | - |
| **Stockwell et al., 2014**[f] | 0.043 ± 0.004 | 0.959 ± 0.008 | - |
| **Stockwell et al., 2014**[g] | 0.245 ± 0.005 | 0.803 ± 0.009 | - |
| **Mean** | **0.143 ± 0.028** | **0.875 ± 0.053** | |
| *Siberian wildfires – sampled by aircraft or surface tower* | | | |
| Cofer et al., 1998 (A) | 0.224 ± 0.036 | 0.817 ± 0.025 | 1 |
| McRae et al., 2006 (A & S) | 0.249 ± 0.064 | 0.800 ± 0.043 | 6 |
| Vasileva et al., 2017 (S) | 0.126 ± 0.007 | 0.888 ± 0.005 | 2 |
| **Fire-weighted mean** | **0.219 ± 0.048** | **0.822 ± 0.033** | **9** |
| *North American wildfires sampled by surface tower* | | | |
| Wiggins et al., 2016 | 0.128 ± 0.023 | 0.887 ± 0.018 | 3 |
| This study | 0.142 ± 0.051 | 0.878 ± 0.039 | 34 |
| **Fire-weighted mean** | **0.141 ± 0.049** | **0.879 ± 0.027** | **37** |

1   <sup>a</sup> Moss (Alaska), <sup>b</sup> Peat (Alaska), <sup>c</sup> White Spruce (Alaska), <sup>d</sup> Duff Jack Pine/Black Spruce (Canada), <sup>e</sup> Duff Black Spruce (Alaska),

2   <sup>f</sup> Black Spruce (Alaska), and <sup>g</sup> Peat (Canada).

1 **Table 2.** Intervals with elevated trace gas mole fractions at CRV associated with fire emissions. Columns show the number of 30
2 s measurements used to calculate emission factors for each interval (N), the time of the interval (units of day of year (DOY)),
3 emission ratios (ppmv ppmv$^{-1}$), emission factor (g kg$^{-1}$ dry biomass burned), and modified combustion efficiency (MCE). The
4 primary combustion process is denoted as flaming, mixed, or smoldering using thresholds on MCE defined in the text.

| Interval number | N | Time of Event (DOY) | CO Emission Ratio | CO Emission Factor | CH$_4$ Emission Ratio | CH$_4$ Emission Factor | MCE | Combustion Phase |
|---|---|---|---|---|---|---|---|---|
| 1 | 82 | 173.27 - 173.30 | 0.161 ± 0.004 | 144 ± 4 | 0.012 ± 0.0003 | 6.1 ± 0.2 | 0.861 ± 0.004 | Mixed |
| 2 | 95 | 173.32 - 173.35 | 0.151 ± 0.004 | 136 ± 4 | 0.011 ± 0.0002 | 5.8 ± 0.2 | 0.869 ± 0.004 | Mixed |
| 3 | 95 | 173.36 - 173.39 | 0.141 ± 0.003 | 128 ± 3 | 0.010 ± 0.0002 | 5.5 ± 0.1 | 0.877 ± 0.003 | Mixed |
| 4 | 83 | 173.40 - 173.43 | 0.149 ± 0.008 | 135 ± 8 | 0.011 ± 0.0005 | 5.5 ± 0.3 | 0.870 ± 0.008 | Mixed |
| 5 | 95 | 173.45 - 173.48 | 0.130 ± 0.006 | 120 ± 6 | 0.009 ± 0.0004 | 5.0 ± 0.3 | 0.885 ± 0.006 | Mixed |
| 6 | 95 | 173.84 - 173.87 | 0.136 ± 0.008 | 124 ± 8 | 0.014 ± 0.0009 | 7.3 ± 0.5 | 0.880 ± 0.008 | Mixed |
| 7 | 85 | 174.27 - 174.30 | 0.170 ± 0.008 | 152 ± 8 | 0.008 ± 0.0003 | 4.3 ± 0.2 | 0.855 ± 0.008 | Mixed |
| 8 | 95 | 175.15 - 175.18 | 0.08 ± <0.001 | 78 ± 0.3 | 0.004 ± <1e4 | 2.3 ± <0.1 | 0.926 ± 1e4 | Flaming |
| 9 | 95 | 175.19 - 175.22 | 0.143 ± 0.007 | 131 ± 7 | 0.008 ± 0.0004 | 4.2 ± 0.3 | 0.875 ± 0.007 | Mixed |
| 10 | 58 | 175.23 - 175.25 | 0.091 ± 0.002 | 87 ± 2 | 0.005 ± 0.0002 | 2.5 ± 0.1 | 0.916 ± 0.002 | Mixed |
| 11 | 88 | 175.27 - 175.30 | 0.091 ± 0.001 | 87 ± 1 | 0.005 ± 0.0001 | 2.9 ± <0.1 | 0.917 ± 0.001 | Mixed |
| 12 | 95 | 175.32 - 175.35 | 0.153 ± 0.003 | 138 ± 4 | 0.009 ± 0.0002 | 4.5 ± 0.1 | 0.867 ± 0.003 | Mixed |
| 13 | 89 | 175.40 - 175.44 | 0.187 ± 0.012 | 164 ± 12 | 0.013 ± 0.0008 | 6.4 ± 0.5 | 0.842 ± 0.012 | Smoldering |
| 14 | 95 | 175.66 - 175.70 | 0.060 ± 0.003 | 59 ± 3 | 0.005 ± 0.0002 | 2.6 ± 0.1 | 0.943 ± 0.003 | Flaming |
| 15 | 55 | 175.75 - 175.77 | 0.129 ± 0.001 | 119 ± 1 | 0.009 ± 0.0001 | 4.5 ± 0.1 | 0.886 ± 0.001 | Mixed |
| 16 | 35 | 175.77 - 175.79 | 0.237 ± 0.015 | 198 ± 15 | 0.017 ± 0.0010 | 8.1 ± 0.6 | 0.809 ± 0.014 | Smoldering |
| 17 | 95 | 175.80 - 175.83 | 0.147 ± 0.002 | 133 ± 2 | 0.011 ± 0.0001 | 5.5 ± 0.1 | 0.872 ± 0.002 | Mixed |
| 18 | 95 | 175.88 - 175.91 | 0.155 ± 0.003 | 139 ± 3 | 0.009 ± 0.0002 | 4.9 ± 0.2 | 0.866 ± 0.003 | Mixed |
| 19 | 95 | 175.92 - 175.96 | 0.198 ± 0.004 | 172 ± 4 | 0.012 ± 0.0001 | 6.1 ± 0.1 | 0.835 ± 0.004 | Smoldering |
| 20 | 80 | 175.98 - 176.00 | 0.193 ± 0.003 | 169 ± 3 | 0.011 ± 0.0001 | 5.4 ± 0.1 | 0.838 ± 0.003 | Smoldering |
| 21 | 95 | 176.06 - 176.09 | 0.119 ± 0.007 | 111 ± 7 | 0.008 ± 0.0004 | 4.4 ± 0.3 | 0.893 ± 0.007 | Mixed |
| 22 | 85 | 177.06 - 177.09 | 0.108 ± 0.001 | 102 ± 1 | 0.010 ± 0.0001 | 5.3 ± <0.1 | 0.902 ± 0.001 | Mixed |
| 23 | 75 | 177.11 - 177.14 | 0.122 ± 0.002 | 113 ± 2 | 0.011 ± 0.0001 | 5.6 ± 0.1 | 0.892 ± 0.002 | Mixed |
| 24 | 95 | 177.15 - 177.18 | 0.129 ± 0.001 | 119 ± 1 | 0.010 ± 0.0001 | 5.5 ± 0.1 | 0.886 ± 0.001 | Mixed |
| 25 | 95 | 177.19 - 177.22 | 0.102 ± 0.002 | 96 ± 2 | 0.008 ± 0.0002 | 4.4 ± 0.1 | 0.908 ± 0.002 | Mixed |
| 26 | 58 | 177.23 - 177.25 | 0.148 ± 0.011 | 134 ± 12 | 0.012 ± 0.0009 | 6.0 ± 0.5 | 0.871 ± 0.011 | Mixed |
| 27 | 94 | 177.27 - 177.31 | 0.060 ± 0.002 | 59 ± 2 | 0.004 ± 0.0001 | 2.3 ± 0.1 | 0.944 ± 0.002 | Flaming |
| 28 | 95 | 177.80 - 177.83 | 0.094 ± 0.002 | 89 ± 2 | 0.008 ± 0.0001 | 4.1 ± 0.1 | 0.914 ± 0.002 | Mixed |
| 29 | 95 | 177.88 - 177.91 | 0.120 ± 0.006 | 111 ± 6 | 0.020 ± 0.0012 | 10.7 ± 0.7 | 0.893 ± 0.006 | Mixed |
| 30 | 93 | 177.92 - 177.96 | 0.164 ± 0.006 | 146 ± 7 | 0.018 ± 0.0007 | 8.9 ± 0.4 | 0.859 ± 0.006 | Mixed |
| 31 | 95 | 184.23 - 184.26 | 0.232 ± 0.014 | 196 ± 15 | 0.013 ± 0.0007 | 6.5 ± 0.4 | 0.811 ± 0.014 | Smoldering |
| 32 | 80 | 186.49 - 186.52 | 0.025 ± 0.002 | 25 ± 2 | 0.002 ± 0.0001 | 1.2 ± 0.1 | 0.976 ± 0.002 | Flaming |
| 33 | 64 | 188.07 - 188.09 | 0.188 ± 0.012 | 165 ± 13 | 0.013 ± 0.0008 | 6.6 ± 0.5 | 0.842 ± 0.012 | Smoldering |
| 34 | 95 | 188.10 - 188.13 | 0.106 ± 0.002 | 100 ± 2 | 0.008 ± 0.0002 | 4.5 ± 0.1 | 0.904 ± 0.002 | Mixed |
| 35 | 54 | 188.14 - 188.16 | 0.109 ± 0.001 | 102 ± 1 | 0.008 ± 0.0001 | 4.3 ± <0.1 | 0.902 ± 0.001 | Mixed |
| 36 | 64 | 188.20 - 188.22 | 0.104 ± 0.004 | 99 ± 4 | 0.008 ± 0.0003 | 4.2 ± 0.2 | 0.906 ± 0.004 | Mixed |
| 37 | 52 | 188.23 - 188.25 | 0.080 ± 0.007 | 77 ± 7 | 0.006 ± 0.0004 | 3.2 ± 0.2 | 0.926 ± 0.007 | Flaming |
| 38 | 95 | 188.40 - 188.44 | 0.194 ± 0.003 | 169 ± 3 | 0.012 ± 0.0002 | 6.1 ± 0.1 | 0.837 ± 0.003 | Smoldering |
| 39 | 95 | 188.45 - 188.48 | 0.131 ± 0.004 | 120 ± 4 | 0.013 ± 0.0006 | 6.9 ± 0.3 | 0.884 ± 0.004 | Mixed |
| 40 | 36 | 188.53 - 188.55 | 0.146 ± 0.002 | 132 ± 2 | 0.012 ± 0.0001 | 6.0 ± 0.1 | 0.873 ± 0.002 | Mixed |
| 41 | 54 | 188.59 - 188.61 | 0.163 ± 0.002 | 145 ± 2 | 0.012 ± 0.0001 | 6.3 ± 0.1 | 0.860 ± 0.002 | Mixed |
| 42 | 95 | 188.62 - 188.65 | 0.179 ± 0.002 | 158 ± 2 | 0.014 ± 0.0002 | 6.9 ± 0.1 | 0.848 ± 0.002 | Smoldering |
| 43 | 74 | 188.66 - 188.69 | 0.214 ± 0.011 | 183 ± 12 | 0.015 ± 0.0008 | 7.4 ± 0.5 | 0.824 ± 0.011 | Smoldering |
| 44 | 95 | 188.71 - 188.74 | 0.138 ± 0.005 | 126 ± 5 | 0.010 ± 0.0004 | 5.1 ± 0.2 | 0.879 ± 0.005 | Mixed |
| 45 | 95 | 188.75 - 188.78 | 0.055 ± 0.003 | 54 ± 3 | 0.006 ± 0.0002 | 3.3 ± 0.1 | 0.948 ± 0.003 | Flaming |
| 46 | 95 | 188.79 - 188.83 | 0.272 ± 0.009 | 223 ± 10 | 0.012 ± 0.0005 | 5.7 ± 0.3 | 0.786 ± 0.009 | Smoldering |
| 47 | 52 | 188.84 - 188.85 | 0.120 ± 0.002 | 112 ± 2 | 0.009 ± 0.0001 | 4.9 ± 0.1 | 0.893 ± 0.002 | Mixed |
| 48 | 39 | 188.86 - 188.87 | 0.091 ± 0.002 | 87 ± 2 | 0.007 ± 0.0001 | 4.0 ± 0.1 | 0.916 ± 0.002 | Mixed |
| 49 | 59 | 189.03 - 189.05 | 0.154 ± 0.012 | 139 ± 13 | 0.010 ± 0.0008 | 5.3 ± 0.5 | 0.867 ± 0.012 | Mixed |
| 50 | 95 | 189.27 - 189.31 | 0.149 ± 0.008 | 135 ± 9 | 0.011 ± 0.0005 | 5.6 ± 0.3 | 0.871 ± 0.008 | Mixed |
| 51 | 30 | 189.34 - 189.35 | 0.090 ± 0.009 | 86 ± 9 | 0.006 ± 0.0005 | 3.2 ± 0.3 | 0.917 ± 0.009 | Mixed |
| 52 | 89 | 189.49 - 189.52 | 0.165 ± 0.009 | 147 ± 9 | 0.012 ± 0.0007 | 6.1 ± 0.4 | 0.858 ± 0.009 | Mixed |
| 53 | 48 | 195.10 - 195.12 | 0.212 ± 0.019 | 181 ± 20 | 0.016 ± 0.0014 | 8.0 ± 0.9 | 0.825 ± 0.018 | Smoldering |
| 54 | 37 | 195.12 - 195.13 | 0.262 ± 0.027 | 215 ± 28 | 0.020 ± 0.0020 | 9.5 ± 1.2 | 0.792 ± 0.026 | Smoldering |
| 55 | 95 | 195.14 - 195.17 | 0.140 ± 0.007 | 128 ± 8 | 0.010 ± 0.0006 | 5.5 ± 0.3 | 0.877 ± 0.007 | Mixed |
| Mean | | | 0.142 ± 0.051 | 127 ± 40 | 0.010 ± 0.0038 | 5.3 ± 1.8 | 0.878 ± 0.039 | |

**Table 3.** All fires that contributed to at least 1% of the total CO anomaly observed at CRV, in order from largest CO contribution to smallest CO contribution. The distance column represents the distance of the center of the fire perimeter to CRV tower. Contribution is the percent contribution to the total integral of fire CO at CRV for the entire 2015 fire season. Some fires were grouped together if they were inside the same 0.5° grid cell during model coupling. For those cases, individual fire contribution to the CO anomaly observed at CRV tower was weighted based on fire size.

| | Fire Name | Distance (km) | Contribution (%) | Total Hectares | Fuel Type | Ignition Source |
|---|---|---|---|---|---|---|
| 1 | Tozitna | 229 | 10.74 | 31652 | Black Spruce | Lightning |
| 2 | Kobe | 119 | 7.20 | 3444 | Black Spruce | Lightning |
| 3 | Blair | 82 | 6.31 | 15217 | Black Spruce | Lightning |
| 4 | Aggie Creek | 41 | 5.63 | 12829 | Black Spruce | Lightning |
| 5 | Spicer Creek | 195 | 5.30 | 39761 | Black Spruce | Lightning |
| 6 | Blind River | 252 | 3.87 | 24608 | Black Spruce | Lightning |
| 7 | Holtnakatna | 404 | 3.44 | 90308 | Mixed | Lightning |
| 8 | Blazo | 514 | 3.39 | 49106 | Black Spruce | Lightning |
| 9 | Big Creek 2 | 351 | 3.23 | 126637 | Black Spruce | Lightning |
| 10 | Chitanana River | 241 | 3.12 | 17483 | Black Spruce | Lightning |
| 11 | Sea | 309 | 3.06 | 172 | Black Spruce | Human |
| 12 | Sushgitit Hills | 276 | 2.92 | 111712 | Black Spruce | Lightning |
| 13 | Big Mud River 1 | 254 | 2.72 | 42076 | Black Spruce | Lightning |
| 14 | Lost River | 347 | 2.58 | 21088 | Black Spruce | Lightning |
| 15 | Munsatli 2 | 302 | 2.36 | 40682 | Black Spruce | Lightning |
| 16 | FWA Small Arms Complex | 19 | 2.31 | 740 | Black Spruce | Prescribed |
| 17 | Tobatokh | 280 | 2.24 | 21868 | Black Spruce | Lightning |
| 18 | Trail Creek | 363 | 2.24 | 11939 | Black Spruce | Lightning |
| 19 | Lloyd | 201 | 2.22 | 26818 | Black Spruce | Lightning |
| 20 | Isahultila | 342 | 2.17 | 60445 | Black Spruce | Lightning |
| 21 | Nulato | 499 | 2.17 | 449 | Black Spruce | Lightning |
| 22 | Three Day | 472 | 2.17 | 39378 | Black Spruce | Lightning |
| 23 | Hay Slough | 188 | 1.90 | 37007 | Black Spruce | Lightning |
| 24 | Rock | 316 | 1.83 | 3714 | Other | Lightning |
| 25 | Sulukna | 329 | 1.77 | 6760 | Black Spruce | Lightning |
| 26 | Titna | 273 | 1.77 | 12415 | Black Spruce | Lightning |
| 27 | Quinn Creek | 657 | 1.49 | 2002 | Other | Lightning |
| 28 | Harper Bend | 188 | 1.45 | 17555 | Black Spruce | Lightning |
| 29 | Hard Luck | 328 | 1.43 | 5230 | Black Spruce | Lightning |
| 30 | Fox Creek | 369 | 1.42 | 2346 | Black Spruce | Lightning |
| 31 | Bering Creek | 280 | 1.36 | 45654 | Black Spruce | Lightning |
| 32 | Eden Creek | 324 | 1.16 | 18614 | Black Spruce | Lightning |
| 33 | Falco | 390 | 1.10 | 1817 | Mixed | Lightning |
| 34 | Jackson | 202 | 1.00 | 2969 | Black Spruce | Lightning |