# Peer review of "Boreal forest fire CO and CH4 emission factors derived from tower 2 observations in Alaska during the extreme fire season of 2015 3"

_Atmospheric Chemistry and Physics, 2019_

## Referee Comment (RC1) · Robert Yokelson (Referee) · 6 Feb 2020

Review of Wiggins et al Boreal Fire EFs

By Bob Yokelson

General:

This manuscript reports much needed, very important, high-quality boreal forest fire smoke measurements with impressive modeling support and the work should very much be published. Unfortunately, there seems to be an error in the calculation of emission factors (EFs) explained in detail below. If so, that will require revisions to reported values, re-interpretation of the implications, and re-review. As explained below, the data may in fact support earlier EFs rather than suggest they should be higher. I am submitting a quick, rough review so the authors can correct this if needed or validate their calculation if appropriate. I'm happy to communicate directly with the authors about the calculations and to review the paper in more detail after the calculations can be verified to be correct, and, if needed, the analysis and conclusions are appropriately modified.

A second, relatively minor, general comment is that there is some missing context that could be added to the intro or discussion that could help motivate why the authors data is so valuable and perhaps inform the interpretation. I'll summarize that next.

Bertschi et al., (2003) showed that adjusting EFs for rarely sampled residual smoldering combustion (RSC) led to important adjustments in the EFs for all fire types and especially for fires burning heavy or duff fuels. Christian et al., (2007), Burling et al., (2012), Akagi et al., (2013; 2014) and others all supplemented airborne measurements with ground-based measurements on the same fire to explore this, but the relative importance of weakly lofted smoldering and flaming emissions could only be crudely estimated from size-/type-resolved fuel consumption measurements, which are challenging and rare. Yates et al., (2015) showed that even airborne measurements can imply a much larger smoldering/flaming ratio late in long-lasting fires. Saide et al., (2015, and references therein) showed that rarely sampled nighttime combustion is both important and underestimated in some cases using commonly assumed diurnal cycles. So there is precedent and ample support in the literature for factoring in smoldering and nighttime combustion, but little data to judge the potential differences in emissions or the relative production. For this reason, Selimovic et al., (2019a, b) deployed ground-based smoke monitoring downwind of hundreds of fires burning at all stages for two fire seasons. A priori, one might suspect that ground-based sampling could be biased towards smoldering and airborne sampling to flaming, but these authors found that conserved tracers sensitive to flaming (BC) and smoldering (CO) had a similar ratio from both air and ground. This implies both platforms are relevant and maybe even in sufficient agreement for some purposes. Other findings from this work are relevant/comparable to the authors work as well. Even earlier, the widely used Akagi et al., (2011) recommendations for boreal forest fire EFs had been based on averaging ground and airborne measurements together as a "best guess" at overall EFs. Finally, It's very likely that ground-based downwind measurements are best for validating AQ models, but it may be that satellite or aircraft vertical profiles will be needed to best probe overall emissions. Climate assessments may be more interested in smoke in higher layers, which may be missed by towers? However, this work is an extensive and welcome addition to the information available.

Next some details on why it is unclear if the authors got "much higher EFCO" and whether their work actually implies more smoldering than previously assumed since MCEs are directly measured and similar to some widely used previous work.

To start, I compare the authors EFs at face value to those from some widely-used recommendations: namely Andreae and Merlet 2001, now updated (Andreae, 2019) and Akagi et al., (2011). Akagi et al recommended a 50/50 average of the ground-based and airborne EFs in their boreal recommendations. For boreal the 50/50 ground/air led to EFCO of 127(45) g/kg compared to the authors 145(46) g/kg in their Table 1. So if there EF is correct it is 14% higher. They are closer to the A11 ground-based average of 157. Andreae (2019) recommend the straight average of 20+ studies, which is 121.4(46.6). Putting the EFs and MCEs together in a table reveals some things.

|         | EFCO       | MCE          | EFCH4      | n                                |
|---------|------------|--------------|------------|----------------------------------|
| A11     | 127(45)    | .881         | 5.96(3.14) | 7 studies multiple fires per study |
| A19     | 121.4(46.6)| .89          | 5.5(2.5)   | 20+ studies                      |
| Wiggins | 145(46)    | 0.879(0.068) | 6.05(2.09) | 35 fires                         |

While the authors current calculated EFCO is about 15% larger, the directly-measured MCEs are very close so maybe this new data does not imply more smoldering? Also this works MCE of 0.879(0.068) is not far from Selimovic et al (2019) estimated MCE (based on BC/CO) for similar long-lasting fires in heavy fuels of 0.87(0.02).

The similarity in MCE along with non-standard notation in eqn 2 and a lack of definition for the authors "S" scalar inspired me to calculate EF directly from their emission ratios (ERs). Using the authors quoted assumption of 45% C for the fuel; I get different EF values than them: EFCO2 1437, EFCO 126.2, EFCH4 5.23. These EF values are the same or lower. If my calculation is right, then this new work supports the previous work rather than suggesting the values in use should be increased. It's still good data even if it agrees with previous work. Also %C > 45% is possible for boreal fires. 50% C is often assumed though one study (Santin et al., 2015) did measure fuel C close to 45% for a fire in boreal forest. But using the authors average ERs I have to assume 519 gC/kg to get close to their EF for CH4 and CO; and ~52%C seems to high. I'd be happy to share my calculation (Yokelson et al., 1999) and re-review a revised paper if necessary.

Another possible reason for an ER-EF mismatch is using different averaging schemes for these two quantities? Ideally the averaging scheme should be the same for both quantities. If possible, it might be good to weight for how much smoke was produced at the fire, received at the tower, duration of events, or etc. Exploring how the average depends on the scheme employed is always useful and could be reported along with a clear explanation of how the averaging was done for the reported values.

A few other things I noticed in order of Page, Line. This is a one-skim set of potentially useful comments. A more careful review could be done after ensuring the calculations are accurate.

P1, L14: define CRV

P1, L33 – P2, L2: will these aggressively lofted emissions impact tower? Run some forward/back trajectories? Vertical mixing?

P2, L6: "deadly" AQ is over-simplified

2, 13: Andreae and Merlet was updated in 2019

2, 18: The updated Andreae paper lists more than 20 studies, so there may be more worth including in Table 1.

2, 28: Ground-based data downwind of fires has also been collected in Selimovic et al (2019a, b) and e.g. in the Colorado front range (Gilman, Benedict) and MBO (Collier and references there-in). The Wiggins 2016 data doesn't appear in this paper anywhere that I saw.

2, 33-34: Akagi et al., 2011 explain how MCE can be used to estimate an arbitrary mix of smoldering and flaming over a continuous range.

2, 37: Real fires often don't have phases - rather a dynamic mix of processes. Change "phase" to "process" throughout?

3,22: The tower results, even if lowered are higher than "some" airborne studies. E.g. Cofer 98 is the same. If the EFs stay the same they are "a bit higher" than "some" previous estimates or recommendations.

4, 6: Confusing, is it just 50 minute samples with ten minutes downtime per hour?

4, 30: EFs are usually given for one species so the meaning of the ratio in the subscript here and in eqn 2 is not apparent. Suggest adopting standard notation?

4, 32: Do these references unambiguously support 45% C? Revisit, consider reference above, and explain in detail in revised text.

Eqn (1) Sum or slope or simple subtraction?

Eqn (2) Again, notation unusual, something common should work, or explain?

5, 1-5: I get computing MCE for each sample, but what's the point of the categories that don't seem to be used?

Sec 2.3: There is not much detail on how AKFED is driven. One thing that stands out though is that the day/night split for fuel consumption is likely not right for 64 N! See Vermote et al, (2009); MODIS FRP can be higher at "night" than during the "day" in high latitude summer. This is relevant later.

6, 1-9: It's likely that some weaker smoke peaks are more distorted by background variability and increase the range of values, but there is not necessarily bias.

6, 9: Many references support high correlation of EFCH4 with MCE.

6, 38-39: This needs to be thought through a bit. Does the high impact of night smoke at the tower compared to the assumed low fraction of smoke produced at night mean day smoke was under-sampled? Or does this imply AKFED underestimates night smoke?

7, 3: "emissions" to "consumption"

7, 8: 15 total previous fires sampled may be too low if you check updated compilations.

7, 9: Convection entrains some smoldering.

7, 10-17: True, but a tower could potentially undersample flaming. Flaming is associated with rapid fuel consumption so not a negligible concern. Try forward trajectories from high injection altitudes to see if they impact tower or compare to column data?

7, 19: Night may have been oversampled? But maybe not if there really is more emissions at night than was assumed in AKFED? As noted above, there is sometimes more MODIS FRP at night than day in boreal regions. Also, as above, towers may not be sensitive to the entire range of injection altitudes? Explore?

7, 20-21: Quote these values from 2016 paper in Table 1?

Table 1 header or caption: It's enormous! Move part elsewhere. The text mentions CH4 data which I did not see in table.

7, 27: Cofer 98 agrees with this studies current values and the real average may be in middle of all this data somewhere.

7, 28-31: This data should certainly be used, but rarely does new data replace old data completely. More often new data contributes to an evolving literature average – sometimes with weighting by n factor.

7, 36: You can often see smoke by satellite even when you can't detect FRP.

8, 3-4: To claim a difference with the studies above you would have to know proportion of above-/below-ground fuel consumption that goes with those studies.

8, 8: This is a common error to assume that increased EFs will lead to increased, modeled health impacts. Models use EF*biomass burned to get a-priori emissions. Then the modeled impacts are compared to downwind monitors and the a-priori emissions are adjusted to best match reality. A higher EF may change the details of the tweaking procedure, but not change the downwind PM. What would change the latter is discovering a problem with the PM monitors.

8, 10: "lead to"

8, 17: Towers are not completely new. There was a long history of sampling prescribed fires from towers carried out by the Fire Lab.

References:

Akagi, S. K., Yokelson, R. J., Burling, I. R., Meinardi, S., Simpson, I., Blake, D. R., McMeeking, G. R., Sullivan, A., Lee, T., Kreidenweis, S., Urbanski, S., Reardon, J., Griffith, D. W. T., Johnson, T. J., and Weise, D. R.: Measurements of reactive trace gases and variable O3 formation rates in some South Carolina biomass burning plumes, Atmos. Chem. Phys., 13, 1141-1165, doi:10.5194/acp-13-1141-2013, 2013.

Akagi, S. K., Burling, I. R., Mendoza, A., Johnson, T. J., Cameron, M., Griffith, D. W. T., Paton-Walsh, C., Weise, D. R., Reardon, J., and Yokelson, R. J.: Field measurements of trace gases emitted by prescribed fires in southeastern US pine forests using an open-path FTIR system, Atmos. Chem. Phys., 14, 199-215, doi:10.5194/acp-14-199-2014, 2014.

Andreae, M. O.: Emission of trace gases and aerosols from biomass burning – an updated assessment, Atmos. Chem. Phys., 19, 8523–8546, https://doi.org/10.5194/acp-19-8523-2019, 2019.

Benedict, K. B., Prenni, A. J., Carrico, C. M., Sullivan, A. P., Schichtel, B. A., and Collett Jr., J. L.: Enhanced concentrations of reactive nitrogen species in wildfire smoke, Atmos. Environ., 148, 8–15, 2017.

Burling, I. R., Yokelson, R. J., Akagi, S. K., Urbanski, S. P., Wold, C. E., Griffith, D. W. T., Johnson, T. J., Reardon, J., and Weise, D. R.: Airborne and ground-based measurements of the trace gases and particles emitted by prescribed fires in the United States, Atmos. Chem. Phys., 11, 12197-12216, doi:10.5194/acp-11-12197-2011, 2011.

Christian, T.J., R.J. Yokelson, J.A. Carvalho Jr., D.W.T. Griffith, E.C. Alvarado, J.C. Santos, T.G.S. Neto, C.A.G. Veras, and W.M. Hao, The tropical forest and fire emissions experiment: Trace gases emitted by smoldering logs and dung on deforestation and pasture fires in Brazil, J. Geophys. Res., 112, D18308, doi:10.1029/2006JD008147, 2007.

Collier, S., Zhou, S., Onasch, T., Jaffe, D., Kleinman, L., Sedlacek, A., Briggs, N., Hee, J., Fortner, E., Shilling, J., Worsnop, D., Yokelson, R., Parworth, C., Ge, X., Xu, J., Butterfield, Z., Chand, D., Dubey, M., Pekour, M., Springston, S., and Zhang, Q.: Regional influence of aerosol emissions from wildfires driven by combustion efficiency: Insights from the BBOP campaign, Environ. Sci. Technol., 50, 8613-8622, doi:10.1021/acs.est6b01617, 2016.

Gilman, J. B., Lerner, B. M., Kuster, W. C., Goldan, P. D., Warneke, C., Veres, P. R., Roberts, J. M., de Gouw, J. A., Burling, I. R., and Yokelson, R. J.: Biomass burning emissions and potential air quality impacts of volatile organic compounds and other trace gases from fuels common in the US, Atmos. Chem. Phys., 15, 13915-13938, doi:10.5194/acp-15-13915-2015, 2015.

Saide, P. E., Peterson, D., da Silva, A., Anderson, B., Ziemba, L. D., Diskin, G., Sachse, G., Hair, J., Butler, C., Fenn, M., Jimenez, J. L., Campuzano-Jost, O., Perring, A., Schwarz, J., Markovic, M. Z., Russell, P., Redemann, J., Shinozuka, Y., Streets, D. G., Yan, F., Dibb, J., Yokelson, R., Toon, O. B., Hyer, E., and Carmichael, G. R.: Revealing important nocturnal and day-to-day variations in fire smoke emissions through a multiplatform inversion, Geophys. Res. Lett., 42, 3609-3618, doi:10.1002/2015GL063737, 2015.

Santín, C., S. H. Doerr, C. M. Preston, and G. González-Rodríguez (2015), Pyrogenic organic matter production from wildfires: a missing sink in the global carbon cycle, Glob. Change Biol., 21(4), 1621-1633, doi: 10.1111/gcb.12800.

Selimovic, V., Yokelson, R. J., McMeeking, G. R., and Coefield, S.: In situ measurements of trace gases, PM, and aerosol optical properties during the 2017 NW US wildfire smoke event, Atmos. Chem. Phys., 19, 3905-3926, https://doi.org/10.5194/acp-19-3905-2019, 2019.

Selimovic, V., Yokelson, R., McMeeking, G. R., and Coefield, S. Aerosol mass and optical properties, smoke influence on O3, and high NO3 production rates in a western US city impacted by wildfires, submitted, J. Geophys. Res., MS No. 2019JD032324, 2019.
https://www.essoar.org/doi/abs/10.1002/essoar.10501529.1

Vermote, E., E. Ellicott, O. Dubovik, T. Lapyonok, M. Chin, L. Giglio, and G. J. Roberts (2009), An approach to estimate global biomass burning emissions of organic and black carbon from MODIS fire radiative power, J. Geophys. Res., 114, D18205, doi:10.1029/2008JD011188.

Yates, E. L., Iraci, L. T., Singh, H. B., Tanaka, T., Roby, M. C., Hamill, P., Clements, C. B., Lareau, N., Contezac, J., Blake, D. R., Simpson, I. J., Wisthaler, A., Mikoviny, T., Diskin, G. S., Beyersdorf, A. J., Choi, Y., Ryerson, T. B., Jimenez, J. L., and Gore, W.: Airborne measurements and emissions estimates of greenhouse gases and other trace constituents from the 2013 California Yosemite Rim wildfire, Atmos. Environ., 127, 293–302, https://doi.org/10.1016/j.atmosenv.2015.12.038, 2016.

Yokelson, R.J., J.G. Goode, D.E. Ward, R.A. Susott, R.E. Babbitt, D.D. Wade, I. Bertschi, D.W. T. Griffith, and W.M. Hao, Emissions of formaldehyde, acetic acid, methanol and other trace gases from biomass fires in North Carolina measured by airborne Fourier transform infrared spectroscopy (AFTIR), J. Geophys. Res., 104, 30109-30125, 1999.

---

## Referee Comment (RC2) · Anonymous Referee #2 · 12 Feb 2020

Review criteria: Scientific significance: Excellent (4) Scientific quality: Good/Excellent (3.5) Presentation quality (2)

General comments: The discussion paper presents important research into the characteristics of wildfire emissions using established techniques, but novel analysis. Teasing apart the contributions of various fire events and the combustion stage (Flaming vs. smoldering) is a new and valuable way to understand nuances of boreal fire relevant to many needs, such as human health, carbon cycling, and smoke planning. However, the paper falls short in many ways, and will need some extensive modification to reach its potential. I strongly suggest a re-focus on a more relevant outcome from

the work (rather than the fact that previous work was not catching smoldering as well as they could), a fully revised Discussion (some ideas below), and some attention to references (see notes below). This work is very important, and when presented well will make a great contribution to the literature on this subject. Specific comments:

1. The title will need modification. It is unclear what "larger" refers to – larger than what? Than previous studies (yes, but I know that only when I get to the end of the Abstract). It could be larger than flaming combustion. The point is that having an unreferenced comparative adjective can be troublesome, especially in a title where you want to be clear. The title could be the same, but with the first four words dropped: "Contribution of . . .". Also, it is my opinion that, while this may show larger contribution than previous studies, this work has a lot of other implications and contribution that could be highlighted in the title. In some ways the community would not be too surprised to learn that the smoldering fire signal has not been captured in previous studies, so highlighting this part of it is not needed to make this an impactful paper/study.

2. The comparison to previous studies would more naturally go into the discussions, rather than the introduction/background. I suggest revising to put Table 1 into the discussion where you can make the case more directly, rather than introducing the previous work without yet seeing your results.

3. There is a blatant and concerning misuse of terminology on Page 2, line 34: The sentence "Smoldering combustion can be defined as combustion with a degree of combustion completeness, or modified combustion efficiency, less than 0.9 [Urbanski 2014]." First, MCE and combustion completeness (CC) are very different things. CC is the proportion of fuels consumed/combusted, while MCE is defined as the proportion of a gas to $CO_2$. Second, the Urbanski paper puts MCE of 0.65 to 0.85 as "smoldering", and references Akagi et al. 2011 so I don't know where the 0.9 figure comes from. The choice of the thresholds stated on page 5 lines 1-4 need to be better justified.

4. I found a couple of instances where the citations used are inappropriate. While I mention only 2 here, I would suspect others, so the citations need to be fully vetted for appropriateness. First: "Rogers et al. 2015" in Page 1 line 33 is not a review of boreal fire regime. It may mention this, but it is not what that study provides to the literature. Second: "Bertschi et al. 2003" in Page 7 line 34 is of laboratory experiments and work in savannah ecosystems, not boreal forest fires. In both of these cases, it could be argued that no reference is needed. If you do include a reference, it needs to be a paper or resource where the statement made is shown or studied, not where it was stated. I suggest the co-authors assist with improving the citations.

5. The discussion would benefit from more regarding the implications of the results. What is the data showing us that is relevant? Some possible ideas to highlight/discuss (these need to be discussed with co-authors, so are only representative):

a. Figure 5 (Page 6 line 9) shows a linear relationship between CH4 and MCE. Provide a short discussion of this in the discussion – what does this mean for using the data?

b. Page 6 line 22 – "…attributed to boreal fire emissions." – As opposed to what?? Or why? A bit of discussion on what other factors contribute to the signal, and why there are some difference in the model will help non-atmospheric modelers better understand why these results are so powerful

c. The temporal distribution data (Fig 10) is very interesting and could be helpful for exposure assessment for health studies. (although PM, rather than CO would be of interest).

d. Page 7, line 24: I am not sure I see a temporal trend in the old data, and I am not sure why this would be something to note. This statement is best dropped. Table 1 presents past results that are collected in a variety of settings, so (in my assessment) represents some data on the range of variability, not a record of change over time.

I hope these comments inspire the authors to revise the manuscript for a more useful

product.
* * *

---

## Referee Comment (RC3) · Anonymous Referee #3 · 6 Mar 2020

The authors present an impressive set of CO, CH4 and CO2 measurements in boreal forest fire smoke to calculate emission factors for CO and CH4. Such accurate data are neccesarry to soundly test models used to quantify the impact of big fires on the air quality and climate. Therefore, the paper is highly suitable to be published in this journal.

Yet, due to error in the emission factors calculation I suggest a revision of this manuscript. Re-interpretation of results should be done by the authors before resubmission.

Firstly, Eq 2 on Page 4 should be revised, using e.g. Eq. 1 and 2 in Yokelson et al.

JGR1999.

I'm looking forward to review the revised manuscript in detail. I add here only a potential useful comment. Are there any other measurements of specific tracers to be used to quantify the smoldering/flaming contributions? Flaming is likely under-represented in the used sampling height.

---

## Author Response (AR1)

**Reviewer #1 (Bob Yokelson) Comments and Responses**

**General Comments:**

This manuscript reports much needed, very important, high-quality boreal forest fire smoke measurements with impressive modeling support and the work should very much be published. Unfortunately, there seems to be an error in the calculation of emission factors (EFs) explained in detail below. If so, that will require revisions to reported values, re-interpretation of the implications, and re-review. As explained below, the data may in fact support earlier EFs rather than suggest they should be higher. I am submitting a quick, rough review so the authors can correct this if needed or validate their calculation if appropriate. I'm happy to communicate directly with the authors about the calculations and to review the paper in more detail after the calculations can be verified to be correct, and, if needed, the analysis and conclusions are appropriately modified.

A second, relatively minor, general comment is that there is some missing context that could be added to the intro or discussion that could help motivate why the authors data is so valuable and perhaps inform the interpretation. I'll summarize that next.

Bertschi et al., (2003) showed that adjusting EFs for rarely sampled residual smoldering combustion (RSC) led to important adjustments in the EFs for all fire types and especially for fires burning heavy or duff fuels. Christian et al., (2007), Burling et al., (2012), Akagi et al., (2013; 2014) and others all supplemented airborne measurements with ground-based measurements on the same fire to explore this, but the relative importance of weakly lofted smoldering and flaming emissions could only be crudely estimated from size-/type-resolved fuel consumption measurements, which are challenging and rare. Yates et al., (2015) showed that even airborne measurements can imply a much larger smoldering/flaming ratio late in long-lasting fires. Saide et al., (2015, and references therein) showed that rarely sampled nighttime combustion is both important and underestimated in some cases using commonly assumed diurnal cycles. So there is precedent and ample support in the literature for factoring in smoldering and nighttime combustion, but little data to judge the potential differences in emissions or the relative production. For this reason, Selimovic et al., (2019a, b) deployed ground-based smoke monitoring downwind of hundreds of fires burning at all stages for two fire seasons. A priori, one might suspect that ground-based sampling could be biased towards smoldering and airborne sampling to flaming, but these authors found that conserved tracers sensitive to flaming (BC) and smoldering (CO) had a similar ratio from both air and ground. This implies both platforms are relevant and maybe even in sufficient agreement for some purposes. Other findings from this work are relevant/comparable to the authors work as well. Even earlier, the widely used Akagi et al., (2011) recommendations for boreal forest fire EFs had been based on averaging ground and airborne measurements together as a "best guess" at overall EFs. Finally, It's very likely that ground-based downwind measurements are best for validating AQ models, but it may be that satellite or aircraft vertical profiles will be needed to best probe overall emissions. Climate assessments may be more interested in smoke in higher layers, which may be missed by towers? However, this work is an extensive and welcome addition to the information available.

Next some details on why it is unclear if the authors got "much higher EFCO" and whether their work actually implies more smoldering than previously assumed since MCEs are directly measured and similar to some widely used previous work.

To start, I compare the authors EFs at face value to those from some widely-used recommendations: namely Andreae and Merlet 2001, now updated (Andreae, 2019) and Akagi et al., (2011). Akagi et al recommended a 50/50 average of the ground-based and airborne EFs in their boreal recommendations. For boreal the 50/50 ground/air led to EFCO of 127(45) g/kg compared to the authors 145(46) g/kg in their Table 1. So if there

EF is correct it is 14% higher. They are closer to the A11 ground-based average of 157. Andreae (2019) recommend the straight average of 20+ studies, which is 121.4(46.6). Putting the EFs and MCEs together in a table reveals some things.

|  | EFCO | MCE | EFCH4 | n |
|---|---|---|---|---|
| A11 | 127(45) | .881 | 5.96(3.14) | 7 studies multiple fires per study |
| A19 | 121.4(46.6) | .89 | 5.5(2.5) | 20+ studies |
| Wiggins | 145(46) | 0.879(0.068) | 6.05(2.09) | 35 fires |

While the authors current calculated EFCO is about 15% larger, the directly-measured MCEs are very close so maybe this new data does not imply more smoldering? Also this works MCE of 0.879(0.068) is not far from Selimovic et al (2019) estimated MCE (based on BC/CO) for similar long-lasting fires in heavy fuels of 0.87(0.02).

The similarity in MCE along with non-standard notation in eqn 2 and a lack of definition for the authors "S" scalar inspired me to calculate EF directly from their emission ratios (ERs). Using the authors quoted assumption of 45% C for the fuel; I get different EF values than them: EFCO2 1437, EFCO 126.2, EFCH4 5.23. These EF values are the same or lower. If my calculation is right, then this new work supports the previous work rather than suggesting the values in use should be increased. It's still good data even if it agrees with previous work. Also %C > 45% is possible for boreal fires. 50% C is often assumed though one study (Santin et al., 2015) did measure fuel C close to 45% for a fire in boreal forest. But using the authors average ERs I have to assume 519 gC/kg to get close to their EF for CH4 and CO; and ~52%C seems to high. I'd be happy to share my calculation (Yokelson et al., 1999) and re-review a revised paper if necessary.

Another possible reason for an ER-EF mismatch is using different averaging schemes for these two quantities? Ideally the averaging scheme should be the same for both quantities. If possible, it might be good to weight for how much smoke was produced at the fire, received at the tower, duration of events, or etc. Exploring how the average depends on the scheme employed is always useful and could be reported along with a clear explanation of how the averaging was done for the reported values.

A few other things I noticed in order of Page, Line. This is a one-skim set of potentially useful comments. A more careful review could be done after ensuring the calculations are accurate.

**We appreciate Dr. Yokelson's positive comments that our manuscript provides needed and important CO and CH4 emission factor measurements from boreal forest fires.**

**We agree with his assessment that there was an important omission in our emission factor calculation. Finally, Dr. Yokelson provides a valuable perspective (and references) on past work that has compared ground-based and aircraft-based estimates of emission factors.**

**We propose the following major revisions to our paper to address these issues.**

**First, our emission factor calculations will be corrected. This is easy and relatively straightforward to implement. More importantly, we will reinterpret our results and their implications in our revised manuscript, taking into account the revised emission factor information. This will require revisions to the title and abstract as well as main text.**

Second, we will change equation 2 in our manuscript to standard notation as published in previous studies (Yokelson et al., 1999) and offer more clarity on the definition of the variables.

Third, will modify the introduction and discussion to include more context to motivate the importance of this study and inform the interpretation of our results. We plan to include the studies Dr. Yokelson highlighted in our revised manuscript. Specifically, we plan to integrate work by Bertschi et al. (2003), Christian et al. (2007), Burling et al. (2011), Akagi et al (2014), Santin et al. (2015), Yates et al. (2016), Andreae (2019), and Selimovic et al. (2019a,b), and Yokelson et al. (1999).

Fourth, as describe above, we have carefully evaluated emission ratio and emission factor observations from past measurements of boreal forest fires in North America and Siberia, taking into account studies reported by Andreae (2019) and also reports provided by Yokelson in his review. A compilation of these studies will now be provided in our revised Table 1. In this context, we note that in comparison to North American boreal forest fires sampled by aircraft, our CO emission ratios are still considerably higher, implying our observations do provide evidence for stronger role of smoldering combustion. Fires in boreal Siberia tend to have even higher CO emission ratios than North American fires, which is consistent with well known differences in fire behavior between the continents (Rogers et al., 2015). We look forward to the reviewer's perspective on the new analysis in this revised Table. A draft of the table is included here.

| Study | CO Emission Ratio | MCE | # Fires |
|---|---|---|---|
| *Airborne Wildfires North America* | | | |
| Cofer et al., 1989 | 0.069 ± 0.004 | 0.935 ± 0.004 | 1 |
| Cofer et al., 1998 | 0.140 ± 0.012 | 0.878 ± 0.009 | 1 |
| Friedli et al., 2003 | 0.100 ± 0.020 | 0.909 ± 0.017 | 1 |
| Goode et al., 2000 | 0.085 ± 0.008 | 0.922 ± 0.007 | 4 |
| Laursen et al., 1992 | 0.050 ± 0.007 | 0.953 ± 0.006 | 1 |
| Nance et al., 1993 | 0.078 ± 0.012 | 0.928 ± 0.011 | 1 |
| O'Shea et al., 2013 | 0.150 ± 0.024 | 0.871 ± 0.012 | 4 |
| Radke et al., 1991 | 0.116 ± 0.087 | 0.896 ± 0.075 | 1 |
| Simpson et al., 2011 | 0.110 ± 0.070 | 0.901 ± 0.061 | 5 |
| **Fire Weighted Mean** | **0.102 ± 0.033** | **0.908 ± 0.027** | **19** |
| *Airborne Management Fires North America* | | | |
| Cofer et al., 1990 | 0.086 ± 0.008 | 0.921 ± 0.007 | 2 |
| Cofer et al., 1998 | 0.095 ± 0.016 | 0.913 ± 0.013 | 7 |
| Radke et al., 1991 | 0.047 ± 0.032 | 0.956 ± 0.030 | 4 |
| Susott et al., 1991 | 0.060 ± 0.061 | 0.943 ± 0.058 | 1 |
| **Fire Weighted Mean** | **0.077 ± 0.022** | **0.929 ± 0.020** | **14** |
| *Laboratory North America* | | | |
| Bertschi et al., 2003 | 0.151 ± 0.040 | 0.870 ± 0.030 | - |
| Burling et al., 2010 | 0.209 | 0.827 | - |
| Mcmeeking et al., 2009 | 0.091 ± 0.038 | 0.917 ± 0.068 | - |
| **Mean** | **0.150 ± 0.039** | **0.871 ± 0.049** | |
| *Siberia – Surface and Airborne* | | | |
| Cofer et al., 1998 (A) | 0.224 ± 0.036 | 0.817 ± 0.025 | 1 |
| McRay et al., 2006 (A & S) | 0.249 ± 0.064 | 0.800 ± 0.043 | 6 |
| Vasileva et al., 2017 (S) | 0.126 ± 0.007 | 0.888 ± 0.005 | 2 |
| **Fire Weighted Mean** | **0.219 ± 0.048** | **0.822 ± 0.033** | **9** |
| *Ground Wildfires North America* | | | |
| Wiggins et al., 2016 | 0.128 ± 0.023 | 0.887 ± 0.018 | 3 |
| This study | 0.142 ± 0.051 | 0.878 ± 0.039 | 35 |
| **Fire Weighted Mean** | **0.141 ± 0.049** | **0.879 ± 0.027** | **38** |

Fifth, we will add text to the methods to describe how we average the different individual fire events together to come up with a season-wide mean. We will explore the sensitivity of this to the averaging techniques, also reporting at $CO_2$ anomaly-weighted mean.

We respectfully ask the editors to allow us to update our calculations and revise the corresponding manuscript introduction and discussion prior to the next iteration of reviews. Below we address specific comments but note some of the responses will depend on our corrected results.

**Specific Comments:**

**Comment 1:**
P1, L14: define CRV
**Response: In our revised paper, we plan to change the sentence to read: "Here we quantified emission factors for CO and CH4 from a massive regional fire complex in interior Alaska during the summer of 2015 using continuous high-resolution trace gas observations from the Carbon in Arctic Reservoirs Vulnerability Experiment (CRV) tower in Fox, Alaska."**

**Comment 2:**
P1, L33 – P2, L2: will these aggressively lofted emissions impact tower? Run some forward/back trajectories? Vertical mixing?
**Response: To address this comment, in the methods section of the revised manuscript, we will add the following sentences: " Here we emitted fire emissions into the surface influenced volume of PWRF-STILT, which extends from the surface to the top of the planetary boundary layer, with the assumption that fire emissions were equally distributed within the planetary boundary layer [*Turquety et al.*, 2007; *Kahn et al.*, 2008]. In a previous study using the same tower, a sensitivity study revealed that plume injection height contributed only minimally to variability in simulated fire-emitted CO with PWRF-STILT [*Wiggins et al.,* 2016]."**

**Comment 3:**
P2, L6: "deadly" AQ is over-simplified
**Response: We will change "deadly" to "unhealthy."**

**Comment 4:**
2, 13: Andreae and Merlet was updated in 2019
**Response: We will update all of the appropriate references to Andreae and Merlet (2001) to Andreae (2019).**

**Comment 5:**
2, 18: The updated Andreae paper lists more than 20 studies, so there may be more worth including in Table 1.
**Response: As described above, we will update Table 1 and corresponding text to include the missing studies of field measurements of boreal forest fire emissions from Andreae (2019), along with other studies from laboratory measurements, studies that measured emissions from land management fires, and studies from Eurasian boreal forest fires that exist in the literature.**

**Comment 6:**

2, 28: Ground-based data downwind of fires has also been collected in Selimovic et al (2019a, b) and e.g. in the Colorado front range (Gilman, Benedict) and MBO (Collier and references there-in). The Wiggins 2016 data doesn't appear in this paper anywhere that I saw.

**Response: We will update this sentence to now read: "This approach has been used to estimate CO emission ratios during a moderate fire season in Alaska [*Wiggins et al.*, 2016] and for fires in other ecosystem types [*Gilman et al.*, 2015; *Collier et al.*, 2016; *Benedict et al.*, 2017; *Selimovic et al.*, 2019a,b]." We also added the fires sampled in Wiggins et al. (2016) to Table 1, using the same approach as described by our revised equation 2 and 3 to calculate emission factors.**

**Comment 7:**
2, 33-34: Akagi et al., 2011 explain how MCE can be used to estimate an arbitrary mix of smoldering and flaming over a continuous range.

**Response: We will add the following sentence on Page 2 Line 35 to include an explanation of how MCE and be used to estimate contributions from smoldering and flaming combustion: "The relative amounts smoldering and flaming combustion are difficult to measure, but can be estimated using the modified combustion efficiency (MCE) defined as $\Delta CO_2/(\Delta CO_2 + \Delta CO)$. Fire emissions dominated by flaming combustion have an MCE up to 0.99 while emissions dominated by smoldering combustion have an MCE often between 0.65 and 0.85 (Akagi et al., 2011). MCE can be used to understand the relative contributions from both flaming and smoldering fire processes." We also changed our criteria for separating the different combustion phases to align with previous studies.**

**Comment 8:**
2, 37: Real fires often don't have phases - rather a dynamic mix of processes. Change "phase" to "process" throughout?

**Response: This is a good point and we agree with the reviewer. We will change "phase" to "process" where appropriate throughout the manuscript.**

**Comment 9:**
3,22: The tower results, even if lowered are higher than "some" airborne studies. E.g. Cofer 98 is the same. If the EFs stay the same they are "a bit higher" than "some" previous estimates or recommendations.

**Response: We will revise the text to reflect our modified perspective after recomputing the emission factors. The text in section 4.2 will change to "Our emission factors for CO and CH₄ were in agreement with the mean of previous estimates for boreal fires derived from a compilation of all past studies. However, if studies that are not representative of North American boreal wildfires are excluded, including measurements from prescribed fires, laboratory studies, and studies of fires in the Eurasian boreal forest, our emission factors are 39% higher than average emission factors derived primarily from aircraft studies of wildfires in the North American boreal forest."**

**Comment 10:**
4, 6: Confusing, is it just 50 minute samples with ten minutes downtime per hour?

**Response: Yes, the tower collects continuous measurements for 50 minutes out of the hour. We will clarify this point by changing the text on Page 4 line 5 to "…to separate the dataset into a set of continuous 50-minute intervals of trace gas observations…"**

**Comment 11:**
4, 30: EFs are usually given for one species so the meaning of the ratio in the subscript here and in eqn 2 is not apparent. Suggest adopting standard notation?

**Response: We will remove the ratio in the subscript of our emission factors to align with standard notion.**

**Comment 12:**

4, 32: Do these references unambiguously support 45% C? Revisit, consider reference above, and explain in detail in revised text.

**Response: We now provide a referfence to Santin et al. (2015) and add text to explain the variability can range from 45 – 50%.**

Eqn (1) Sum or slope or simple subtraction?

**Response: To clarify we will add the following text to Page 4 Line 30: "Excess mole fractions denoted with a Δ symbol refer to observations of trace gas mole fractions during intervals when fire had a dominant influence on tower trace gas variability with background values subtracted."**

Eqn (2) Again, notation unusual, something common should work, or explain?

**Response: We will change the notation to: $EF_X = F_C * (1000g/kg) * MM_X/12.01 * ER_X/C_T$**

**Where Fc refers to the carbon content of the fuel (45), $MM_{CO}$ is the molecular mass of CO, $ER_{CO}$ is the emission ratio of CO relative to $CO_2$ and $C_T = \Sigma N_i * \Delta C_i/\Delta CO_2$.**

**Comment 13:**

5, 1-5: I get computing MCE for each sample, but what's the point of the categories that don't seem to be used?

**Response: We use the categories to separate our emission factor calculations and aid in the interpretation of our results as shown in Table 1, in Figure 4, and as discussed in section 3.1. Specifically, we use these categories to allow the reader to visually identify whether there is a trend toward one or another emissions type throughout the fire season. As shown in Figure 4, intervals with smoldering, mixed, and flaming emissions types were interspersed throughout the fire season.**

**Comment 14:**

Sec 2.3: There is not much detail on how AKFED is driven. One thing that stands out though is that the day/night split for fuel consumption is likely not right for 64 N! See Vermote et al, (2009); MODIS FRP can be higher at "night" than during the "day" in high latitude summer. This is relevant later.

**Response: We will add the following text to Section 2.3 to better explain how AKFED is created: "AKFED burned area is mapped using perimeters from the Alaska Large Fire Database combined with imagery from the Moderate Resolution Imaging Spectroradiometer (MODIS). Both above and belowground carbon consumption are modeled based on elevation, day of burning, pre-fire tree cover, and the difference normalized burn ratio (dNBR) [Veraverbeke et al. 2015]. AKFED predicts carbon emissions from fires with a temporal resolution of 1 day and a spatial resolution of 450 m."**

**We created the diurnal cycle of emissions specifically for the analysis here. We conducted additional analysis of the active fires and fire radiative power (FRP) from the MODIS fire detection products measured during the 2015 fire season in Alaska to assess our approach. The satellite data analysis reveals that the product of total number of active fires and FRP during the daytime Terra and Aqua overpasses accounts for 83% of total fire activity (the sum of fire activity from both daytime and nighttime overpasses). This is in line with our 90% day /10% night emissions split prescribed in the model. In this context, its important to note that if there was an afternoon satellite overpass 3 hour after Aqua (at 4:30pm), it would like be higher than the 10:30am Terra overpass, because relative humidity is lower and temperatures are considerably higher in mid-afternoon as measured from our earlier eddy covariance observation [Liu et al., 2005]. So the 83% estimate from MODIS is likely an underestimate of daytime fire activity. Vermote et al. (2009) concluded MODIS FRP can be higher at "night" than during the "day" in the boreal forest during summer, where "night" FRP is defined as the sum of FRP from both Terra overpasses and "day" is the sum of FRP from both Aqua overpasses.**

However, in this analysis for the summer of 2015, we found that the sum of FRP from both Aqua overpasses was higher than the sum from both Terra overpasses.

We changed the text to provide more justification for the 90/10 emission split we used:"Analysis of the product of fire radiative power and fire detections from the MODIS MCD14ML C6 product showed that 83% of fire activity occurred during daytime overpasses (10:30am and 1:30pm) relative to the sum across both daytime and nighttime overpasses during the 2015 Alaskan wildfire season (data not shown). The satellite observations provide broad support for the diurnal cycle we prescribed for emissions in the model."

Here is a figure showing the FRP, sum of active fires, and product of the two that illustrates how FRP and total number of active fires was considerably elevated during daytime overpasses during the 2015 fire season in Alaska.

[Figure]

**Figure 1. Panel A shows the mean FRP (MW) normalized by area (km$^2$) for all fires that occurred during the 2015 fire season in Alaska organized by the time and satellite of detection. B shows the total number of active fires, and C shows the product of A and B or the product of the area normalized mean FRP and the number of fires.**

**Comment 15:**
6, 1-9: It's likely that some weaker smoke peaks are more distorted by background variability and increase the range of values, but there is not necessarily bias.
**Response: We agree with the reviewer and believe our methodology is strict but unbiased.**

**Comment 16:**
6, 9: Many references support high correlation of EFCH4 with MCE.
**Response: In addition to the references already listed in section 4.3 we will add the following references to strengthen this point. The text will change to "A strong linear relationship existed between the CH$_4$ emission factor and MCE across the different sampling intervals (Figure 5). Linear relationships between CH$_4$ emission factors and MCE have also been observed in previous studies [*Yokelson et al.*, 2007; *Burling et al.*, 2011; *Van Leeuwen and van der Werf*, 2011; *Yokelson et al.*, 2013; *Urbanski*, 2014; *Smith et al.*, 2014; *Strand et al.*, 2016, *Guerette et al.*, 2018]. The relationship shown in Figure 5 implies MCE can be used as a metric for CH$_4$ emission factors from North American boreal forest wildfires when measurements of CH$_4$ are not available."**

**Comment 17:**
6, 38-39: This needs to be thought through a bit. Does the high impact of night smoke at the tower compared to the assumed low fraction of smoke produced at night mean day smoke was under-sampled? Or does this imply AKFED underestimates night smoke?
**Response: We will modify the analysis to span the same time intervals of the diurnal cycle that was applied to AKFED (0600 to 1800 for day and 1800 to 0600 for night). The text will read: "Overall, 73% of the fire emissions that impacted the tower occurred during the day (0600 to 1800 local time) and 27% occurred at night (1800 – 0600 local time)." AKFED has a daily resolution, but we accounted for diurnal variability in emissions by applying a diurnal cycle as explained in section 2.3. Our imposed diurnal cycle could be underestimating night smoke, or we could be measuring a slightly greater proportion of night smoke at the tower.**

**Comment 18:**
7, 3: "emissions" to "consumption"
**Response: We will change "emissions" to "consumption." The sentence will read: "The relative contributions of consumption from flaming and smoldering fires are uncertain for boreal forest fires…."**

**Comment 19:**
7, 8: 15 total previous fires sampled may be too low if you check updated compilations.
**Response: We will update the total number of previous fires sampled using studies included in updated compilations.**

**Comment 20:**
7, 9: Convection entrains some smoldering.
**Response: We changed the text in section 4.1 to the following: " … airborne sampling techniques struggle to measure emissions from less energetic smoldering combustion that emits smoke lower in the atmosphere [Selimovic et al., 2019a,b]. Emissions from smoldering boreal forest fires can**

sometimes be entrained in the convective columns of certain flaming fires and can be sampled by aircraft, but nighttime emissions or residual smoldering emissions from fires that have weak convective columns usually cannot [Ward and Radke, 1993; Bertschi et al., 2003; Burling et al., 2010]."

5 **Comment 21:**
7, 10-17: True, but a tower could potentially undersample flaming. Flaming is associated with rapid fuel consumption so not a negligible concern. Try forward trajectories from high injection altitudes to see if they impact tower or compare to column data?
**Response: We believe the tower is at an optimal location and height to sample integrated emissions**
10 **from both flaming and smoldering fires. The tower is on average 295 km away from the fires we sampled and located on a ridge that is over 600m above sea level. The long distance the emissions have to travel in order to reach the tower allows for mixing throughout the planetary boundary layer. Most of the fire emissions from boreal forest fires in Alaska remain in the PBL as shown by a MISR plume height analysis in Wiggins et al. (2016). To clarify, we will include the following text in section 3.2:**
15 **"CRV tower is sufficiently downwind to integrate both flaming and smoldering processes from fires across interior Alaska."**

**Comment 22:**
7, 19: Night may have been oversampled? But maybe not if there really is more emissions at night than was
20 assumed in AKFED? As noted above, there is sometimes more MODIS FRP at night than day in boreal regions. Also, as above, towers may not be sensitive to the entire range of injection altitudes? Explore?
**Response: We will add the following text to explain why the tower is not sensitive to injection altitude: "In a previous study using the same tower, a sensitivity analysis that included modifying the vertical resolution of the surface influenced volume of PWRF-STILT revealed that plume injection**
25 **height contributed only minimally to variability in simulated fire-emitted CO with PWRF-STILT [*Wiggins et al.,* 2016]."**

**Comment 23:**
7, 20-21: Quote these values from 2016 paper in Table 1?
30 Table 1 header or caption: It's enormous! Move part elsewhere. The text mentions CH4 data which I did not see in table.
**Response: We will add values from Wiggins et al. (2016) and other recent studies and updates as requested by the reviewer. We will also remove the reference to $CH_4$ data. We will move some of the caption to table footnotes.**
35
**Comment 24:**
7, 27: Cofer 98 agrees with this study's current values and the real average may be in middle of all this data somewhere.
**Response: This section of the discussion will change to reflect a new interpretation of our corrected**
40 **emission factor data.**

**Comment 25:**
7, 28-31: This data should certainly be used, but rarely does new data replace old data completely. More often new data contributes to an evolving literature average – sometimes with weighting by n factor.
45 **Response: We agree with the reviewer and offer a weighted average to use instead.**

**Comment 26:**
7, 36: You can often see smoke by satellite even when you can't detect FRP.
**Response: Although this is true, FRP is often used to estimate emissions and missing detections**
50 **correspond to missing emissions. We will change the original sentence "This residual smoldering**

**combustion could substantially contribute to trace gas emissions, but is difficult to detect and quantify using remote sensing because of low radiative power associated with this phase of combustion" to the following for clarity: "This residual smoldering combustion could substantially contribute to trace gas emissions but is usually excluded from FRP based fire emissions inventories because of the difficulty in detecting low FRP associated with this process of combustion."**

**Comment 27:**
8, 3-4: To claim a difference with the studies above you would have to know proportion of above-/below-ground fuel consumption that goes with those studies.
**Response: We will revisit this discussion section following our corrected results. We cannot directly compare with the overall magnitude of emissions, but we can compare with the emission ratios measured in previous studies.**

**Comment 28:**
8, 8: This is a common error to assume that increased EFs will lead to increased, modeled health impacts. Models use EF*biomass burned to get a-priori emissions. Then the modeled impacts are compared to downwind monitors and the a-priori emissions are adjusted to best match reality. A higher EF may change the details of the tweaking procedure, but not change the downwind PM. What would change the latter is discovering a problem with the PM monitors.
**Response: We appreciate the reviewer's comment, but respectfully disagree. There has been a long standing low discrepancy between fire emissions and observed $PM_{2.5}$ [Huang et al., 2013; Redding et al., 2016; Christopher et al., 2019; Liu et al., 2020]. Higher emission factors will require much less tweaking to the a-priori emissions by increasing the accuracy of the magnitude of the emissions.**

**Comment 29:**
8, 10: "lead to"
**Response: We will change "lead" to "lead to."**

**Comment 30:**
8, 17: Towers are not completely new. There was a long history of sampling prescribed fires from towers carried out by the Fire Lab.
**Response: We agree with the reviewer and added the following text to clarify our approach refers to towers in the boreal forest: "Our tower-based approach to calculate emission factors has been used in other ecosystems, and is a technique that significantly improves our understanding of trace gas emissions specifically from boreal forest fires."**

**Reviewer #2 Comments and Responses**

**General comments:**

The discussion paper presents important research into the characteristics of wildfire emissions using established techniques, but novel analysis. Teasing apart the contributions of various fire events and the combustion stage (Flaming vs. smoldering) is a new and valuable way to understand nuances of boreal fire relevant to many needs, such as human health, carbon cycling, and smoke planning. However, the paper falls short in many ways, and will need some extensive modification to reach its potential. I strongly suggest a re-focus on a more relevant outcome from the work (rather than the fact that previous work was not catching smoldering as well as they could), a fully revised Discussion (some ideas below), and some attention to references (see notes below). This work is very important, and when presented well will make a great contribution to the literature on this subject.

**Response: We appreciate the reviewers comment that this paper offers important insight into the characteristics of boreal forest fire emissions. We will systematically revise the discussion in response the the reviewer's comments and those from the other reviewers.**

**Specific comments:**

**Comment 1:**
1. The title will need modification. It is unclear what "larger" refers to – larger than what? Than previous studies (yes, but I know that only when I get to the end of theAbstract). It could be larger than flaming combustion. The point is that having an unreferenced comparative adjective can be troublesome, especially in a title where you want to be clear. The title could be the same, but with the first four words dropped: "Contribution of . . .". Also, it is my opinion that, while this may show larger contribution than previous studies, this work has a lot of other implications and contribution that could be highlighted in the title. In some ways the community would not be too surprised to learn that the smoldering fire signal has not been captured in previous studies, so
highlighting this part of it is not needed to make this an impactful paper/study.
**Response: Our title will change to align with our updated calculations and their implications. Our new title is "Boreal forest fire CO and $CH_4$ emission factors from tower observations in Alaska during the extreme fire season of 2015." We chose this title to highlight that our emission factors were measured during an extreme fire year using an approach that integrates emissions from fires over longer time scales than traditional aircraft based studies. We appreciate the suggestion to broaden the implications of our study and avoid highlighting undersampled smoldering fire emissions. It is likely our new emission ratios are higher than reports in previous aircraft studies, and we discuss this in the main text.**

**Comment 2:**
2. The comparison to previous studies would more naturally go into the discussion, rather than the introduction/background. I suggest revising to put Table 1 into the discussion where you can make the case more directly, rather than introducing the previouswork without yet seeing your results.
**Response: We appreciate the reviewer's suggestion, and will bring the reader's attention to Table 1 in the introduction. We will add new discussion and analysis of the implications from an updated version of Table 1.**

**Comment 3:**
3. There is a blatant and concerning misuse of terminology on Page 2, line 34: The sentence "Smoldering combustion can be defined as combustion with a degree ofcombustion completeness, or modified combustion efficiency, less than 0.9 [Urbanski2014]." First, MCE and combustion completeness (CC) are very different things. CC is the proportion of fuels consumed/combusted, while MCE is defined as the proportion of a gas to CO2. Second, the Urbanski paper puts MCE of 0.65 to 0.85 as "smoldering", and references Akagi et al. 2011 so I don't know where the 0.9 figure
comes from. The choice of the thresholds stated on page 5 lines 1-4 need to be better justified.
**Response: We agree with the reviewer that combustion completeness needs to be removed from the sentence. We have updated our criteria for separating the combustion processes to align with previous studies. The revised text now reads "The relative amounts smoldering and flaming combustion are difficult to measure, but can be estimated using the modified combustion efficiency (MCE) defined as $\Delta CO_2/(\Delta CO_2 + \Delta CO)$. Fire emissions dominated by flaming combustion have an MCE while emissions dominated by smoldering combustion have an MCE often between 0.65 and 0.85 [Akagi et al., 2011; Urbanski et al., 2014]. MCE can be used to understand the relative contributions from both flaming and smoldering fire processes."**

**Comment 4:**

4. I found a couple of instances where the citations used are inappropriate. While I mention only 2 here, I would suspect others, so the citations need to be fully vetted for appropriateness. First: "Rogers et al. 2015" in Page 1 line 33 is not a review of borealfire regime. It may mention this,￼ it is not what that study provides to the literature. Second: "Bertschi et al. 2003" in Page 7 line 34 is of laboratory experiments and work in savannah ecosystems, not boreal forest fires. In both of these cases, it could be argued that no reference is needed. If you do include a reference, it needs to be a paper or resource where the statement made is shown or studied, not where it was stated. I suggest the co-authors assist with improving the citations.

**Response: We thank the reviewer for pointing out our errors in citations. We will thoroughly revisit the citations throughout the manuscript and make adjustments where necessary. We will edit the reference in Page 1 line 33 to include Johnstone et al. (2011), but prefer to keep the reference to Rogers et al. (2015) because although the primary goal of this paper was to highlight differences between the boreal fire regime in North America and Eurasia, it highlights the high energy crown fires that occur in the North American boreal forest. Rogers et al. (2015) is also used as a reference in section 4.1 of the discussion. In the introduction we will will make the following citation changes: change McGuire et al. (2010) to Kasischke (2000), remove a reference to French et al. (2004), replace Turquety et al. (2004) with Harden et al. (2000), and add Fromm et al. (2000).**

**Comment 5:**

5. The discussion would benefit from more regarding the implications of the results. What is the data showing us that is relevant? Some possible ideas to highlight/discuss
(these need to be discussed with co-authors, so are only representative):

**Response: We plan to completely revise the main text of the discussion. The discussion will have a detailed discussion of the implications of our findings relative to past work summarized in Table 1.**

a. Figure 5 (Page 6 line 9) shows a linear relationship between CH4 and MCE. Provide a short discussion of this in the discussion – what does this mean for using the data?

**Response: We added the following text to the discussion: "We found a strong linear relationship between $CH_4$ emission factors and MCE that has also been observed in previous studies [*Yokelson et al.*, 2007; *Burling et al.*, 2011; *Van Leeuwen and Van Der Werf*, 2011; *Yokelson et al.*, 2013; *Akagi et al.*, 2014; *Smith et al.*, 2014; *Urbanski et al.*, 2014; *Strand et al.*, 2016, *Guerette et al.*, 2018]. There is a wide range of slopes between $CH_4$ and MCE that have been found in prior studies and could be dependent on fuel type and burning conditions [*Smith et al.*, 2014]. This implies MCE could be used as a metric for $CH_4$ emissions when measurements of $CH_4$ are not available, but care should be taken to ensure the MCE and $CH_4$ relationship used is for the correct ecosystem."**

b. Page 6 line 22 – ". . .attributed to boreal fire emissions." – As opposed to what?? Or why? A bit of discussion on what other factors contribute to the signal, and why there are some difference in the model will help non-atmospheric modelers better understand why these results are so powerful

**Response: We changed the text to read: "The forward model simulations combining AKFED fire emissions with PWRF-STILT confirmed that the elevated CO signals at the CRV tower can be attributed primarily to boreal forest fire emissions (Figure 7), as opposed to fossil fuel or other CO emissions sources. The AKFED model had a Pearson's correlation coefficient of 0.61 with observed daily mean CO and had a low bias of approximately 7%. Differences between the model simulations and observations were likely￼ caused by errors in the magnitude and timing of fire emissions within AKFED as well as the limited spatial resolution and incomplete representation of atmospheric transport within PWRF-STILT."**

c. The temporal distribution data (Fig 10) is very interesting and could be helpful for

exposure assessment for health studies. (although PM, rather than CO would be of
interest).
**Response: We added the following sentence: "The timing of emissions is important for quantifying the impact on human health, and enhanced nighttime emissions (Figure 10) when the boundary layer is much lower could increase surface concentrations and exacerbate negative health effects."**

d. Page 7, line 24: I am not sure I see a temporal trend in the old data, and I am not
sure why this would be something to note. This statement is best dropped. Table 1
presents past results that are collected in a variety of settings, so (in my assessment)
represents some data on the range of variability, not a record of change over time.
I hope these comments inspire the authors to revise the manuscript for a more useful.
**Response: We appreciate the insight offered by the reviewer to improve our discussion. We agree the implications need to be revisited and the discussion will be significantly revised based on our corrected emission factor calculations and themes suggested by the reviewer.**

**Reviewer #3 Comments and Responses**

**General comments:**

The authors present an impressive set of CO, CH4 and CO2 measurements in boreal forest fire smoke to calculate
emission factors for CO and CH4. Such accurate data are neccesarry to soundly test models used to quantify the impact
of big fires on the air quality and climate. Therefore, the paper is highly suitable to be published in this journal. Yet,
due to error in the emission factors calculation I suggest a revision of this manuscript. Re-interpretation of results
should be done by the authors before resubmission. Firstly, Eq 2 on Page 4 should be revised, using e.g. Eq. 1 and 2 in
Yokelson et al. I'm looking forward to review the revised manuscript in detail. I add here only a potential useful
comment. Are there any other measurements of specific tracers to be used to quantify the smoldering/flaming
contributions? Flaming is likely under-represented in the used sampling height.
**Response: We appreciate the reviewer's comment that our study provides accurate CO and CH$_4$ emission factor data needed to quantify the impact of boreal fires on air quality and climate. We agree that the manuscript will need revision and reinterpretation of results to reflect the corrected emission factor calculations. We have added text in section 2.1 and 3.2 to explain why the tower is an ideal location to measure emissions from both smoldering and flaming fires. The text in section 2.1 reads "In a previous study using the same tower, the authors conducted a sensitivity study on the CRV tower and found little influence of plume injection height on CRV tower trace gas observations [Wiggins et al., 2016]." The text in section 3.2 reads "CRV tower is sufficiently downwind to integrate both flaming and smoldering processes from fires across interior Alaska."**

[revised manuscript text omitted]

Emissions from boreal fires can significantly influence atmospheric composition throughout the Northern Hemisphere. Fire plumes from regional fire complexes in Alaska and western Canada, for example, have been shown to influence air quality over Nova Scotia [*Duck et al.*, 2007] and across the south-central US [*Wotawa et al.*, 2001; *Kasischke et al.*, 2005] and Europe [*Forster et al.*, 2001]. Similarly, emissions from boreal forest fires in Russia have caused unhealthy air quality in Moscow [*Konovalov et al.*, 2011] and have affected ozone and other trace gases concentrations across the western US [*Jaffe et al.*, 2004]. Over the past few decades, annual burned area in several regions in boreal North America has increased [*Gillett et al.*, 2004; *Kasischke and Turetsky*, 2006; *Veraverbeke et al.*, 2017], and future projections suggest further increases may occur in response to changes in fire weather and a lengthening of the fire season [*Flannigan et al.*, 2001; *de Groot et al.*, 2013; *Young et al.*, 2017]. As a consequence, fires are likely to play an increasingly important role in regulating air quality and climate feedbacks during the remainder of the 21st century.

Emission factors provide a straightforward way to convert fire consumption of dry biomass into specific trace gas species, such as CO, CH$_4$, and CO$_2$. This technique is commonly used to model emissions of select species and or to compare model results with in-situ or remotely sensed observations. The most frequently used boreal forest fire emission factors are derived from meta-analyses that average together information from individual field campaigns [*Andreae and Merlet*, 2001; *Akagi et al.*, 2011; *Andreae*, 2019]. These syntheses often include in-situ airborne and ground based measurements along with laboratory measurements of combusted fuels. There is no consensus on how to combine information from different studies, and in past work individual studies have been given equal weight when estimating biome-level means, even when the number of fires and duration of sampling has varied considerably from one field campaign to another. A summary of previous studies that measured CO emission ratios for boreal forest fires is shown in Table 1.

[revised manuscript text omitted]

**4 Discussion**

**4.1 Measurement technique, and ecosystem type as drivers of variability in boreal forest fire emission ratios**

The most widely used emission factors for boreal forest fires are derived from syntheses that average together data from individual field campaigns [*Andreae and Merlet*, 2001; *Akagi et al.*, 2011; *Andreae*, 2019]. In order to investigate the possible influence of sampling strategy employed by previous studies, and variations caused by ecosystem type, we compiled available studies that report CO emission ratios for boreal forest fires and organized the studies into several categories with common characteristics, including aircraft sampling of North American boreal forest wildfires, aircraft sampling of North American boreal forest management or prescribed fires, combustion of North American boreal forest fuels measured in the laboratory, and sampling of Siberian boreal fires from both aircraft and surface platforms (Table 1). All previous studies combined have sampled a total of 39 individual boreal forest fires for CO emission ratios, and additional measurements have been made by combusting fuels in a laboratory setting. We found several important differences in emission ratios that may be linked with the measurement technique and ecosystem type.

First, solely considering emission ratio measurements from boreal North America, our surface tower measurements of about 35 fires, along with earlier tower measurements from Wiggins et al. [2016] have a considerably higher mean (0.141) than the mean of aircraft measurements sampling wildfires (0.102) or management and prescribed fires (0.077). We believe these differences are linked, in part, with sampling strategy. Aircraft-based studies often sample fires that have a strong contribution from flaming combustion, which releases enough energy to generate well-defined plumes at an altitude accessible by the aircraft. This methodology provides an opportunity to comprehensively measure the vertical and horizontal distribution of emissions from an individual fire and their atmospheric evolution in a smoke plume. However, airborne sampling techniques are often limited to daytime periods with good visibility, making it difficult to comprehensively measure emissions over a diurnal cycle or over the full lifetime of a fire which may span several periods with inclement weather. Due to these sampling constraints, aircraft studies are less likely to measure emissions from less energetic smoldering combustion, since these emissions are more likely to remain near the surface [*Ward and Radke*, 1993; *Selimovic et al.,* 2019a]. Emissions from smoldering boreal forest fires can sometimes be entrained in the convective columns of certain flaming fires and can be sampled by aircraft, but nighttime emissions or residual smoldering emissions from fires that have weak convective columns usually cannot be measured in this way [*Bertschi et al.,* 2003; *Burling et al.,* 2012]. Near the end of the lifetime of a long-lived fire, aircraft measurements have sometimes observed a larger smoldering to flaming ratio [*Yates et al.*, 2016]. A few previous studies have investigated the differences in emissions measurements from ground and aircraft sampling of the same fire, reporting significant differences between the relative abundance of the emissions observed depending on the sampling method [*Christian et al.,* 2007; *Burling et al.,* 2011; *Akagi et al.,* 2014]. Emission ratios derived from aircraft measurements are more likely to sample fires during times when flaming combustion processes are dominant [*Babbitt et al.,* 1996, *Akagi et al.*, 2014], yet rarely sample residual smoldering combustion that can substantially contribute to emissions over the full lifetime of an individual fire [*Bertschi et al.*, 2003].

Second, we also separated aircraft-based studies that measured emissions from wildfires from those that measured emissions from prescribed slash and land management fires, where trees are bulldozed, dried and intentionally arranged to promote
* * *
**Moved down [4]:** The time delays between emission and detection of trace gas anomalies at CRV allowed for atmospheric mixing of signals from dozens of different fires in different stages of growth and extinction.

maximum fuel consumption [*Cofer et al.*, 1990; *Cofer et al.*, 1998]. Land management fires consume dried aboveground fuels with a different fuel structure and moisture content than fuels consumed in a wildfire, where combustion from soil organic material layers is a dominant component of bulk emissions [Boby et al., 2010; *Dieleman et al.*, 2020]. Although the number of land management fires is relatively small, the mean from these studies suggest flaming processes are a more important contributor to
5 this fire type than for wildfires, and some consideration of this difference should be factored into regional and global syntheses.

Third, three studies report emission ratios from laboratory combustion of fuels collected from North American boreal forests including biomass samples from black spruce, white spruce, and jack pine, as well as moss and surface organic material (duff). The laboratory studies have considerable variability that can be attributed to the type of fuel combusted and fuel moisture content. This work indicates duff consumption yields much higher emission ratios for CO and $CH_4$ than combustion of black spruce
10 or jack pine needles and other fine fuels [*Bertschi et al.*, 2003; *Mcmeeking et al.*, 2009]. The fuels used in laboratory studies are usually dried and burned individually, although some studies have attempted to mimic natural fires by placing dried fine fuels on top of damp fuels that undergo residual smoldering combustion [*Bertschi et al.*, 2003]. The structure, composition, and moisture content of fuels are well known as key drivers of the composition and magnitude of emissions. Although these laboratory studies provide valuable information on emissions from individual fuel components, they are not able to capture the full complexity of a
15 wildfire.

Fourth, emission factors from the Siberian boreal forest are often grouped together with emission factors from North American boreal forest in biome-level syntheses [*Andreae*, 2019]. Yet, Table 1 shows emission ratios from wildfires in boreal Siberia tend to be higher than emission ratios from North American wildfires. Although more measurements are needed, higher CO emission ratios for Siberian fires appears consistent with past work showing that boreal fire behavior is fundamentally different
20 between North American and Siberian continents as a consequence of differences in tree species and their impacts on fire dynamics. Notably, as consequence of the presence of black spruce in many boreal forests of North America, fires tend to burn hotter and faster, traveling through the crowns of trees and inducing higher levels of tree mortality [*Rogers et al.*, 2015]. This occurs because black spruce is a well-known fire embracer, retaining dead branches that serve as ladder fuels– carrying fire into the overstory where seeds in serotinous cones are activated by fire. Black spruce trees are absent from Siberia, where many pine and larch tree
25 species lack ladder fuels and are known as fire resistors. In Siberian ecosystems ground fires are more common [*Rogers et al.*, 2015], a finding that appears consistent with the higher CO emission ratios (and stronger contribution of smoldering combustion) shown in Table 1.

**4.2 Evidence for a stronger role of smoldering combustion in emissions from North American boreal wildfires**

Our mean emission factor for CO ($127 \pm 59$ g CO per kg of dry biomass consumed) is similar to the mean reported in past
30 syntheses for boreal forests, including estimates by *Andreae* [2019] ($121 \pm 47$ g CO per kg of dry biomass consumed) and *Akagi et al.* [2011] ($127 \pm 45$ g CO per kg of dry biomass consumed). However, if studies that are not representative of North American boreal forest wildfires are excluded (including measurements from prescribed fires, laboratory studies, and studies of fires from the Siberian boreal forests) and we focus on emission ratios, to avoid uncertainties introduced by the limited number of measurements that report the carbon content of combusted fuels, our estimate is 39% higher (and significantly different at a $p <$
35 0.01 level using a Student's t test) than the mean derived from aircraft studies of North American boreal wildfires (Table 1).

Considering the higher emission ratio of our measurements, we believe the CRV observations we analyzed here provide evidence that boreal forest fires in North America have a stronger contribution from smoldering combustion than what has been estimated in previous reports. Our CRV tower-based sampling was able to integrate over day-night burning cycles, flaming combustion at active fire fronts as well as residual smoldering combustion in soils that persists for days after the fire front moves

**4.2 Integration of emission factor observations across studies and time intervals**→

through an area, and emissions associated with a wide range of environmental conditions that occurred during 2015 fire season. This integration was possible because the tower was located at a higher elevation (611 m above sea level) and several hundred kilometers downwind of the core fire complex located in western Alaska. The time delays between emission and detection of trace gas anomalies at CRV allowed for atmospheric mixing of signals from dozens of different fires in different stages of growth and extinction. Collectively, these fires appeared to experience time-varying environmental conditions that were less ideal for flaming combustion than the fire plumes sampled in past work by aircraft.

Following ignition, North American boreal forest fires generally expand with flaming combustion in the crown. Smoldering combustion in organic soil layers and coarse woody debris behind the fire front that can continue for weeks after ignition [*Bertschi et al.*, 2003]. This residual smoldering combustion could substantially contribute to trace gas emissions but is usually excluded from FRP based fire emissions inventories because of the difficulty in detecting low FRP associated with this process of combustion. The relative contributions of consumption from flaming and smoldering are uncertain for boreal forest fires, although several previous studies have assumed 80% of aboveground carbon is consumed in flaming combustion, 20% is consumed in smoldering combustion, and vice versa for belowground carbon [*French et al.*, 2002; *Kasischke and Bruhwiler*, 2002]. Our results suggest that the smoldering process of combustion contributes to more to CO emissions than previously thought.

In the context of interpreting the CRV measurements, it's important to note that MISR satellite observations from Alaskan wildfires indicate most fire plumes reside within the planetary boundary layer, which is typically between 1 and 3 km during midday in summer [*val Martin et al.* 2010; *Wiggins et al.*, 2016]. Combining this length scale with the mean distance of the 35 fires that most influenced CO at CRV (295 km), we obtain a factor of about 100 for a back-of-the-envelope ratio of horizontal to vertical mixing processes. This implies that mesoscale atmospheric circulation played an important role in delivering fire-emitted trace gases to CRV. This ratio is considerably larger than what would be inferred from the location and sampling strategy of several past studies that have used surface towers to sample fires near or within fire perimeters [*Collier et al.*, 2016; *Benedict et al.*, 2017; *Selimovic et al.*, 2019a,b], band highlights the unique role that a remote tower can have in providing an integrated assessment of a large regional fire complex.

Finally, we note that during the latter half of June and early July of 2015, weather in Alaska was very hot and dry, allowing a record number of fires to rapidly expand in size, and yielding the second highest level of annual burned area in the observed record. The extreme fire weather conditions would be expected to reduce fuel moisture content, thus promoting flaming combustion processes. This raises the question of whether longer term monitoring of many normal and low fire years (which tend to co-occur in cooler and wetter conditions) would provide evidence for an even larger role of smoldering combustion for wildfire emissions from these ecosystems. Another related question is whether even within a fire season, do day-to-day or week-to-week variations in fire weather influence variability in emission ratios. We explored this latter question with the datasets described here but were unable to uncover structural relationships between daily meteorological variables such as vapor pressure deficit and CO emission ratios. Together, these questions represent important directions for future research and emphasize the critical need of sustained long-term support for trace gas monitoring networks and field campaigns.

**4.3 Synthesis of emission factor observations across studies**

With many new field campaigns measuring fire emissions, there is a need to revisit how information from different studies is combined to generate the most reliable set of emission factors for regional and global atmospheric models. Several ideas for an improved synthesis have emerged from our study.

First, it may make sense to separately report emission factors for Siberian and North American boreal forest fires, given what we know about differences in species composition, fire dynamics, and measurements of emission factors between the two
* * *
**Moved (insertion) [4]**

→ Our modeling study confirms that the entire day/night fire cycle was captured by anomalous trace gas observations at CRV tower that was used to calculate emission factors. *Wiggins et al.* [2016] used a similar tower-based approach to estimate boreal forest emission factors during a moderate fire year, and they found CO and CH₄ emission factors that were higher than the compiled mean from previous studies. We found a strong linear relationship between CH₄ emission factors and MCE that has also been observed in previous studies [*Van Leeuwen and Van Der Werf*, 2011; *Yokelson et al.*, 2013; *Urbanski*, 2014]. ¶
→ Although Table 1 appears to suggest CO emission factors from boreal forest fires are increasing over time, it is more likely that studies using the tower approach are better suited to sample a more thorough representation of all the phases of combustion that can occur in boreal forest fires. The tower approach is not limited by the time or scale of sampling, unlike aircraft measurement techniques. Aircraft based emission factors are often biased towards flaming fires, because most measurements are acquired during the afternoon when active fire plumes are visible. The emission factors derived from this study provide a more robust estimate of the mean, and indicate that the smoldering phase and nighttime emissions of boreal fires have likely been underestimated in previous studies. The improved emission factors from this study can be used in future modeling efforts to convert carbon emissions to CO and CH₄ trace gas emissions from boreal forest fires more accurately. ¶

**4.3 Relative Contributions of Smoldering and Flaming Combustion** ¶
→

continents. More data, particularly for Siberian fires, is needed to assess whether the differences in emission factors noted here are robust.

Second, it's important to further explore ways to weight the information content from different studies, considering the number of fires sampled, the duration and intensity of sampling, the representativeness of the sampling approach, and the representativeness of the fire complexes that were sampled relative to the typical pattern of burning within a biome. Here using a remote surface tower, we were able to get an integrated estimate of CO and $CH_4$ emission ratios from about 35 wildfires from an ecologically significant regional fire complex. While these observations represent a step change in CO and $CH_4$ data availability for North American boreal forest fires, more work is needed to find a way to systematically combine this information with other observations generated using different sampling techniques.

Third, even for an individual fire, steps toward flux-weighting different emission factors would be an important path toward reducing uncertainties, yet this goal remains technically challenging given existing measurement techniques. Our sensitivity analysis, in which we computed a weighted-mean CO emission ratio using CO or $CO_2$ concentrations during each fire-affected sampling interval, provided an indication of the robustness of our mean estimates to weighting scheme. This approach falls short, however, of providing a flux-weighted estimate given atmospheric processes that may decouple concentration from flux, including, for example, variations in windspeed, diurnal variations in planetary boundary layer height, and the distance between the emissions source and measurement point. To make progress on this issue, a closer integration is needed in future field campaigns between instantaneous measurements of fire behavior (temperature, fire radiative power, and spread rate), measurements of emissions composition, and post-fire sampling of fuel structure and consumption during times when fire dynamics were fundamentally different. This coordination across disciplines in both study design, data analysis, and modeling is rare and may provide a path toward creating the observations needed to dynamically model the temporal evolution of the chemical composition of wildland fire emissions over the lifetime of an individual fire and during different phases of a fire season.

**4.4 Implications of a larger contribution of smoldering combustion**

Smoldering combustion produces significantly more CO and $PM_{2.5}$ than flaming combustion [*Bertschi et al.*, 2003; *Chen et al.*, 2007; *Stockwell et al.*, 2016], and our work suggests North American boreal forest fire emissions of these species are likely higher than previous thought. This conclusion implies changes to the overall impact of boreal forest fires on human health, atmospheric composition, and climate. Emissions from boreal forest fires have the potential to be transported long distances across the Northern Hemisphere [*Forster et al.*, 2001], implying large-scale impacts. CO can lead to enhanced tropospheric ozone production downwind of a fire [*Lapina et al.*, 2006], and higher concentrations of CO from fires may indirectly contribute to radiative forcing by consuming hydroxyl radicals and extending the lifetime of $CH_4$ [*Levine and Cofer*, 2000]. $PM_{2.5}$ emissions, in contrast, can significantly degrade regional air quality, endanger cardiovascular and respiratory health, and influence the radiative balance of the planet [*Reid et al.*, 2016]. The timing of emissions is important for quantifying the impact on human health, and enhanced nighttime emissions (Figure 10) when the boundary layer is much lower could increase surface concentrations and exacerbate negative health effects. Much of the $PM_{2.5}$ emitted by smoldering fires is composed of organic carbonaceous aerosol that often leads to climate cooling [*Tosca et al.*, 2010; *Jayarathne et al.*, 2018].

**5 Conclusions**

Using a remote tower downwind of a large regional fire complex in interior Alaska, we measured CO and $CH_4$ emission factors from about 35 individual fires during the summer of 2015. This is more than the number of individual wildfires that have
* * *
been sampled in North America in all previous studies combined. Our results suggest smoldering combustion processes in North American boreal forest fires contribute more trace gas emissions than previously thought, and as a consequence, total CO emissions may have been underestimated in model simulations of boreal forest fire impacts on atmospheric composition. Long-term monitoring from remote towers may provide a means to quantitatively sample fire complexes in other biomes, integrating across day-night variations in fire behavior, periods with different environmental conditions, and across multiple fires in different stages of growth and extinction.

**Acknowledgements**

E.B.W. thanks the U.S. National Science Foundation for a Graduate Research Fellowship (NSF 2013172241). The CRV tower observations and footprints used in our analysis are archived at the U.S. Oak Ridge National Laboratory Distributed Active Archive Center for Biogeochemical Dynamics (http://dx.doi.org/10.3334/ORNLDAAC/1316). The trace gas observations, fire emissions time series, and WRF-STILT model were created through funding support to NASA's CARVE field program led by C. Miller. JTR acknowledges additional NASA support from CMS (80NSSC18K0179), IDS (80NSSC17K0416), and SMAP (NNX16AQ23G) programs. We thank the staff of the NOAA Fairbanks Command and Data Acquisition Station that hosts CRV, and especially Frank Holan and Marc Meindl, for their technical support. We also thank NOAA/ESRL/GMD staff, especially Phil Handley, Jon Kofler, and Tim Newberger for ongoing remote maintenance of CRV.

**Figures**

[Figure]

**Figure 1.** The location of wildfires in Alaska during 2015, with color representing the day of burning from the Alaska Fire Emissions Database (AKFED). The black circle denotes the location of CRV tower.

[Figure]

**Figure 2.** A)  Observations of $CO_2$ mole fraction from CRV tower in 2012 (black) along with model estimates of the $CO_2$ background (green) at CRV using the approach described in the main text. Very few fires occurred during 2012, and as a consequence most of the $CO_2$ variability in the observations and in the model is associated with terrestrial net ecosystem exchange. B) In 2015 wildfires in interior Alaska contributed significantly to $CO_2$ variability at the CRV tower, causing positive anomalies in the observations shown in black, particularly between days 170 and 190. The modeled background for 2015 is shown in red. The $CO_2$ mole fraction observations and model estimates have a 1 hour temporal resolution.

Deleted: **Figure 2.** Panel A: 2012 $CO_2$ observations from CRV tower (black) during a low fire year versus modeled background $CO_2$ (green). Panel B: 2015 $CO_2$ observations from CRV tower (black) versus modeled background $CO_2$ (red)....

[Figure]

[Figure]

**Figure 3.** Trace gas observations at the CRV tower during the summer of 2015 for A) CO, B) CH₄, and C) CO₂. The trace gas observations are plotted at a 30 s temporal resolution. Daily active fire detections derived the MODIS instrument on Terra and Aqua satellites (MCD14ML C6) are shown in panel D.

[Figure]

[Figure]

**Figure 4.** CRV tower observations of A) CO, B) CH₄, and C) CO₂ are shown along with periods used to calculate emission ratios. The dominant process of combustion is noted with blue for smoldering (blue), purple for mixed, and red for flaming. The trace gas observations are plotted at a 30 s temporal resolution.

[Figure]

[Figure]

**Figure 5.** Relationship between CH$_4$ emission factor and modified combustion efficiency (MCE). The strong linear relationship indicates that periods with more smoldering combustion (with a smaller MCE) produce significantly higher levels of CH$_4$ emissions. The relationship was defined by a slope of -46.37 ± 4.13 g CH$_4$ per kg dry biomass per MCE, an X intercept of -0.47 ± 0.05 g CH$_4$ per kg dry biomass, an R$^2$ of 0.54, and a significance value of p <0.01.

[revised manuscript text omitted]

factors (g per kg of dry biomass combusted), and modified combustion efficiency (MCE). Dominant combustion process (CP) is described as flaming, mixed, or smoldering.

|   | Fire Name | Distance (km) | Contribution (%) | Total Hectares | Fuel Type | Ignition Source |
|---|---|---|---|---|---|---|
| 1 | Tozitna | 229 | 10.74 | 31652 | Black Spruce | Lightning |
| 2 | Kobe | 119 | 7.20 | 3444 | Black Spruce | Lightning |
| 3 | Blair | 82 | 6.31 | 15217 | Black Spruce | Lightning |
| 4 | Aggie Creek | 41 | 5.63 | 12829 | Black Spruce | Lightning |
| 5 | Spicer Creek | 195 | 5.30 | 39761 | Black Spruce | Lightning |
| 6 | Blind River | 252 | 3.87 | 24608 | Black Spruce | Lightning |
| 7 | Holtnakatna | 404 | 3.44 | 90308 | Mixed | Lightning |
| 8 | Blazo | 514 | 3.39 | 49106 | Black Spruce | Lightning |
| 9 | Big Creek 2 | 351 | 3.23 | 126637 | Black Spruce | Lightning |
| 10 | Chitanana River | 241 | 3.12 | 17483 | Black Spruce | Lightning |
| 11 | Sea | 309 | 3.06 | 172 | Black Spruce | Human |
| 12 | Sushgitit Hills | 276 | 2.92 | 111712 | Black Spruce | Lightning |
| 13 | Big Mud River 1 | 254 | 2.72 | 42076 | Black Spruce | Lightning |
| 14 | Lost River | 347 | 2.58 | 21088 | Black Spruce | Lightning |
| 15 | Munsatli 2 | 302 | 2.36 | 40682 | Black Spruce | Lightning |
| 16 | FWA Small Arms Complex | 19 | 2.31 | 740 | Black Spruce | Prescribed |
| 17 | Tobatokh | 280 | 2.24 | 21868 | Black Spruce | Lightning |
| 18 | Trail Creek | 363 | 2.24 | 11939 | Black Spruce | Lightning |
| 19 | Lloyd | 201 | 2.22 | 26818 | Black Spruce | Lightning |
| 20 | Isahultila | 342 | 2.17 | 60445 | Black Spruce | Lightning |
| 21 | Nulato | 499 | 2.17 | 449 | Black Spruce | Lightning |
| 22 | Three Day | 472 | 2.17 | 39378 | Black Spruce | Lightning |
| 23 | Hay Slough | 188 | 1.90 | 37007 | Black Spruce | Lightning |
| 24 | Rock | 316 | 1.83 | 3714 | Other | Lightning |
| 25 | Sulukna | 329 | 1.77 | 6760 | Black Spruce | Lightning |
| 26 | Titna | 273 | 1.77 | 12415 | Black Spruce | Lightning |
| 27 | Quinn Creek | 657 | 1.49 | 2002 | Other | Lightning |
| 28 | Harper Bend | 188 | 1.45 | 17555 | Black Spruce | Lightning |
| 29 | Hard Luck | 328 | 1.43 | 5230 | Black Spruce | Lightning |
| 30 | Fox Creek | 369 | 1.42 | 2346 | Black Spruce | Lightning |
| 31 | Bering Creek | 280 | 1.36 | 45654 | Black Spruce | Lightning |
| 32 | Eden Creek | 324 | 1.16 | 18614 | Black Spruce | Lightning |
| 33 | Falco | 390 | 1.10 | 1817 | Mixed | Lightning |
| 34 | Jackson | 202 | 1.00 | 2969 | Black Spruce | Lightning |
| 35 | Dulbi River | 404 | 0.95 | 22057 | Black Spruce | Lightning |

**Table 3.** All fires that contributed to at least 1% of the total CO anomaly observed at CRV tower ordered by largest CO contribution. The distance column represents the distance of the center of the fire perimeter to CRV tower. Contribution is the percent contribution to the total integral of fire CO at CRV for the entire 2015 fire season. Some fires were grouped together if they were inside the same 0.5° grid cell during model coupling. For those cases, individual fire contribution to the CO anomaly observed at CRV tower was weighted based on fire size.

**Page 45: [1] Formatted  Wiggins, Elizabeth B. (LARC–E3)[UNIVERSITIES SPACE RESEARCH ASSOCIATION]**                    **8/11/20 10:42:00 AM**

Not Highlight

**Page 45: [1] Formatted  Wiggins, Elizabeth B. (LARC–E3)[UNIVERSITIES SPACE RESEARCH ASSOCIATION]**                    **8/11/20 10:42:00 AM**

Not Highlight

**Page 45: [1] Formatted  Wiggins, Elizabeth B. (LARC–E3)[UNIVERSITIES SPACE RESEARCH ASSOCIATION]**                    **8/11/20 10:42:00 AM**

Not Highlight

**Page 45: [1] Formatted  Wiggins, Elizabeth B. (LARC–E3)[UNIVERSITIES SPACE RESEARCH ASSOCIATION]**                    **8/11/20 10:42:00 AM**

Not Highlight

**Page 45: [1] Formatted  Wiggins, Elizabeth B. (LARC–E3)[UNIVERSITIES SPACE RESEARCH ASSOCIATION]**                    **8/11/20 10:42:00 AM**

Not Highlight

**Page 45: [1] Formatted  Wiggins, Elizabeth B. (LARC–E3)[UNIVERSITIES SPACE RESEARCH ASSOCIATION]**                    **8/11/20 10:42:00 AM**

Not Highlight

**Page 49: [2] Deleted  Wiggins, Elizabeth B. (LARC–E3)[UNIVERSITIES SPACE RESEARCH ASSOCIATION]**                    **8/11/20 10:44:00 AM**

**Page 51: [3] Deleted  Wiggins, Elizabeth B. (LARC–E3)[UNIVERSITIES SPACE RESEARCH ASSOCIATION]**                    **8/11/20 10:44:00 AM**

---

## Referee Report (RR1)

Review 2 of Wiggins et al by Bob Yokelson

The authors monitored three stable trace gases ($CO_2$, $CO$, and $CH_4$) that were emitted by fires located upwind of a tower in Alaska. They derived emission ratios and emission factors for two of the gases (not sure why $EFCO_2$ was not reported?). The study sampled smoke, when present, 24/7 for a whole fire season so it has a big effective sample size compared to individual past studies. It was also sensitive to examples of much, if not most, of the lifecycle of the upwind fires with exceptions including e.g. intense combustion episodes that lead to free-troposphere injection and long-range transport. In theory, the most important use of this tower data is to test model predictions of smoke production and transport for the stable species measured. This is discussed a little and could be very valuable for future model evaluation in other papers. The work is new, very valuable, and should definitely be published with minor revisions as summarized next and also pointed out in the specific comments.

The study has some weaknesses, which need to be recognized in a more balanced discussion. In no particular order:

1. Towers can only monitor upwind fires limiting the range of sampling.

2. Any ground-based site may have some bias to smoldering or miss the type of emissions subject to long-range transport in the free troposphere. This is a difficult topic to achieve certainty on.

3. The uncertainty in the background at the tower is pretty large compared to the observed enhancements (in 2015) when far downwind and so the tower-based approach may only work in near-record fire years whose representativeness is unknown.

4. The initial emissions can only be measured for a few stable species but the vast majority of interesting fire products are reactive.

The current discussion is written as if the authors discovered potential sampling biases specific to geographic regions and platforms that have already been major concerns in mainstream thinking for decades. At the same time, they fail to emphasize the exciting finding, which is that past attempts to overcome the limitations of any one sampling platform appear to have worked pretty well according to the perspective provided by this novel, unique study. In other words, past compilations averaged together the results from multiple platforms in an attempt to overcome the limitations of using just airborne, ground-based, or lab data. The results in these compilations are virtually indistinguishable from the authors results for the two species they report, which is pretty remarkable. It inspires more confidence in the previous recommendations for countless other species reported in those compilations, which is good news from a fresh perspective. The authors miss the mark by instead dwelling on air/ground differences, which are worth pointing out, but were already well-known. I think the authors deserve credit for recognizing the unique opportunity they had to evaluate past recommendations, but mistakenly focus their discussion on the limitations of a subset of previous work. The value of validating previous recommendations is huge because past work was actually vastly more complete chemically and probed many other fire seasons and geographic areas. Imagine the millions of dollars it would cost to outfit a tower

with instrumentation similar to that on the NASA DC-8 for just one summer and then maybe have a year like 2012 with no smoke or only downwind fires!

In addition, the study makes speculative, unsupported tangential claims about the particles from boreal fires despite the lack of any PM data. Despite validating previous recommendations, it is guessed that EFPM, and therefore health and climate effects, might be underestimated in models. However, the authors a) did not sample PM, b) may not have sampled the type of combustion that leads to long-range transport and wider impacts, c) did not consider secondary aerosol processes such as evaporation (see detailed comments), and d) fail to recognize that a PM network is in place that constrains the amount of PM in populated areas.

A brief warning, compared to other journals, ACP has pretty lax quality control and rarely sends papers back to the Referees for a second look. Thus the authors will be well advised to proofread future versions more carefully. There are typos that could be recycled or should have been caught by a spell-checker that I note along with other specific comments below by page and line number.

Specific comments format is page, line number: "comment"

1, 18: example typo, see page 6, line 35 EFCH4 is 5.3+/-1.8

1,22-24: How does smoke age impact sampling times? I.e. can't you measure 24/7 from anywhere?

1, 24: high compared to what? not recommendations. How does "variable" inform a comparison? delete "high and variable"

1, 25: more prominent than what? Keep "prominent", delete "more", "continuously" > "continuous"

1, 26: change "typical" to "a range of" since 2015 not a typical year according to authors.

1, 29: could add albedo and aerosol for completeness of overview here

1, 32 – 2, 2 – 2, 7: Exactly, but these "many" fires are forgotten about in the rest of the paper as it stands now.

2, 9: delete "future"

2, 11: delete "feedbacks"

2, 13: add "emissions of" before "specific" or it makes no sense.

2, 19: "have sometimes been" … Recommendations from Andreae weight all studies included equally, but the Akagi recommendations often consider amount of sampling, representativeness, quality of technique, etc. in recommendation as explained for each fire type in Sect 2. Users are encouraged to change the averaging formulas in the supplemental tables if justified for their application.

2, 22: "near and within" or "through" or "across"

2, 23: not just IR and WAS, other instruments include diode lasers, mass spec, and many others too, especially in ARCTAS.

2, 24: I'm not checking this number of fires, but note past work coves a variety of places and years, which is good.

2, 22-30: This is a nice overview of limitations of aircraft sampling, but equal attention is needed on limitations of fixed surface sites as noted in general comment.

2, 31: I would change "surface tower" to "fixed surface site" to make it more general and include the work by Collier, Gilman, Selimovic et al cited just below. Selimovic et al., 2019a is now just "2019" and "2019b" is now "2020."

Selimovic, V., Yokelson, R. J., McMeeking, G. R., and Coefield, S.: Aerosol mass and optical properties, smoke influence on $O_3$, and high $NO_3$ production rates in a western US city impacted by wildfires, J. Geophys. Res., 125, e2020JD032791, 2020.

2, 34: delete "]."

2, 37: add "of" before "smoldering"

2, 40: "fromfrom"

3, 3: fyi, smoldering converts solid biomass to gases, flaming oxidizes some of those gases. Yokelson et al., 1996, 1997

3, 6-7: Actually no way to have an open fire with low oxygen so delete "in a high oxygen environment."

3, 3 – 3, 13 and 3, 14 – 3, - 21: good overviews

3, 24: 5,858,000 30 s samples would be almost 3 million minutes, >48,000 hours, or >2000 days all within a ~90 day period! 58,000 samples is only 20 days….?

3, 26-28: "Analysis of these data indicate that smoldering processes may have a higher contribution to total wildfire emissions from North American boreal forests than previous estimates derived from aircraft measurements." Out of place as a result in the intro and also comes across as a random change of subject.

4,4 move sentence till after next on or rephrase as "… data stream we used …"

4,1-17: Take a few sentences to explain the data collection and analysis better and refer to tables. Clarify the following:

1) If you shifted to make continuous data, the time base would get further and further off or have jumps making it harder to compare to model?

2) The instrument sampled for 30 s then did something else for "<15s" then repeated until 50 minutes was up?

3) If 30 or more of the 30 s samples within one 50 min interval each had CO > 0.5 ppm the series was denoted as an emission factor event as shown in tables?

4) elevated CO for less than 30 of the 30s samples was ignored?

5) no emission factor events were allowed to span two different 50 min intervals?

6) How does the sample size criteria impact continuity?

More important than justifying any choice as the best choice is to explain once clearly what was done in section 2.1, how the instrument sampled and how data was reduced and tie that explanation to the Tables – making sure tables are called out in right order.

4, 21: Correlation among these species occurs for all combustion, including traffic in Fairbanks, but hopefully low anthropogenic influence at tower.

4, 23-24: So assumed a flat background for CO and CH4 for the whole summer regardless of wind direction, etc. rather than fitting a baseline from before to after each peak? Aren't ecosystem CH4 fluxes potentially variable?

4, 24-26: So the model reproduced 2012 when few fires occurred and then was run with 2015 input to get a 2015 calculated background?

4, 34-36: Even if the calculated background level changes slowly it could be the wrong level. Fractional uncertainty in the fire excess CO2 is roughly the uncertainty in the background (~3 ppm from Fig. 2a) divided by the size of the enhancement (~15 ppm from Fig 2b) for about 20% uncertainty on average? Or, if you just want one ER for the whole season you could just integrate the excess over the whole summer or do regression on the whole summer and get uncertainty from the uncertainty in the slope. Computing integrals for the whole summer might be a step closer toward a flux-based EF? Could be interesting to see how the result of that approach differs?

4, 37: did you get a slope for each 30 data-point+ "interval" and are "intervals" individual peaks or could they be partial or multiple peaks? Are intervals typically associated mainly with one fire?

5, 1-9: I did not check formulas, but got same EF results for CO and CH4 from reported ER so probably no typos? Also, why not report EFCO2?

5, 17: "the sampled combustion processes"

5, 24-26: Varying plume injection height within the boundary layer may not impact result at tower a lot if PBL well-mixed, but it excludes injection into the free troposphere during intense combustions and arguably would reduce the importance of long range transport, which is highlighted in the intro and conclusions.

5, 28: "isis" hacked your paper:)

5, 33: It's not dark yet at 6 pm in summer in AK? But with this definition, 10% at night seems low, is there GOES FRP to back that up?

5, 35: "83% of detected fire activity"

6, 2: "roughly consistent" i.e. almost a factor of two different

6, 3-13: Nice modeling application here. Were the individual fire contributions too mixed-fire events at the tower computed on a whole season daily, hourly, or interval basis? Some large fires may not have grown much on the day they impacted the tower?

6, 14: units are not immediately understandable, maybe explain in a bit more detail?

6, 18 & 20: Useful to define emission factor "event" or "period" earlier when describing how data stream analyzed?

6, 20-25: This is a cool analysis and useful that likely represents a lot of work! Not a criticism, but the finding that 27% of smoke impacting the tower was emitted at night, but the model assumes 10% of total AK smoke was emitted at night kind of shows the difficulty in proving representative sampling. Or what else does it mean? One general philosophy for dealing with this quandary has traditionally been to sample in multiple ways and synthesize the results; and simultaneously take the differences between approaches as a rough estimate of overall uncertainty. This is sort of what happens when using a literature average/stdev, while I acknowledge weighted averages can be better than straight averages in some cases.

6, 29: Are these 55 events the same as the EF events or periods? Are they all < ~50 minutes long? If the CO rose for two 50 minute periods and then fell for two 50 minute periods, is that one peak an emission factor event or is it 4 events? The data reduction can easily be spelled out clearly at the outset for folks that did not do the calculations and might wonder. Has the table of events been called out yet?

6, 29-30: The definition of an event earlier was lasting ~900 or more s? Here all the events lasted 50 minutes? So each hourly measurement interval with high enough CO was an event? I think it might be easy to take a few sentences above to just spell out how data was analyzed. Then I look at Table 2, are these the events and is N the number of 30 s increments? Maybe explain that earlier and include if each of these events is separated by a clean period?

6, 31: it would be interesting to see range in CH4/CO also in this sentence.

7, 8: "within" should be "with"? Table 3 called out by mistake? Also on line 16?

7, 20: diddid

7, 21, 22, 23: Variability < 5% probably not significant. Were events actually different fires? What is meant by flux-weighted estimates? Accounting for fuel consumption rate in a weighted average EF or windspeed at tower? The highest flux periods at the fire may produce high injection altitudes.

7, 25-27: Figure 7 shows some big peaks at tower, but not in model (doy ~188) or modeled peaks not seen at tower. The text says the model confirms elevated CO was primarily from fires. So I guess "primarily" signals > 50% and signals rough agreement? The authors stand by the unmodeled peaks being due to fires? How was it possible to get the fires contributing to the signal at the tower when the model did not capture a peak?

7, 28: "likelycaused"

7, 34: average distance weighted by fractional contribution?

7, 36: What is meant by "integrate emissions from multiple fires through the full planetary boundary layer"?

7, 39: > 8% in Table 3

8, 1: delete "significantly"

8, 3: 4646%

8, 8: Andreae-associated recommendations averaged the values from studies using different platforms partly in recognition of bias being possible for any one platform. Akagi et al pioneered splitting extratropical forests into boreal and temperate. They (Sect 2.3.2) actually used a pretty complex scheme averaging smoldering fuels from lab studies by fuel type rather than by study to get a ground-based average, which was then averaged with airborne results for an overall average roughly consistent with about 70% of overall fuel consumption by smoldering. They mentioned evidence that smoldering might be even more important. They devised formulas to estimate compounds measured only in lab or air and invited users to modify any of the formulas in their Table S2 if they preferred. Remarkably, their default recommendations are almost indistinguishable from this work. Regarding "important" differences on P8, L15, keep in mind that modelers determine the level of detail that works for them and it often involves model domain, scope of study, availability, reliability, and complexity of operational input, but also completeness, i.e. they need ERs/EFs for more than 2 species!

8, 13 re Table 1: Good idea to parse out data by location and platform and nice overview of data collected. Note Yokelson et al 1997 is missing (used in Akagi Table S2). Boreal peat was burned in Stockwell et al., 2015. Double check if Siberian fires were wild or prescribed, I think at least some were prescribed. Split Siberian fires out by air or ground? Siberian average row has possibly wrong total? Remove line numbers from number of fires column, "McMeeking has two capital "M"s, etc…

Stockwell, C. E., Veres, P. R., Williams, J., and Yokelson, R. J.: Characterization of biomass burning emissions from cooking fires, peat, crop residue, and other fuels with high-resolution proton-transfer-reaction time-of-flight mass spectrometry, Atmos. Chem. Phys., 15, 845-865, doi:10.5194/acp-15-845-2015, 2015.

Yokelson, R.J., D.E. Ward, R.A. Susott, J. Reardon, and D.W.T. Griffith, Emissions from smoldering combustion of biomass measured by open-path Fourier transform infrared spectroscopy, J. Geophys. Res., 102, 18865-18877, 1997.

8, 15: "measurement technique" should be "sampling strategy" to be consistent and precise?

8, 17-36: The overview of air versus ground sampling of sources is pretty good, a little disorganized but all the most important points emerge clearly! A few points to add could be: Aircraft can replicate tower-based sampling with downwind vertical profiles, but not on a continuous basis like a tower. Also, any aircraft bias toward flaming combustion may actually be partly okay if it weights the EF towards times of higher fuel consumption, relevant to author's desire for flux-based EFs? Flaming always entrains some smoldering, and the entrainment footprint is larger with more intense flaming. Best not to oversimplify a complicated situation.

8, 29: "weak or non-existent" convection columns (aka "updraft cores"). Mostly true for fresh RSC emissions, so "usually" is a good qualifier since some RSC may get to aircraft altitude by non-fire uplift or be sampled in rare missed approaches.

8, 35: "yet rarely" is okay – a fresh RSC sample would require "a really good drill on the front of the plane" to quote a DC-8 pilot.

9, 4: "combustion of"

9, 4-5: Organic soils were focus of lab study of Yokelson et al., 1997 and included in Stockwell et al., 2015 during FLAME-4.

9, 7: "should" > "could" or "might" (see above on models)

9, 8: At least five lab studies burned boreal fuels, the $CO/CO_2$ ratios for FLAME-4 for black spruce and boreal peat are in supplement of Stockwell et al., 2015. Listing what fuels were included in averages in Table 1 would be helpful.

9,12 "McMeeking"

9, 18-29: The claim of different ERs is not strongly supported. The quoted (Table 1), purely surface-based sampling of Siberian fires had *lower* $CO/CO_2$ than the authors NA work, and, even more remarkably, only about half the $CO/CO_2$ ratio as the studies that included some airborne sampling of Siberian fires. So maybe better to say, the ecosystems differ and the emissions might as well, but not enough data to know yet.

Also work on the Siberia/NA differences goes back to at least 1993 when the Bor Island Experiment was started. Differences in Siberian and North American boreal fires were noted in publications 20-24 years ago with hundreds of references cited and a more recent review on that:

Goldammer, J.G., and V.V. Furyaev. 1996. Fire in ecosystems of boreal Eurasia. Ecological impacts and links to the global system. In: Fire in ecosystems of boreal Eurasia (J.G. Goldammer and V.V. Furyaev, eds.), 1-20. Kluwer Academic Publ., Dordrecht, 528 pp. https://link.springer.com/chapter/10.1007%2F978-94-015-8737-2_1

E.S.Kasischke and B.J.Stocks, eds. 2000. Fire, climate change, and carbon cycling in the boreal forest. Ecological Studies 138, Springer-Verlag, Berlin-Heidelberg-New York, 461 p.

Goldammer, J.G. (ed.) 2013. Prescribed Burning in Russia and Neighbouring Temperate-Boreal Eurasia. A publication of the Global Fire Monitoring Center (GFMC). Kessel Publishing House, 326 p. (ISBN 978-3-941300-71-2). http://www.forestrybooks.com/

9, 23: "hotter" okay, but there is no single temperature that defines any landscape fire, more aggressive flaming is probably what is meant.

9, 30: "Stronger" than what? Not a complete thought. Here the work goes off on a random tangent rehashing a long-recognized issue. Concerns about air/ground bias are discussed in Andreae and Merlet, 2001, which supports this with the following citation:

Andreae, M. O., E. Atlas, H. Cachier, W. R. Cofer, III, G. W. Harris, G. Helas, R. Koppmann, J.-P. Lacaux, and D. E. Ward, Trace gas and aerosol emissions from savanna fires, in Biomass Burning and Global Change, edited by J. S. Levine, pp. 278 – 295, MIT Press, Cambridge, Mass., 1996.

Previous recommendations by Akagi and Andreae appear to have compensated adequately for this issue according to this studies results to the extent that we are ever likely to know. The authors could claim that they have investigated the extent of platform-based bias in additional detail and present a useful contribution in that way, but the issue of the existence of differences is not a new finding.

Perhaps an appropriate header is: "A detailed examination of tower versus airborne sampling". Either include or don't include the enigmatic data from Siberia and make a new, useful point if you can, perhaps: a) mean difference is "X", or b) surprisingly no conclusion.

9, 38 – 10, 1: The authors have good evidence that tower-based platforms see more smoldering that the aircraft studies to date (in NA) and that is useful, but you don't know for sure if the tower might under-estimate flaming or why the Siberian data is enigmatic. And "previous reports" should be changed to "the average of previous airborne studies" since "previous reports" could imply all studies.

10,1-2: at a minimum change to "some previous" , "some flaming" , "some residual"

10, 1 – 8: Showing that the tower and aircraft got different overall average CO/CO2 ratios is straightforward and useful. But both platforms could have some error so proving that 100% of the error in representativeness is with the aircraft is not really doable. Every fire that impacted the tower also, undoubtedly produced some emissions that did not impact the tower due to wind shifts, altitude, or whatever, it's just basic common sense. The most exciting thing about this work is not even stressed. That is, by measuring downwind of many fires burning at all stages of their life cycle around-the clock, the authors have created a high-quality data set for evaluating fire emissions models performance at a regional level (as in Selimovic et al., 2019; 2020). I.e. AKFED did pretty well integrating the effects of many fires around the clock and predicting the tower "point CO" specifically. What would need to be changed in AKFED/STILT to improve performance could be a great follow-on study along with how do larger-scale models such as GFED, GFAS, FINN, etc., perform against the tower observations! Regardless of the "real, unknown total fire emissions," the signal at the tower is well-measured now and very useful to test models!

10, 9: This next paragraph repeats some of the material in the previous paragraph. If the text survives editing, change "crown" to "surface fuels" since the NW Territory crown fire experiment found that often the fires propagate in surface fuels followed by torching

10, 10: not a sentence, delete "that" to fix?

10, 11: substantial contributions of RSC were stressed by Bertschi et al 2003 in their Table 3.

10, 12: not only FRP-based, but any thermal signal. Smoldering involves temperature high enough to saturate the 3.9 micron channel if widespread enough, but obstruction is more of an issue for deep smoldering.

10,16: "previously thought" by who? If you mean the studies mentioned directly above, no comparison is possible unless you also have the relative consumption of above- and below-ground fuels. If you mean a previous compilation, Akagi et al estimated 70% of all boreal fuel was consumed by smoldering based on an MCE similar to this work.

10, 18: "most" is not "all" and MISR only looks at 1030 AM long before both the most intense combustion and diurnal cycle fuel consumption peak. I would change "most" to "many" or "some"

10, 19: "length scale"?

10, 20 – 21: good point there is time for vertical mixing, if the atmosphere is not too stratified.

10, 22-25: This sentence is not accurate as Collier et al sampled smoke up to 48 hours old and Selimovic et al 2019; 2020 sampled smoke from fires in the range 20-800 km upwind.

10, 24: if text survives "band" > "and"

10, 26-35: This discussion is oversimplified since dry weather can make larger fuels that tend to smolder more likely to burn. See section 2.4 in Akagi et al and the papers referenced therein by Hoffa et al., 1999, Shea et al., 1996, Kauffman et al., 1998; 2003, etc. Hot, dry weather can increase smoldering. Note also that most airborne studies occurred in years that were arguably more "typical".

10, 40: I don't think any new ideas "emerged", but the authors work can help continue to evaluate some long-recognized issues and maybe help reduce uncertainty.

11, 1: it always makes sense to "report" what you measured and studies of Siberian fires report their location. "Using" regionally-specific EFs might make sense, but is a separate decision for the modelers that is hard because few measurements have occurred in Russia where research access is super-problematic. A colleague had their canisters confiscated by the Russian military and "filled for them at undisclosed locations."

11, 5 – 11: This is not that big a deal, the complex averaging scheme of Akagi gave almost the same answer as this study or the simple averaging scheme of Andreae. Adding this studies extensive results in a weighted or simple average to the "evolving literature average" is BAU and

will have little impact on the average; though it is important to be clear about how things are synthesized.

11, 12 - 18: It is not clear what is meant by flux-weighted EF? Aircraft measurements may in fact be weighted towards times with high fuel consumption rates. Fires can produce multiple plumes. A flux of emissions in models results from the fuel consumption rate assumptions.

11, 18-23: This is just stating obvious that if we could measure everything, we'd know more. I would very strongly recommend deleting sections 4.3 (and 4.4) and instead have a section to highlight the exciting model evaluation now possible with what you *already* measured.

11, 24: "larger" than what?

4.4 is all speculation about PM, which was not even measured and the section doesn't consider SOA or PM evaporation where the latter was significant in Selimovic et al., 2019, 2020, and references there-in. Also, health impacts are based on measured PM and this study does not suggest the regional PM networks are inaccurate.

11, 27: higher than some studies doesn't equal higher than "previously thought".

11, 28 – 29: Long range transported smoke was not sampled in this study and that type of smoke may actually be better sampled from aircraft.

11, 25 – 36: delete, all speculation, not a topic or result of this paper.

12, 3: after "Our results" I would delete the rest and fill in the valuable insights that you actually learned about the AKFED model. I.e. it underestimated nighttime combustion impacts at the tower, it captured X of the Y peaks, seasonal average CO was within Z%, etc… Highlight potential for additional, future model evaluation and improvement.

---

## Author Response (AR2)

**Response to Reviewer #1 Comments (Bob Yokelson)**
The authors monitored three stable trace gases (CO2, CO, and CH4) that were emitted by fires located upwind of a tower in Alaska. They derived emission ratios and emission factors for two of the gases (not sure why EFCO2 was not reported?). The study sampled smoke, when present, 24/7 for a whole fire season so it has a big effective sample size compared to individual past studies. It was also sensitive to examples of much, if not most, of the lifecycle of the upwind fires with exceptions including e.g. intense combustion episodes that lead to free-troposphere injection and long-range transport. In theory, the most important use of this tower data is to test model predictions of smoke production and transport for the stable species measured. This is discussed a little and could be very valuable for future model evaluation in other papers. The work is new, very valuable, and should definitely be published with minor revisions as summarized next and also pointed out in the specific comments.

**Response:** We are grateful to Dr. Yokelson for providing additional detailed and valuable feedback on our manuscript. His suggestions have considerably improved the paper. We now mention in the conclusions the potential value of the tower data to test model predictions of smoke emissions and transport for CO, $CO_2$, and $CH_4$.

**General Comments**
The study has some weaknesses, which need to be recognized in a more balanced discussion. In no particular order:

**Response:** We completely revised the discussion to address these reviewer comments.

1. Towers can only monitor upwind fires limiting the range of sampling.

   We now include a paragraph in the discussion discussing the limits of ground-based sampling with towers. We make this point in that new paragraph.

2. Any ground-based site may have some bias to smoldering or miss the type of emissions subject to long-range transport in the free troposphere. This is a difficult topic to achieve certainty on.

   We agree and recognize this in the discussion paragraph by describing the limits to ground based sampling. We also acknowledge this in the revised conclusions. In the revised discussion, we also provide arguments that the CRV tower is not highly sensitive to this type of bias, because it is at a higher elevation than most of the fires and far downwind. We also now make the point to the reader that analysis of MISR satellite observation suggest most (but not all) fire plumes reside within the PBL in boreal North America, again suggesting the CRV tower measurements can provide representative estimates.

3. The uncertainty in the background at the tower is pretty large compared to the observed enhancements (in 2015) when far downwind and so the tower-based approach may only work in near-record fire years whose representativeness is unknown.

   We agree and make this point now in the discussion paragraph describing the limits of ground-based sampling.

4. The initial emissions can only be measured for a few stable species but the vast majority of interesting fire products are reactive.

We agree and make this point now in the first paragraph in the discussion (final sentence).

In summary, we added the following paragraph in the discussion to address many of these reviewer concerns:

"In the context of these comparisons among ecoregions and sampling strategies, it is important to recognize that tower-based sampling strategies, including the methodology presented in this study, have important limits. Ground-based sites may potentially miss some of the emissions injected above the planetary boundary layer which are subject to long-range transport in the free troposphere. The fixed nature of this sampling technique also restricts the range of sampling, because towers can only monitor upwind fires. Although the tower-based sampling strategy allows for integration of emissions from fires across a range of environmental conditions and at different stages of fire life cycles, it does not allow for emission ratio measurements of non-conserved species, including particulate matter and many fire-emitted volatile organic compounds that have short lifetimes. The technique is also subject to higher uncertainty in the definition of background mole fractions for fire-affected trace gases, because of the dilution and mixing of fire emissions that occurs during transport, and thus may not be a feasible sampling methodology during years with low fire activity. "

The current discussion is written as if the authors discovered potential sampling biases specific to geographic regions and platforms that have already been major concerns in mainstream thinking for decades. At the same time, they fail to emphasize the exciting finding, which is that past attempts to overcome the limitations of any one sampling platform appear to have worked pretty well according to the perspective provided by this novel, unique study. In other words, past compilations averaged together the results from multiple platforms in an attempt to overcome the limitations of using just airborne, ground-based, or lab data. The results in these compilations are virtually indistinguishable from the authors results for the two species they report, which is pretty remarkable. It inspires more confidence in the previous recommendations for countless other species reported in those compilations, which is good news from a fresh perspective.

**Response:** We have fully revised the discussion, carefully considering these reviewer points. The first paragraph of the revised discussion highlights the agreement of our measurements with past studies and the validation these measurements provide for non-conserved species that cannot be measured with a tower-based sampling approach.

The authors miss the mark by instead dwelling on air/ground differences, which are worth pointing out, but were already well-known. I think the authors deserve credit for recognizing the unique opportunity they had to evaluate past recommendations, but mistakenly focus their discussion on the limitations of a subset of previous work. The value of validating previous recommendations is huge because past work was actually vastly more complete chemically and probed many other fire seasons and geographic areas. Imagine the millions of dollars it would cost to outfit a tower with instrumentation similar to that on the NASA DC-8 for just one summer and then maybe have a year like 2012 with no smoke or only downwind fires!

**Response:** We have considerably revised our discussion with this reviewer concern in mind. Again, we note that we emphasize the agreement between our measurements and the mean reported in past syntheses in the first paragraph of our discussion. We specifically note the point that this validation is important because it confirms estimates made for short-lived species that cannot be measured by a remote tower-sampling approach.

However, we have not previously seen a breakdown and synthesis of ground-based versus aircraft-based sampling approaches for northern boreal forests. While we make it clear in the revised text that its been well appreciated in the literature for quite some time that aircraft-based and ground-based sampling approaches are known to yield different outcomes, *the magnitude of these differences and comparison with our new*

*measurements is a new finding* that we think is important for readers, and advances the field. We are more careful in our comparisons in the revised discussion, making it clear our measurements have a mean that is 39% higher than the mean solely derived from aircraft sampling in North America. We also show that emission ratios for Eurasian forests are quite a bit higher than those from North America. We also more forcefully make the case for why our remote tower-based sampling approach is likely to yield a more representative estimate of emission ratios than one might expect in other places.

In addition, the study makes speculative, unsupported tangential claims about the particles from boreal fires despite the lack of any PM data. Despite validating previous recommendations, it is guessed that EFPM, and therefore health and climate effects, might be underestimated in models. However, the authors a) did not sample PM, b) may not have sampled the type of combustion that leads to long-range transport and wider impacts, c) did not consider secondary aerosol processes such as evaporation (see detailed comments), and d) fail to recognize that a PM network is in place that constrains the amount of PM in populated areas.

**Response:** We have removed this paragraph and discussion of implications for PM and organic aerosols, following the reviewer's suggestion.

A brief warning, compared to other journals, ACP has pretty lax quality control and rarely sends papers back to the Referees for a second look. Thus the authors will be well advised to proofread future versions more carefully. There are typos that could be recycled or should have been caught by a spell-checker that I note along with other specific comments below by page and line number.

**Response:** We have carefully revised the manuscript following the reviewers detailed suggestions below, making changes for most (but not all) of the reviewer's specific comments. We have carefully spell-checked and proof read the revised manuscript.

**Specific Comments** (format is page, line number: "comment")

1, 18: example typo, see page 6, line 35 EFCH4 is 5.3+/-11.8

**Response:** We apologize for the typo and have corrected it. The mean and standard deviation for the CH4 emission factor should be $5.3 \pm 1.8$.

1,22-24: How does smoke age impact sampling times? I.e. can't you measure 24/7 from anywhere?

**Response:** We modified the sentence to make it clear we are describing the transit times between combustion within a fire perimeter and downwind measurement at the tower. We describe carefully in the main text what we mean by transit times. The new sentence reads: "The model also indicated that typical mean transit times between trace gas emission within a fire perimeter and tower measurement were 1-3 days, indicating that the time series sampled combustion across day and night burning phases (Figure 3)."

1, 24: high compared to what? not recommendations. How does "variable" inform a comparison? delete "high and variable"

**Response:** We deleted "variable" from the sentence. We retained "high" because this is a major point of our analysis and paper, that emission factors from our tower observations are higher than the mean of past aircraft sampling from boreal North America.

1, 25: more prominent than what? Keep "prominent", delete "more", "continuously" > "continuous"

**Response:** We agree with the reviewer and have removed "more" and changed "continuously" to "continuous." The new sentence reads: "The high CO emission ratio estimates reported here provide evidence

for a prominent role of smoldering combustion, and illustrate the importance of continuously sampling fires across time-varying environmental conditions that are representative of a range of fire season."

1, 26: change "typical" to "a range of" since 2015 not a typical year according to authors.

**Response:** As noted in the response above, we changed "typical" to "a range of".

1, 29: could add albedo and aerosol for completeness of overview here

**Response:** We added aerosols to this first sentence, following the reviewer's suggestion. The Johnson book we cite is a classic and we wanted to open the paper with this reference. However, this book does not describe the complex relationship between boreal forest fires and planetary albedo changes (Randerson et al., 2006), so we did not add albedo to the overview.

1, 32 – 2, 2 – 2, 7: Exactly, but these "many" fires are forgotten about in the rest of the paper as it stands now.

**Response:** We respectfully disagree with the reviewer on this point. We are directly reporting on an extreme wildfire season in Alaska. It is in our title. Trace gases and aerosols from the very large complex of 2015 wildfires did get transported widely across the North American continent, in a way that is similar to the examples we provide of other fire events in the introduction.

2, 9: delete "future"

**Response:** We deleted "future" from in front of projections. Thank you.

2, 11: delete "feedbacks"

**Response:** We deleted "feedbacks" following the reviewer suggestion.

2, 13: add "emissions of" before "specific" or it makes no sense.

**Response:** We added "emissions of" before "specific." The sentence now reads: "Emission factors provide a straightforward way to convert fire consumption of dry biomass into emissions of specific trace gas species, such as CO, $CH_4$, and $CO_2$."

2, 19: "have sometimes been" … Recommendations from Andreae weight all studies included equally, but the Akagi recommendations often consider amount of sampling, representativeness, quality of technique, etc. in recommendation as explained for each fire type in Sect 2. Users are encouraged to change the averaging formulas in the supplemental tables if justified for their application.

**Response:** We added "sometimes" in the place recommended by the reviewer.

2, 22: "near and within" or "through" or "across"

**Response:** We changed the sentence to "...fly aircraft through plumes."

2, 23: not just IR and WAS, other instruments include diode lasers, mass spec, and many others too, especially in ARCTAS.

**Response:** We removed "infrared" following the reviewer's suggestion.

2, 24: I'm not checking this number of fires, but note past work coves a variety of places and years, which is good.

**Response:** We believe this number represents all fires measured in previous studies, and agree with the reviewer that the synthesis should cover a representative sampling of location and years.

2, 22-30: This is a nice overview of limitations of aircraft sampling, but equal attention is needed on limitations of fixed surface sites as noted in general comment.

**Response:** We added a sentence to the end of this paragraph to briefly describe limits to surface sampling. The final sentence of this paragraph reads: "Surface sampling near or within fire perimeters may have an advantage with respect to providing measurements during intervals when aircraft are unable to fly, but are also more likely to under sample emissions injected above the boundary layer by fire plumes and within pyro-cumulus clouds."

2, 31: I would change "surface tower" to "fixed surface site" to make it more general and include the work by Collier, Gilman, Selimovic et al cited just below. Selimovic et al., 2019a is now just "2019" and "2019b" is now "2020."

**Response:** We changed "surface tower" to "fixed surface site" and modified the references as suggested.

Selimovic, V., Yokelson, R. J., McMeeking, G. R., and Coefield, S.: Aerosol mass and optical properties, smoke influence on $O_3$, and high $NO_3$ production rates in a western US city impacted by wildfires, J. Geophys. Res., 125, e2020JD032791, 2020.

2, 34: delete "]."

**Response:** We removed the typo.

2, 37: add "of" before "smoldering"

**Response:** We added "of" before "smoldering"

2, 40: "fromfrom"

**Response**: We removed the typo following the reviewer suggestion.

3, 3: fyi, smoldering converts solid biomass to gases, flaming oxidizes some of those gases. Yokelson et al., 1996, 1997

**Response:** We changed the sentence to "Smoldering combustion converts solid biomass to gases and aerosols, while flaming oxidizes some emissions [*Yokelson et al.*, 1996, 1997]."

3, 6-7: Actually no way to have an open fire with low oxygen so delete "in a high oxygen environment."

**Response:** We deleted "in a high oxygen environment."

3, 3 – 3, 13 and 3, 14 – 3, - 21: good overviews.

**Response:** We appreciate the reviewer's comment that our text here in the introduction is clear summary of past work on this topic.

3, 24: 5,858,000 30 s samples would be almost 3 million minutes, >48,000 hours, or >2000 days all within a ~90 day period! 58,000 samples is only 20 days….?

**Response:** This is a typo. We updated the text to "59,800." The datastream from June 9 – August 13[th] (65 days based on figure 3) had 59824 individual 30s long samples. This excludes 13 mins out of every hour as the Picarro cycles through the lower levels (10 mins of sampling lower levels + 3 mins to flush the lines) and ~ 8 mins out of every 8 hours when the Picarro samples reference gases (5 mins) + 3 mins to flush. We had 4362 total individual 30s long samples used to calculate the emission ratios.

3, 26-28: "Analysis of these data indicate that smoldering processes may have a higher contribution to total wildfire emissions from North American boreal forests than previous estimates derived from aircraft measurements." Out of place as a result in the intro and also comes across as a random change of subject.

**Response:** We appreciate the reviewer's perspective, but wish to note there are many different possible stylistic approaches for writing the last paragraph in the introduction. It is not uncommon to provide the reader with an overview statement of a main finding at the end of the introduction, and in this context, we would respectfully request to keep this sentence in its present form, changing "measurements" to "sampling".

4,4 move sentence till after next one or rephrase as "… data stream we used …"

**Response:** We rephrased this sentence as "… data stream we used …"

4,1-17: Take a few sentences to explain the data collection and analysis better and refer to tables. Clarify the following:

1) If you shifted to make continuous data, the time base would get further and further off or have jumps making it harder to compare to model?

2) The instrument sampled for 30 s then did something else for "<15s" then repeated until 50 minutes was up?

3) If 30 or more of the 30 s samples within one 50 min interval each had CO > 0.5 ppm the series was denoted as an emission factor event as shown in tables?

4) elevated CO for less than 30 of the 30s samples was ignored?

5) no emission factor events were allowed to span two different 50 min intervals?

6) How does the sample size criteria impact continuity?

More important than justifying any choice as the best choice is to explain once clearly what was done in section 2.1, how the instrument sampled and how data was reduced and tie that explanation to the Tables – making sure tables are called out in right order.

**Response:** We considerably revised and clarified the sampling protocol of the spectrometer at CRV tower:

In section 2.1:
    "Atmospheric CO, $CH_4$, and $CO_2$ mole fractions were measured using a cavity ring-down spectrometer (CRDS, Picarro models 2401 and 2401m) [*Karion et al.*, 2016] at the CRV tower in Fox, Alaska (64.986°N, 147.598°W, ground elevation 611m above sea level). The tower is located about 20 km northeast of Fairbanks Alaska on top of a hill in hilly terrain (Figure 1), and within the interior lowland and upland forested ecoregion in interior Alaska [*Cooper et al.*, 2006]. There are three separate inlets on CRV tower at

different heights above ground level from which the spectrometer draws air for sampling. The spectrometer samples air from the highest level for 50 minutes out of every hour, and then draws air from the other two levels for 5 minutes at each level [Karion et al., 2016]. Standard reference gases are sampled every 8 hours for 5 minutes, and measurements are removed for a time equivalent to three flushing volumes of the line, approximately 3 minutes, after a level change or switch to or from a calibration tank. All raw 30 s average measurements were calibrated according to Karion et al. [2016].

We used observations from air drawn from the top intake height at a height of 32 m above ground level in our analysis because this level had the highest measurement density and the smallest sensitivity to local ecosystem $CO_2$ fluxes near the tower [Karion et al., 2016]. We used gaps in this time series, created when the spectrometer cycled to the lower inlets and following calibration, to separate the time series into discrete time intervals for the calculation of emission ratios. Each 30 s average measurement within a 47-minute sampling interval served as an individual point in our calculation of an emission ratio described below (Table 2)."

We also modified the text to better explain our data screening methodology. It now reads:

"We isolated intervals when fire had a dominant influence on trace gas variability observed at CRV to calculate emission ratios. An interval with dominant fire influence was defined as a continuous 47-minute measurement period that had: 1) a minimum of at least 30 trace gas measurements (with each measurement representing a mean over 30 seconds), 2) a mean CO over the entire interval exceeding 0.5 ppm, and 3) significant correlations between CO and $CO_2$, and between $CH_4$ and $CO_2$, with $r^2$ values for both relationships exceeding 0.80.

For each interval, we required a sample size of at least 30 individual 30 s measurements. For each interval meeting this criterion, we calculated the mean CO mole fraction and discarded intervals that had a mean CO less than 0.5 ppm. For each of the intervals with mean CO that exceeded the 0.5 ppm threshold, we then extracted the 30 s measurement time series of CO, $CH_4$, and $CO_2$ mole fractions and calculated correlation coefficients between the trace gas time series. Only intervals with high and significant correlations between CO and $CO_2$ and between $CH_4$ and $CO_2$ ($r^2 > 0.80$; $p < 0.01$, $n > 30$) were retained, because covariance among these co-emitted species is a typical signature of combustion [*Urbanski*, 2014]. Data from each of the intervals that met the three criteria described above were used to compute emission ratios, emission factors, and MCE. These intervals are reported in chronological order in Table 2."

4, 21: Correlation among these species occurs for all combustion, including traffic in Fairbanks, but hopefully low anthropogenic influence at tower.

**Response:** We agree that there can be a significant $CO:CO_2$ correlation generated from traffic, but the $CH_4$ levels emitted from this activity are quite small compared to fire emissions based on measurements we have made in Los Angeles and other cities [Hopkins et al., 2016], and so our requirement for a significant $CH_4:CO_2$ correlation reduces our sensitivity to an influence from this source. Especially since during summer, $CO_2$ fluxes from the terrestrial biosphere are large relative to anthropogenic emissions [Commane et al., 2017]. This site was selected to be 20 km outside of Fairbanks to provide a background station for interior Alaska [Karion et al., 2016]. Finally, in other work, we surveyed Fairbanks for methane leaks using a portable Picarro cavity ringdown spectrometer. The city does not have substantial natural gas infrastructure (and leaks), which was somewhat surprising to us. Thus, we believe our criteria of simultaneous high correlations between CO and $CO_2$ and between $CH_4$ and $CO_2$ are likely to screen out any periods with anthropogenic influence. We also note that our modeling analysis confirms fires were a dominant driver of CO variability at the Fox during the summer of 2015.

4, 23-24: So assumed a flat background for CO and CH4 for the whole summer regardless of wind direction, etc. rather than fitting a baseline from before to after each peak? Aren't ecosystem CH4 fluxes potentially variable?

**Response:** As shown in Figure 4, fires were burning continuously from DOY 165 through DOY 220. This made it impossible to fit a baseline before or after each 47-minute interval we used to compute an emission ratio. $CH_4$ levels in interior Alaska at this tower were more variable during the summer 2015 than in other years because of large fire source. As described below in response to the reviewer comment on page 4, 34-36, because we use a linear regression to compute a slope using up to 95 30-second points during each 47-minute interval, our approach is insensitive to background variability on longer timescales.

4, 24-26: So the model reproduced 2012 when few fires occurred and then was run with 2015 input to get a 2015 calculated background?

**Response:** That is correct. We changed the ordering of the text in this paragraph and added the following sentence to clarify: "After training on data from the summer of 2012, the model was then run using 2015 input variables to calculate time evolving $CO_2$ background mole fractions during our analysis period."

4, 34-36: Even if the calculated background level changes slowly it could be the wrong level. Fractional uncertainty in the fire excess CO2 is roughly the uncertainty in the background (~3 ppm from Fig. 2a) divided by the size of the enhancement (~15 ppm from Fig 2b) for about 20% uncertainty on average? Or, if you just want one ER for the whole season you could just integrate the excess over the whole summer or do regression on the whole summer and get uncertainty from the uncertainty in the slope. Computing integrals for the whole summer might be a step closer toward a flux-based EF? Could be interesting to see how the result of that approach differs?

**Response:** We have changed the text in the paragraph to clarify how we computed the emission ratio for each 50-minute measurement interval. Specifically, we first compute the excess mole fractions for CO (or $CH_4$) and for $CO_2$. We do this by removing the background value for CO (or $CH_4$) (this step removes the same value from each 30s mean observation within a measurement interval). We then remove the background level for $CO_2$, which evolves slowly over time during the 47-minute interval. Once we have the 30s time series of excess mole fractions, we then perform a linear regression with CO (or $CH_4$) molar excess serving as the y variable and $CO_2$ molar excess serving as the x variable. The slope of this linear regression is the emission ratio. In this context, a bias in the background subtracted from CO or $CH_4$ that remains the same over the sampling interval will have no effect on the slope of the regression line. An offset in the baseline will influence the intercept but not the slope.

So this approach is different from what might occur when the $CO_2$ excess mole fraction is computed using a background from an out-of-plume air sample from an aircraft. In this latter approach, a bias in the $CO_2$ background translates directly to a bias in the reported emission ratio. This is not the case for our approach because the regression line slope is derived from the covariation of CO and $CO_2$ within the measurement interval (the variability shown in Figure 7).

The new text reads:

"We estimated an emission ratio ($ER_X$; equation 1) by calculating the slope from a type II linear regression of CO or $CH_4$ excess mole fractions ($\Delta X$) relative to the $CO_2$ excess mole fraction ($\Delta CO_2$) using all of the 30 s observations available within a single 47-minute sampling interval when fire had a dominant influence on tower trace gas variability (up to 95 pairs of measurements). To estimate excess mole fractions (denoted with a $\Delta$), we first removed background mole fractions (described above) before performing the regression analysis and obtaining the slope. The assumed background levels for CO and $CH_4$ did not influence this

emission ratio estimate because they were assumed to remain constant throughout the duration of each 47-minute interval (i.e., they influenced the intercept but not the slope of the regression). In a sensitivity analysis we found that the removal of the $CO_2$ background, which did evolve within each 47-minute interval, had only a negligible effect, because the $CO_2$ background did not change rapidly over time."

4, 37: did you get a slope for each 30 data-point+ "interval" and are "intervals" individual peaks or could they be partial or multiple peaks? Are intervals typically associated mainly with one fire?

**Response:** To answer the first part of this reviewer comment, the answer is yes, we got a single emission ratio for each 47-minute interval (with at least 30 30-s samples) from the linear regression slope. We believe the trace gas variability within a single 47-minute measurement interval used to compute an emission ratio often contained a composition of emissions from multiple fires.

We added the following text to clarify that multiple fires can contribute to excess mole fractions during a single measurement interval.

"Since multiple fires were often burning simultaneously during the 2015 fire season, the emission ratios we report in Table 2 for each interval likely represent a composite of emissions from several fires."

5, 1-9: I did not check formulas, but got same EF results for CO and CH4 from reported ER so probably no typos? Also, why not report EFCO2?

**Response:** Thank you for checking the ER to EF step. We also doubled checked this using equation 2 and equation 3 and confirmed the numbers in Table 2. The emission factor for $CO_2$ is fundamentally different, having a high degree of sensitivity to the carbon content of fuels. Since we did not make any direct measurement of fuel consumption of different tree, litter, and surface duff pools (and their carbon content) we prefer not to report $CO_2$ emission factors. These, of course, can be computed directly from Table 2 for anyone who really needs this information.

5, 17: "the sampled combustion processes"

**Response:** We added "sampled" to this sentence following the reviewer suggestion.

5, 24-26: Varying plume injection height within the boundary layer may not impact result at tower a lot if PBL well-mixed, but it excludes injection into the free troposphere during intense combustions and arguably would reduce the importance of long range transport, which is highlighted in the intro and conclusions.

**Response:** We included more text in the introduction and in the discussion sections describing the limitation of using a stationary surface sampling location.

5, 28: "isis" hacked your paper:)

**Response:** We apologize for the typo and removed it.

5, 33: It's not dark yet at 6 pm in summer in AK? But with this definition, 10% at night seems low, is there GOES FRP to back that up?

**Response:** Correct, it's not dark at 6 pm or even 1 am at 64°N in late June, but the human eye is very sensitive to low light levels. Eddy covariance tower observations of the diurnal cycles of net radiation and sensible heat fluxes from interior Alaska collected by our group [Liu et al., 2005] show a very clear diurnal cycle and a very much reduced nighttime flux during summer (JJA). This is clearly shown in Figure 8 of that paper.

The collapse of the boundary layer at night, even in Alaska, lowers surface air temperatures and increases relative humidity levels, thus reducing fire activity.

We used FRP to support our partitioning, and we reported on this directly in the previous round of review (and integrated these results from MODIS fire radiative power into the current draft). GOES is not appropriate to use for several reasons: 1) at high northern latitudes with the very large pixel sizes (more than 15 km on a side), threshold fire sizes (and temperatures) for detection are considerable, and may change over the course of a diurnal cycle; 2) there is not a robust FRP product for the GOES-R time series yet.

5, 35: "83% of detected fire activity"

**Response:** We added "detected" to this sentence following the reviewer's suggestion.

6, 2: "roughly consistent" i.e. almost a factor of two different

**Response:** We added "broadly" to the sentence, following the reviewer's suggestion. The model parameterization is a 90:10 split of emissions between day and night intervals, whereas the integral of FRP from MODIS satellite observations suggests an 83:17% split. These are similar given uncertainties and incomplete diurnal coverage of the satellite data.

6, 3-13: Nice modeling application here. Were the individual fire contributions too mixed-fire events at the tower computed on a whole season daily, hourly, or interval basis? Some large fires may not have grown much on the day they impacted the tower?

**Response:** The individual fire contributions were calculated over the 2015 fire season. We modified the following sentence in the methods section to clarify: "The difference between the original model and the updated coupling was equal to an individual fire's contribution to CO at the CRV tower, when integrated over the 2015 fire season."

6, 14: units are not immediately understandable, maybe explain in a bit more detail?

**Response:** We explain the "footprints" in more detail in a later sentence that reads: "These functions provide an estimate of the impact of upwind surface fluxes at different times in the past on CRV tower trace gas mole fraction measurements at a given time."

We also included more information in our model description: "For this application, STILT [*Lin et al.*, 2007] was used to estimate the adjoint of PWRF [*Skamarock et al.*, 2005; *Chang et al.*, 2014; *Henderson et al.*, 2015] during the summer of 2015 at the location of the CRV tower, to generate surface influence functions that relate surface fluxes from Alaska to trace mole fractions at the CRV tower. These gridded influence functions are known as footprints and have units of mole fraction per unit of surface flux (ppm/($\mu$mol m$^{-2}$ s$^{-1}$))."

6, 18 & 20: Useful to define emission factor "event" or "period" earlier when describing how data stream analyzed?

**Response:** We have attempted to standardize our language in response to an earlier reviewer comment. Please see our response to comment 4,1-17. We now use the term "interval" to refer to the period of time over which we compute an emission factor. We modified the text here so it now reads: "We analyzed the footprints for each interval used to calculate emission factors to confirm…"

6, 20-25: This is a cool analysis and useful that likely represents a lot of work! Not a criticism, but the finding that 27% of smoke impacting the tower was emitted at night, but the model assumes 10% of total AK smoke was emitted at night kind of shows the difficulty in proving representative sampling. Or what else does it mean? One general philosophy for dealing with this quandary has traditionally been to sample in multiple ways and synthesize the results; and simultaneously take the differences between approaches as a rough estimate of overall uncertainty. This is sort of what happens when using a literature average/stdev, while I acknowledge weighted averages can be better than straight averages in some cases.

**Response:** We agree with the reviewer that it is important to report these numbers. We also agree it makes sense to combine information from different measurement approaches and models to further reduce uncertainties in emission factors. In this context, it is also important to consider differences in fire behavior and ecosystem type when creating a literature mean and std deviation, especially for use in global models. We return to this issue in the discussion and our response to reviewer comments in the discussion.

6, 29: Are these 55 events the same as the EF events or periods? Are they all $< \sim 50$ minutes long? If the CO rose for two 50 minute periods and then fell for two 50 minute periods, is that one peak an emission factor event or is it 4 events? The data reduction can easily be spelled out clearly at the outset for folks that did not do the calculations and might wonder. Has the table of events been called out yet?

**Response:** Yes, these are the same. We clarified by modifying the following sentence: "We identified 55 individual fire-affected events intervals in the observational data from CRV tower (that each span about 50 minutes each) to calculate emission factors from the elevated trace gas observations (Figure 5; Table 2)." We also refer to table 2 in section 2.2 of the Methods.

6, 29-30: The definition of an event earlier was lasting $\sim 900$ or more s? Here all the events lasted 50 minutes? So each hourly measurement interval with high enough CO was an event? I think it might be easy to take a few sentences above to just spell out how data was analyzed. Then I look at Table 2, are these the events and is N the number of 30 s increments? Maybe explain that earlier and include if each of these events is separated by a clean period?

**Response:** We addressed this comment by modifying the methods to make our approach clearer. Please see response to comment 4,1-17.

6, 31: it would be interesting to see range in CH4/CO also in this sentence.

**Response:** We are using $CO_2$ as our reference species, and prefer to only include the ratios with respect to $CO_2$. This can be computed from Table 2.

7, 8: "within" should be "with"? Table 3 called out by mistake? Also on line 16?

**Response:** We changed "within" to "with" and removed the reference to Table 2.

7, 20: diddid

**Response:** We removed the typo.

7, 21, 22, 23: Variability < 5% probably not significant. Were events actually different fires? What is meant by flux-weighted estimates? Accounting for fuel consumption rate in a weighted average EF or windspeed at tower? The highest flux periods at the fire may produce high injection altitudes.

**Response:** We believe that different fires contributed significantly to emission ratios computed for different time intervals. The temporal evolution of different fires shown in Figure 10 provides evidence for this. To address the reviewer comments we changed the final sentence of this paragraph to read: "Although the variation introduced from different weighting approaches was relatively small, the analysis highlights the challenge of combining information from different individual fires, and the importance of moving toward flux-weighted estimates in future work."

7, 25-27: Figure 7 shows some big peaks at tower, but not in model (doy ~188) or modeled peaks not seen at tower. The text says the model confirms elevated CO was primarily from fires. So I guess "primarily" signals > 50% and signals rough agreement? The authors stand by the unmodeled peaks being due to fires? How was it possible to get the fires contributing to the signal at the tower when the model did not capture a peak?

**Response:** We stand by our assertion that unmodeled peaks are caused by fires. We acknowledge that the model is imperfect. We explain possible causes for the model missing elevated CO peaks in the following sentence: "Differences between the model simulations and observations were likely caused by errors in the magnitude and timing of fire emissions within AKFED as well as the limited spatial resolution and incomplete representation of atmospheric transport within PWRF-STILT. Nevertheless, the broad agreement between the model and the observations, including the timing of the large burning event between DOY 173 and 179, provides some confidence that our model can be used to explore the influence and contribution of individual fires."

7, 28: "likelycaused"

**Response:** We removed the typo, added an extra space.

7, 34: average distance weighted by fractional contribution?

**Response:** We added "average distance weighted by fractional contribution."

7, 36: What is meant by "integrate emissions from multiple fires through the full planetary boundary layer"?

**Response:** We removed this sentence and rewrote the discussion in our revised paper.

7, 39: > 8% in Table 3

**Response:** We removed the typo. We now say more than 10%.

8, 1: delete "significantly"

**Response:** We deleted "significantly."

8, 3: 4646%

**Response:** We removed the typo.

8, 8: Andreae-associated recommendations averaged the values from studies using different platforms partly in recognition of bias being possible for any one platform. Akagi et al pioneered splitting extratropical forests into boreal and temperate. They (Sect 2.3.2) actually used a pretty complex scheme averaging smoldering fuels from lab studies by fuel type rather than by study to get a ground-based average, which was then averaged with airborne results for an overall average roughly consistent with about 70% of overall fuel consumption by smoldering. They mentioned evidence that smoldering might be even more important. They devised formulas to estimate compounds measured only in lab or air and invited users to modify any of the

formulas in their Table S2 if they preferred. Remarkably, their default recommendations are almost indistinguishable from this work. Regarding "important" differences on P8, L15, keep in mind that modelers determine the level of detail that works for them and it often involves model domain, scope of study, availability, reliability, and complexity of operational input, but also completeness, i.e. they need ERs/EFs for more than 2 species!

**Response:** We acknowledge that the Akagi et al. approach for combining smoldering fuels and combining it with aircraft observations is an important advance, especially for shorter lived compounds. However, without long-term environmental sampling over the full lifecycle of fires and time-varying environmental conditions that wildfires are experiencing over a period of weeks to months, it is impossible to know how to combine information from smoldering combustion measurements in the laboratory and aircraft samples that may be sampling more flaming combustion phases. This is where the duration and extent of our observations are valuable, as we develop this idea further in the revised discussion.

We agree with the reviewer that the first step in the discussion is to acknowledge the consistency with past work, and we have modified the first paragraph of the discussion to highlight our agreement with previous compilation studies and their strengths with regard to modeling. It now reads:

"The most widely used emission factors for boreal forest fires are derived from syntheses that average together data from individual field campaigns [*Andreae and Merlet*, 2001; *Akagi et al.*, 2011; *Andreae*, 2019]. Our mean emission factor for CO ($127 \pm 59$ g CO per kg of dry biomass consumed) is similar to the mean reported in past syntheses for boreal forests, including estimates by *Andreae* [2019] ($121 \pm 47$ g CO per kg of dry biomass consumed) and *Akagi et al.* [2011] ($127 \pm 45$ g CO per kg of dry biomass consumed). Considering boreal forests as a whole, our measurements provide a partial validation of the approach taken in previous compilations, which have attempted to combine information from different sampling strategies and boreal forest ecoregions. The broad level of agreement provides confidence in the estimates of emission factors for non-conserved species that cannot be measured using our remote tower-based approach."

8, 13 re Table 1: Good idea to parse out data by location and platform and nice overview of data collected. Note Yokelson et al 1997 is missing (used in Akagi Table S2). Boreal peat was burned in Stockwell et al., 2015. Double check if Siberian fires were wild or prescribed, I think at least some were prescribed. Split Siberian fires out by air or ground? Siberian average row has possibly wrong total? Remove line numbers from number of fires column, "McMeeking has two capital "M"s, etc…

Stockwell, C. E., Veres, P. R., Williams, J., and Yokelson, R. J.: Characterization of biomass burning emissions from cooking fires, peat, crop residue, and other fuels with high-resolution proton-transfer-reaction time-of-flight mass spectrometry, Atmos. Chem. Phys., 15, 845-865, doi:10.5194/acp-15-845-2015, 2015.

Yokelson, R.J., D.E. Ward, R.A. Susott, J. Reardon, and D.W.T. Griffith, Emissions from smoldering combustion of biomass measured by open-path Fourier transform infrared spectroscopy, J. Geophys. Res., 102, 18865-18877, 1997.

**Response:** Table 1 includes both airborne and surface measurements from Siberian fires (as noted with the "a" or "s" and explained in the figure caption).We now include Yokelson et al., 1997 and Stockwell et al., 2014 in Table 1, and we identify the type of fuel burned in the North American laboratory studies.

8, 15: "measurement technique" should be "sampling strategy" to be consistent and precise?

**Response:** We changed "measurement technique" to "sampling strategy" throughout the paper.

8, 17-36: The overview of air versus ground sampling of sources is pretty good, a little disorganized but all the most important points emerge clearly! A few points to add could be: Aircraft can replicate tower-based sampling with downwind vertical profiles, but not on a continuous basis like a tower. Also, any aircraft bias toward flaming combustion may actually be partly okay if it weights the EF towards times of higher fuel consumption, relevant to author's desire for flux-based EFs? Flaming always entrains some smoldering, and the entrainment footprint is larger with more intense flaming. Best not to oversimplify a complicated situation.

**Response:** To simplify the discussion, we revised this paragraph.:

"In contrast with remote tower sampling, aircraft-based studies often sample fires that have a strong contribution from flaming combustion, which releases enough energy to generate well-defined plumes at an altitude accessible by the aircraft. This methodology provides an opportunity to comprehensively measure the vertical and horizontal distribution of emissions from an individual fire and their atmospheric evolution in a smoke plume. However, airborne sampling techniques are often limited to daytime periods with good visibility, making it difficult to comprehensively measure emissions over a diurnal cycle or over the full lifetime of a fire which may span several periods with inclement weather. Due to these sampling constraints, aircraft studies are less likely to measure emissions from less energetic smoldering combustion, since these emissions are more likely to remain near the surface [*Ward and Radke*, 1993; *Selimovic et al.,* 2019]. Emissions from smoldering boreal forest fires can sometimes be entrained in the convective columns of certain flaming fires and can be sampled by aircraft, but nighttime emissions or residual smoldering emissions from fires that have weak convective columns usually cannot be measured in this way [*Bertschi et al.,* 2003; *Burling et al.,* 2012]. While past studies have attempted to combine information from aircraft (more likely sampling flaming combustion phases) with laboratory observations of emissions from smoldering combustion [*Akagi et al.*, 2011], the balance of emissions is well known to be highly sensitive to environmental conditions that can rapidly change over the lifetime of a wildfire; this highlights the importance developing sustained sampling approaches that provide regionally-integrated estimates over the full duration of a wildfire event or regional fire complex."

8, 29: "weak or non-existent" convection columns (aka "updraft cores"). Mostly true for fresh RSC emissions, so "usually" is a good qualifier since some RSC may get to aircraft altitude by non-fire uplift or be sampled in rare missed approaches.

**Response:** We agree and will keep "usually."

8, 35: "yet rarely" is okay – a fresh RSC sample would require "a really good drill on the front of the plane" to quote a DC-8 pilot.

**Response:** We agree and will keep "yet rarely."

9, 4: "combustion of"

**Response:** We changed the text to "combustion of."

9, 4-5: Organic soils were focus of lab study of Yokelson et al., 1997 and included in Stockwell et al., 2015 during FLAME-4.

**Response**: These studies are now included in Table 1 and the type of fuel burned is denoted.

9, 7: "should" > "could" or "might" (see above on models)

**Response:** We changed the text to "should."

9, 8: At least five lab studies burned boreal fuels, the CO/CO2 ratios for FLAME-4 for black spruce and boreal peat are in supplement of Stockwell et al., 2015. Listing what fuels were included in averages in Table 1 would be helpful.

**Response:** The fuels per study are now denoted in Table 1.

9,12 "McMeeking"

**Response:** We changed all references to the correct name: "McMeeking."

9, 18-29: The claim of different ERs is not strongly supported. The quoted (Table 1), purely surface-based sampling of Siberian fires had *lower* CO/CO2 than the authors NA work, and, even more remarkably, only about half the CO/CO2 ratio as the studies that included some airborne sampling of Siberian fires. So maybe better to say, the ecosystems differ and the emissions might as well, but not enough data to know yet.

Also work on the Siberia/NA differences goes back to at least 1993 when the Bor Island Experiment was started. Differences in Siberian and North American boreal fires were noted in publications 20-24 years ago with hundreds of references cited and a more recent review on that:

Goldammer, J.G., and V.V. Furyaev. 1996. Fire in ecosystems of boreal Eurasia. Ecological impacts and links to the global system. In: Fire in ecosystems of boreal Eurasia (J.G. Goldammer and V.V. Furyaev, eds.), 1-20. Kluwer Academic Publ., Dordrecht, 528 pp. https://link.springer.com/chapter/10.1007%2F978-94-015-8737-2_1

E.S.Kasischke and B.J.Stocks, eds. 2000. Fire, climate change, and carbon cycling in the boreal forest. Ecological Studies 138, Springer-Verlag, Berlin-Heidelberg-New York, 461 p.

Goldammer, J.G. (ed.) 2013. Prescribed Burning in Russia and Neighbouring Temperate-Boreal Eurasia. A publication of the Global Fire Monitoring Center (GFMC). Kessel Publishing House, 326 p. (ISBN 978-3-941300-71-2). http://www.forestrybooks.com/

**Response:** We respectfully disagree with the reviewer about this point. A Student's t test shows that the set of the Siberian forest fire emission ratios shown in Table 1 are significantly different (and higher) than the remote tower estimates from boreal North America. While it's true there are two fires that are lower than the NA remote tower observations, 7 other fires are quite a bit higher. We acknowledge that more observations are needed with the sentence: "Although more measurements are needed, higher CO emission ratios for Siberian fires appears consistent with past work showing that boreal fire behavior is considerably different between North American and Eurasian continents as a consequence of differences in tree species and their impacts on fire dynamics [*Goldammer and Furyaev*, 1996; *Cofer et al.,* 1996].

We think its important that readers understand that many of Eurasian boreal forest fire emission ratio values are higher than those reported for North America. This is a contributing factor to why there is apparent agreement between our mean emission factor and the ones reported in Akagi et al. [2011] and Andreae [2019]. Lower North American aircraft studies are being offset in a global average in these syntheses by high values measured in Eurasian boreal forests.

We also note that we include both aircraft and surface sampling of Siberian fires (as noted with the "a" or "s" and explained in the figure caption). We explain that the CO emission factor from Siberian boreal fires is higher than North American boreal fires, but "more measurements are needed."

9, 23: "hotter" okay, but there is no single temperature that defines any landscape fire, more aggressive flaming is probably what is meant.

**Response:** We changed the text to fire radiative power, which was the actual quantity reported in Rogers et al. [2015].

9, 30: "Stronger" than what? Not a complete thought. Here the work goes off on a random tangent rehashing a long-recognized issue. Concerns about air/ground bias are discussed in Andreae and Merlet, 2001, which supports this with the following citation:

Andreae, M. O., E. Atlas, H. Cachier, W. R. Cofer, III, G. W. Harris, G. Helas, R. Koppmann, J.P. Lacaux, and D. E. Ward, Trace gas and aerosol emissions from savanna fires, in Biomass Burning and Global Change, edited by J. S. Levine, pp. 278 – 295, MIT Press, Cambridge, Mass., 1996.

Previous recommendations by Akagi and Andreae appear to have compensated adequately for this issue according to this studies results to the extent that we are ever likely to know. The authors could claim that they have investigated the extent of platform-based bias in additional detail and present a useful contribution in that way, but the issue of the existence of differences is not a new finding.

Perhaps an appropriate header is: "A detailed examination of tower versus airborne sampling". Either include or don't include the enigmatic data from Siberia and make a new, useful point if you can, perhaps: a) mean difference is "X", or b) surprisingly no conclusion.

**Response:** We considerably revised the discussion in response to this reviewer and the other reviewers. We no longer have this section title or use the word "stronger".

9, 38 – 10, 1: The authors have good evidence that tower-based platforms see more smoldering that the aircraft studies to date (in NA) and that is useful, but you don't know for sure if the tower might under-estimate flaming or why the Siberian data is enigmatic. And "previous reports" should be changed to "the average of previous airborne studies" since "previous reports" could imply all studies.

**Response:** We changed the text to "from aircraft studies". We now report the difference in the means in the following sentence in the second paragraph of the discussion.

We make the case now in the revised discussion that the tower-based approach likely does a good job of providing a representative sample over the 2015 fire season (second paragraph of discussion):

"Although differences in reported emission ratios are expected between aircraft and ground based sampling approaches [*Christian et al.*, 2007; *Burling et al.*, 2011; *Akagi et al.*, 2014; *Collier et al.*, 2016; *Benedict et al.*, 2017; *Selimovic et al.*, 2019], several features of the CRV tower sampling are conducive to providing a regionally-representative mean estimate of emission ratios during the 2015 Alaska fire season. First, we note that the CRV tower was located at a higher elevation (611 m above sea level) than the core fire complex located in western Alaska and several hundreds of kilometers downwind. Multi-angle Imaging SpectroRadiometer (MISR) satellite observations from Alaskan wildfires indicate most fire plumes reside within the planetary boundary layer, which is typically between 1 and 3 km during midday in summer [*val Martin et al.*, 2010; *Wiggins et al.*, 2016]. Combining this vertical length scale with the mean horizontal distance of the 34 fires that most influenced CO at CRV (259 km), we obtain a factor of about 100 for a back-of-the-envelope ratio of horizontal to vertical mixing processes. This ratio, together with the simulated time delay of 1-2 days between emission and detection of CO anomalies at CRV (Figure 3), imply that mesoscale atmospheric circulation played an important role in averaging together trace gas emissions from multiple fires before the air masses were sampled (Figure 10). As a result, observations from the CRV tower represent a temporal integration of fire emissions over day-night burning cycles as well as a spatial integration across

flaming combustion at active fire fronts along with residual smoldering combustion in soils that often persists for days after a fire front moves through an area. Collectively, the fires sampled at CRV appeared to experience time-varying environmental conditions that were less ideal for flaming combustion than the fire plumes sampled in past work by aircraft. This finding is consistent with remote tower observations of the black carbon to CO ratio measured for wildfires from temperate North America [*Selimovic et al.*, 2019]."

We also acknowledge that ground-based sampling may under sample some emission injected above the pbl: "Ground-based sites may potentially miss some of the emissions injected above the planetary boundary layer and subject to long-range transport in the free troposphere."

10,1-2: at a minimum change to "some previous" , "some flaming", "some residual"

**Response:** This text has been deleted in the revised discussion.

10, 1 – 8: Showing that the tower and aircraft got different overall average CO/CO2 ratios is straightforward and useful. But both platforms could have some error so proving that 100% of the error in representativeness is with the aircraft is not really doable. Every fire that impacted the tower also, undoubtedly produced some emissions that did not impact the tower due to wind shifts, altitude, or whatever, it's just basic common sense. The most exciting thing about this work is not even stressed. That is, by measuring downwind of many fires burning at all stages of their life cycle around-the-clock, the authors have created a high-quality data set for evaluating fire emissions models performance at a regional level (as in Selimovic et al., 2019; 2020). I.e. AKFED did pretty well integrating the effects of many fires around the clock and predicting the tower "point CO" specifically. What would need to be changed in AKFED/STILT to improve performance could be a great follow-on study along with how do larger-scale models such as GFED, GFAS, FINN, etc., perform against the tower observations! Regardless of the "real, unknown total fire emissions," the signal at the tower is well-measured now and very useful to test models!

**Response:** We modified our conclusions section to highlight the potential use of the CRV tower dataset to evaluate regional fire emissions model performance. Please see response to the reviewer's general comments for more information.

10, 9: This next paragraph repeats some of the material in the previous paragraph. If the text survives editing, change "crown" to "surface fuels" since the NW Territory crown fire experiment found that often the fires propagate in surface fuels followed by torching

**Response:** This text was deleted in the revised discussion.

10, 10: not a sentence, delete "that" to fix?

**Response:** This text was deleted in the revised discussion.

10, 11: substantial contributions of RSC were stressed by Bertschi et al 2003 in their Table 3.

**Response:** This text was deleted in the revised discussion.

10, 12: not only FRP-based, but any thermal signal. Smoldering involves temperature high enough to saturate the 3.9 micron channel if widespread enough, but obstruction is more of an issue for deep smoldering.

**Response:** This text was deleted in the revised draft.

10,16: "previously thought" by who? If you mean the studies mentioned directly above, no comparison is possible unless you also have the relative consumption of above- and belowground fuels. If you mean a

previous compilation, Akagi et al estimated 70% of all boreal fuel was consumed by smoldering based on an MCE similar to this work.

**Response:** This section has been removed from the discussion.

10, 18: "most" is not "all" and MISR only looks at 1030 AM long before both the most intense combustion and diurnal cycle fuel consumption peak. I would change "most" to "many" or "some"

**Response:** We changed "most" to "many."

10, 19: "length scale"?

**Response:** We changed this to "vertical length scale" and denote the distance is in reference to the "horizontal."

10, 20 – 21: good point there is time for vertical mixing, if the atmosphere is not too stratified.

**Response:** We agree and appreciate the reviewer's confirmation.

10, 22-25: This sentence is not accurate as Collier et al sampled smoke up to 48 hours old and Selimovic et al 2019; 2020 sampled smoke from fires in the range 20-800 km upwind.

**Response:** We agree, and have cited the Selimovic et al. and Collier et al. studies in other places for other specific contributions of this work.

10, 24: if text survives "band" > "and"

**Response:** This section was removed.

10, 26-35: This discussion is oversimplified since dry weather can make larger fuels that tend to smolder more likely to burn. See section 2.4 in Akagi et al and the papers referenced therein by Hoffa et al., 1999, Shea et al., 1996, Kauffman et al., 1998; 2003, etc. Hot, dry weather can increase smoldering. Note also that most airborne studies occurred in years that were arguably more "typical".

**Response:** We believe it's important to remind the reader that 2015 was an extreme fire season with very high surface air temperatures during June. This is important context for interpreting our emission ratios. It's also important, we believe, to let the reader know that we attempted to examine day to day variations in regional weather and link these variations to emission ratios.

Coming at the problem from a different perspective, its very clear that periods of hot and dry weather (higher VPD) allow for faster fire spread rates in interior Alaska [*Sedano and Randerson*, 2014], likely as a consequence of fires moving through the crowns of black spruce rather than along the surface where fires move slowly. VPD also may have a small but significant effect on fire severity and fuel consumption [*Veraverbeke et al.*, 2015]. While we agree that hotter and drier weather may allow coarser fuel classes to burn, it's not clear to us that in boreal forests this is enough to offset a stronger crown burning and flaming combustion phase. This why we frame our inquiry here as a question. We revised the text in the paragraph, but would respectfully prefer to keep this paragraph in our revised paper:

"During the latter half of June and early July of 2015, weather in Alaska was very hot and dry, allowing for a record number of fires to rapidly expand in size, and yielding the second highest level of annual burned area in the observed record. The extreme fire weather conditions would be expected to reduce fuel moisture content, thus promoting crown fires and flaming combustion processes [e.g., *Sedano and Randerson*, 2014]. This raises the question of whether longer term monitoring of many normal and low fire

years (which tend to co-occur in cooler and wetter conditions) would provide evidence for an even larger role of smoldering combustion compared to the estimates we report here for 2015. Another related question is whether even within a fire season, do day-to-day or week-to-week variations in fire weather influence variability in emission ratios? We explored this latter question with the datasets described here but were unable to uncover structural relationships between daily meteorological variables such as vapor pressure deficit and CO emission ratios. Together, these questions represent important directions for future research and emphasize the critical need of sustained long-term support for trace gas monitoring networks and field campaigns."

10, 40: I don't think any new ideas "emerged", but the authors work can help continue to evaluate some long-recognized issues and maybe help reduce uncertainty.

**Response:** We considerably revised this paragraph, recognizing the reviewer's suggestion regarding tone. We no longer use the word "emerged". The topic sentence for this paragraph is now: "The observations summarized in Table 1 also show there are several important differences in boreal forest emission ratios that exist as a function sampling strategy and ecoregion."

11, 1: it always makes sense to "report" what you measured and studies of Siberian fires report their location. "Using" regionally-specific EFs might make sense, but is a separate decision for the modelers that is hard because few measurements have occurred in Russia where research access is super-problematic. A colleague had their canisters confiscated by the Russian military
and "filled for them at undisclosed locations."

**Response:** We agree more measurements are needed, and we qualify our statement in the following sentence: "More data, particularly for Siberian fires, is needed to assess whether the differences in emission factors noted here are robust."

11, 5 – 11: This is not that big a deal, the complex averaging scheme of Akagi gave almost the same answer as this study or the simple averaging scheme of Andreae. Adding this studies extensive results in a weighted or simple average to the "evolving literature average" is BAU and will have little impact on the average; though it is important to be clear about how things are synthesized.

**Response:** We modified our discussion section to highlight the agreement between our measurements and the compilation studies: "Considering boreal forests as a whole, our measurements provide a partial validation of the approach taken in previous compilations, which have attempted to combine information from different sampling strategies and boreal forest ecoregions. The broad level of agreement provides confidence in the estimates of emission factors for non-conserved species that cannot be measured using a remote tower sampling approach." However, we also believe it is important to quantify the magnitude of the differences in measured emission ratios between sampling strategies. Please see response to the reviewer's general comments for more information.

11, 12 - 18: It is not clear what is meant by flux-weighted EF? Aircraft measurements may in fact be weighted towards times with high fuel consumption rates. Fires can produce multiple plumes. A flux of emissions in models results from the fuel consumption rate assumptions.

**Response:** We modified this section to better explain flux-weighted emission factors: "Long-term monitoring from remote towers has the potential to provide new information about fire complexes in other biomes, integrating across day-night variations in fire behavior, periods with different environmental conditions, and across multiple fires in different stages of growth and extinction. In this context, more work is needed to find ways to combine tower and aircraft sampling to attain accurate estimates of the total budget of fire-emitted trace gases and aerosols (i.e., estimating flux-weighted emission factors), given the large differences in data density and the different strengths and weaknesses of the two approaches."

11, 18-23: This is just stating obvious that if we could measure everything, we'd know more. I would very strongly recommend deleting sections 4.3 (and 4.4) and instead have a section to highlight the exciting model evaluation now possible with what you *already* measured.

**Response:** We deleted sections 4.3 and 4.4, following the reviewer suggestion. In the conclusions, we now comment on the value of our observations for testing models. We specifically added the following sentence: "Together, the two-month near continuous time series of $CO_2$, CO, and $CH_4$, along with the derived emission ratios reported here, may provide a means to test and evaluate models that couple together fire processes, emissions, and regional atmospheric transport."

11, 24: "larger" than what?
4.4 is all speculation about PM, which was not even measured and the section doesn't consider SOA or PM evaporation where the latter was significant in Selimovic et al., 2019, 2020, and references there-in. Also, health impacts are based on measured PM and this study does not suggest the regional PM networks are inaccurate.

**Response:** This section was removed.

11, 27: higher than some studies doesn't equal higher than "previously thought".

**Response:** We deleted this section.

11, 28 – 29: Long range transported smoke was not sampled in this study and that type of smoke may actually be better sampled from aircraft.

**Response:** We deleted this section.

11, 25 – 36: delete, all speculation, not a topic or result of this paper.

**Response:** We deleted this section.

12, 3: after "Our results" I would delete the rest and fill in the valuable insights that you actually learned about the AKFED model. I.e. it underestimated nighttime combustion impacts at the tower, it captured X of the Y peaks, seasonal average CO was within Z%, etc… Highlight potential for additional, future model evaluation and improvement.

**Response:** We changed this sentence to: "Our results suggest the CRV tower-based dataset can be used to evaluate model predictions of fire emissions and their transport at a regional scale."

This paper presents trace gas observations of CO, CH4, and CO2 at the CRV tower to estimate emission factors from boreal forest fires during the Alaska extreme fire season of 2015. The high-quality boreal forest fire smoke measurements are combined with Lagrangian modelling to characterize wildfire emissions.

The work is of high quality and excepting some few points, that need to be addressed, both description and discussion of measurements/modelling are well founded. The manuscript contributes to scientific progress within the scope of the journal, therefore it is suitable to be published in ACP.

**General comments:**

The resubmitted manuscript has been largely improved. There is only one general point, which needs some detailed discussions. The authors state that PWRF-STILT forward simulations were done (Page7Line25) to determine footprint fields necessary for the convolution with fire emissions from AKFED. To interpret observation at the CRV tower is a classic case of source-receptor studies that usually employ LPDM backward runs starting at the observation site, as shown by Henderson et al. (ACP2015). The authors should comment on that.

**Response:** We added the following sentence in the first paragraph of our model description to address this reviewer comment.

For this application, STILT [*Lin et al.*, 2007] was used to estimate the adjoint of PWRF [*Skamarock et al.*, 2005; *Chang et al.*, 2014; *Henderson et al.*, 2015] during the summer of 2015 at the location of the CRV tower, to generate surface influence functions that relate surface fluxes from Alaska to trace mole fractions at the CRV tower. These gridded influence functions are known as footprints and have units of mole fraction per unit of surface flux (ppm/($\mu$mol m$^{-2}$ s$^{-1}$).

**Specific Comments**

Equation 2 still needs small revisions: replace 12.01 by MM_{C} as carbon molar mass; remove g/kg and explain the conversion factor 1000 in text.

**Response:** Following the reviewer's suggestion, we modified equation 2, replacing 12.01 with the molar mass of carbon and taking the units away from the factor of 1000, but explaining the units of this factor in text.

- Page2Line34 remove [.

**Response:** We removed the extra "[", correcting this typo.

- Page7Line28 a blank is missing between 'likely' and 'caused'

**Response:** We inserted a space between these two words.

- Page9Line insert 'CO' prior to 'emissions': 'Yet, Table 1 shows CO emission ratios from wildfires in boreal Siberia tend to be higher than emission ratios from North American wildfires'

**Response:** We considerably revised the discussion in response to reviewer #2. This discussion header no longer exists. We modified the following sentence in the second paragraph of the discussion to make this point, replacing "fire" with "CO" in the revised sentence: "
[revised manuscript text omitted]

4.2 Evidence for a stronger role of smoldering combustion in emissions from North American boreal wildfires→¶

Our mean emission factor for CO (127 ± 59 g CO per kg of dry biomass consumed) is similar to the mean reported in past syntheses for boreal forests, including estimates by Andreae [2019] (121 ± 47 g CO per kg of dry biomass consumed) and Akagi et al. [2011] (127 ± 45 g CO per kg of dry biomass consumed). However, if studies that are not representative of North American boreal forest wildfires are excluded (including measurements from prescribed fires, laboratory studies, and studies of fires from the Eurasian boreal forests) and we focus on emission ratios, to avoid... [3]

Moved up [6]: Wiggins et al.,

Moved up [7]: were unable to uncover structural relationships between daily meteorological variables such as vapor pressure deficit and CO emission ratios. Together, these questions represent important directions for future

Moved (insertion) [8]

[revised manuscript text omitted]

**Page 26: [1] Deleted  Wiggins, Elizabeth B. (LARC-E3)[UNIVERSITIES SPACE RESEARCH ASSOCIATION]     2/3/21 1:00:00 PM**

**Page 26: [2] Deleted  Wiggins, Elizabeth B. (LARC-E3)[UNIVERSITIES SPACE RESEARCH ASSOCIATION]     2/3/21 1:00:00 PM**

**Page 33: [3] Deleted  Wiggins, Elizabeth B. (LARC-E3)[UNIVERSITIES SPACE RESEARCH ASSOCIATION]     2/3/21 1:00:00 PM**

**Page 33: [4] Deleted  Wiggins, Elizabeth B. (LARC-E3)[UNIVERSITIES SPACE RESEARCH ASSOCIATION]     2/3/21 1:00:00 PM**

**Page 34: [5] Deleted  Wiggins, Elizabeth B. (LARC-E3)[UNIVERSITIES SPACE RESEARCH ASSOCIATION]     2/3/21 1:00:00 PM**

**Page 34: [6] Deleted  Wiggins, Elizabeth B. (LARC-E3)[UNIVERSITIES SPACE RESEARCH ASSOCIATION]     2/3/21 1:00:00 PM**

**Page 34: [7] Deleted  Wiggins, Elizabeth B. (LARC-E3)[UNIVERSITIES SPACE RESEARCH ASSOCIATION]     2/3/21 1:00:00 PM**

**Page 34: [8] Deleted  Wiggins, Elizabeth B. (LARC-E3)[UNIVERSITIES SPACE RESEARCH ASSOCIATION]     2/3/21 1:00:00 PM**

**Page 34: [9] Deleted  Wiggins, Elizabeth B. (LARC-E3)[UNIVERSITIES SPACE RESEARCH ASSOCIATION]     2/3/21 1:00:00 PM**

**Page 34: [10] Formatted  Wiggins, Elizabeth B. (LARC-E3)[UNIVERSITIES SPACE RESEARCH ASSOCIATION]     2/3/21 1:00:00 PM**

Font color: Black, Check spelling and grammar

**Page 34: [11] Formatted  Wiggins, Elizabeth B. (LARC-E3)[UNIVERSITIES SPACE RESEARCH ASSOCIATION]     2/3/21 1:00:00 PM**

Normal, Border: Top: (No border), Bottom: (No border), Left: (No border), Right: (No border), Between : (No border)

**Page 34: [12] Formatted  Wiggins, Elizabeth B. (LARC-E3)[UNIVERSITIES SPACE RESEARCH ASSOCIATION]     2/3/21 1:00:00 PM**

Add space between paragraphs of the same style

**Page 34: [13] Formatted  Wiggins, Elizabeth B. (LARC-E3)[UNIVERSITIES SPACE RESEARCH ASSOCIATION]     2/3/21 1:00:00 PM**

Default Paragraph Font, Font: 12 pt, Font color: Blue, Check spelling and grammar

**Page 35: [14] Deleted  Wiggins, Elizabeth B. (LARC-E3)[UNIVERSITIES SPACE RESEARCH ASSOCIATION]     2/3/21 1:00:00 PM**

**Page 35: [15] Formatted  Wiggins, Elizabeth B. (LARC-E3)[UNIVERSITIES SPACE RESEARCH ASSOCIATION]                2/3/21 1:00:00 PM**

Add space between paragraphs of the same style

**Page 35: [16] Formatted  Wiggins, Elizabeth B. (LARC-E3)[UNIVERSITIES SPACE RESEARCH ASSOCIATION]                2/3/21 1:00:00 PM**

Font color: Black, Check spelling and grammar

**Page 35: [17] Formatted  Wiggins, Elizabeth B. (LARC-E3)[UNIVERSITIES SPACE RESEARCH ASSOCIATION]                2/3/21 1:00:00 PM**

Normal, Add space between paragraphs of the same style, Border: Top: (No border), Bottom: (No border), Left: (No border), Right: (No border), Between : (No border)

**Page 35: [18] Formatted  Wiggins, Elizabeth B. (LARC-E3)[UNIVERSITIES SPACE RESEARCH ASSOCIATION]                2/3/21 1:00:00 PM**

Font color: Black, Check spelling and grammar

**Page 35: [18] Formatted  Wiggins, Elizabeth B. (LARC-E3)[UNIVERSITIES SPACE RESEARCH ASSOCIATION]                2/3/21 1:00:00 PM**

Font color: Black, Check spelling and grammar

**Page 35: [19] Formatted  Wiggins, Elizabeth B. (LARC-E3)[UNIVERSITIES SPACE RESEARCH ASSOCIATION]                2/3/21 1:00:00 PM**

Font color: Black

**Page 35: [19] Formatted  Wiggins, Elizabeth B. (LARC-E3)[UNIVERSITIES SPACE RESEARCH ASSOCIATION]                2/3/21 1:00:00 PM**

Font color: Black

**Page 35: [19] Formatted  Wiggins, Elizabeth B. (LARC-E3)[UNIVERSITIES SPACE RESEARCH ASSOCIATION]                2/3/21 1:00:00 PM**

Font color: Black

**Page 35: [20] Deleted  Wiggins, Elizabeth B. (LARC-E3)[UNIVERSITIES SPACE RESEARCH ASSOCIATION]                2/3/21 1:00:00 PM**

**Page 35: [21] Formatted  Wiggins, Elizabeth B. (LARC-E3)[UNIVERSITIES SPACE RESEARCH ASSOCIATION]                2/3/21 1:00:00 PM**

Font color: Black, Check spelling and grammar

**Page 35: [22] Deleted  Wiggins, Elizabeth B. (LARC-E3)[UNIVERSITIES SPACE RESEARCH ASSOCIATION]                2/3/21 1:00:00 PM**

**Page 35: [23] Deleted  Wiggins, Elizabeth B. (LARC-E3)[UNIVERSITIES SPACE RESEARCH ASSOCIATION]                2/3/21 1:00:00 PM**

**Page 35: [24] Formatted  Wiggins, Elizabeth B. (LARC-E3)[UNIVERSITIES SPACE RESEARCH ASSOCIATION]                2/3/21 1:00:00 PM**

Font color: Black, Check spelling and grammar

**Page 35: [25] Formatted  Wiggins, Elizabeth B. (LARC-E3)[UNIVERSITIES SPACE RESEARCH ASSOCIATION]**                    **2/3/21 1:00:00 PM**

Font color: Black, Check spelling and grammar

**Page 35: [25] Formatted  Wiggins, Elizabeth B. (LARC-E3)[UNIVERSITIES SPACE RESEARCH ASSOCIATION]**                    **2/3/21 1:00:00 PM**

Font color: Black, Check spelling and grammar

**Page 35: [26] Deleted  Wiggins, Elizabeth B. (LARC-E3)[UNIVERSITIES SPACE RESEARCH ASSOCIATION]**                    **2/3/21 1:00:00 PM**

**Page 35: [27] Formatted  Wiggins, Elizabeth B. (LARC-E3)[UNIVERSITIES SPACE RESEARCH ASSOCIATION]**                    **2/3/21 1:00:00 PM**

Font color: Custom Color(RGB(34,34,34)), Pattern: Clear (White)

**Page 35: [28] Formatted  Wiggins, Elizabeth B. (LARC-E3)[UNIVERSITIES SPACE RESEARCH ASSOCIATION]**                    **2/3/21 1:00:00 PM**

Font color: Custom Color(RGB(28,29,30)), Pattern: Clear (White)

**Page 35: [29] Deleted  Wiggins, Elizabeth B. (LARC-E3)[UNIVERSITIES SPACE RESEARCH ASSOCIATION]**                    **2/3/21 1:00:00 PM**

**Page 35: [30] Formatted  Wiggins, Elizabeth B. (LARC-E3)[UNIVERSITIES SPACE RESEARCH ASSOCIATION]**                    **2/3/21 1:00:00 PM**

Font color: Custom Color(RGB(34,34,34)), Pattern: Clear (White)

**Page 35: [31] Formatted  Wiggins, Elizabeth B. (LARC-E3)[UNIVERSITIES SPACE RESEARCH ASSOCIATION]**                    **2/3/21 1:00:00 PM**

Font color: Custom Color(RGB(34,34,34)), Pattern: Clear (White)

**Page 35: [32] Deleted  Wiggins, Elizabeth B. (LARC-E3)[UNIVERSITIES SPACE RESEARCH ASSOCIATION]**                    **2/3/21 1:00:00 PM**

**Page 35: [33] Deleted  Wiggins, Elizabeth B. (LARC-E3)[UNIVERSITIES SPACE RESEARCH ASSOCIATION]**                    **8/3/20 3:34:00 PM**

**Page 37: [34] Formatted  Wiggins, Elizabeth B. (LARC-E3)[UNIVERSITIES SPACE RESEARCH ASSOCIATION]**                    **2/3/21 1:00:00 PM**

Font color: Black, Check spelling and grammar

**Page 37: [35] Formatted  Wiggins, Elizabeth B. (LARC-E3)[UNIVERSITIES SPACE RESEARCH ASSOCIATION]**                    **2/3/21 1:00:00 PM**

Font color: Black, Check spelling and grammar

**Page 37: [36] Formatted  Wiggins, Elizabeth B. (LARC-E3)[UNIVERSITIES SPACE RESEARCH ASSOCIATION]**                    **2/3/21 1:00:00 PM**

Font color: Black, Check spelling and grammar

**Page 37: [37] Formatted  Wiggins, Elizabeth B. (LARC-E3)[UNIVERSITIES SPACE RESEARCH ASSOCIATION]**                    **2/3/21 1:00:00 PM**

Font color: Black, Check spelling and grammar

**Page 37: [38] Formatted  Wiggins, Elizabeth B. (LARC-E3)[UNIVERSITIES SPACE RESEARCH ASSOCIATION]**                    **2/3/21 1:00:00 PM**

Font: (Default) Times New Roman, Font color: Black, Check spelling and grammar

**Page 37: [39] Formatted  Wiggins, Elizabeth B. (LARC-E3)[UNIVERSITIES SPACE RESEARCH ASSOCIATION]                    2/3/21 1:00:00 PM**

Font color: Black, Check spelling and grammar

**Page 37: [40] Formatted  Wiggins, Elizabeth B. (LARC-E3)[UNIVERSITIES SPACE RESEARCH ASSOCIATION]                    2/3/21 1:00:00 PM**

Font: (Default) Times New Roman, Font color: Black, Check spelling and grammar

**Page 37: [41] Formatted  Wiggins, Elizabeth B. (LARC-E3)[UNIVERSITIES SPACE RESEARCH ASSOCIATION]                    2/3/21 1:00:00 PM**

Font color: Black, Check spelling and grammar

**Page 37: [42] Deleted  Wiggins, Elizabeth B. (LARC-E3)[UNIVERSITIES SPACE RESEARCH ASSOCIATION]                    2/3/21 1:00:00 PM**

**Page 37: [43] Formatted  Wiggins, Elizabeth B. (LARC-E3)[UNIVERSITIES SPACE RESEARCH ASSOCIATION]                    2/3/21 1:00:00 PM**

Font color: Custom Color(RGB(28,29,30)), Pattern: Clear (White)

**Page 37: [44] Deleted  Wiggins, Elizabeth B. (LARC-E3)[UNIVERSITIES SPACE RESEARCH ASSOCIATION]                    2/3/21 1:00:00 PM**

**Page 37: [45] Deleted  Wiggins, Elizabeth B. (LARC-E3)[UNIVERSITIES SPACE RESEARCH ASSOCIATION]                    2/3/21 1:00:00 PM**

**Page 47: [46] Deleted  Wiggins, Elizabeth B. (LARC-E3)[UNIVERSITIES SPACE RESEARCH ASSOCIATION]                    2/3/21 1:00:00 PM**

**Page 47: [46] Deleted  Wiggins, Elizabeth B. (LARC-E3)[UNIVERSITIES SPACE RESEARCH ASSOCIATION]                    2/3/21 1:00:00 PM**

**Page 47: [46] Deleted  Wiggins, Elizabeth B. (LARC-E3)[UNIVERSITIES SPACE RESEARCH ASSOCIATION]                    2/3/21 1:00:00 PM**

**Page 47: [46] Deleted  Wiggins, Elizabeth B. (LARC-E3)[UNIVERSITIES SPACE RESEARCH ASSOCIATION]                    2/3/21 1:00:00 PM**

**Page 47: [46] Deleted  Wiggins, Elizabeth B. (LARC-E3)[UNIVERSITIES SPACE RESEARCH ASSOCIATION]                    2/3/21 1:00:00 PM**

**Page 47: [46] Deleted  Wiggins, Elizabeth B. (LARC-E3)[UNIVERSITIES SPACE RESEARCH ASSOCIATION]                    2/3/21 1:00:00 PM**

**Page 52: [47] Deleted  Wiggins, Elizabeth B. (LARC-E3)[UNIVERSITIES SPACE RESEARCH ASSOCIATION]                    2/3/21 1:00:00 PM**

**Page 52: [48] Deleted  Wiggins, Elizabeth B. (LARC-E3)[UNIVERSITIES SPACE RESEARCH ASSOCIATION]                    2/3/21 1:00:00 PM**

**Page 52: [49] Deleted  Wiggins, Elizabeth B. (LARC-E3)[UNIVERSITIES SPACE RESEARCH ASSOCIATION]                    2/3/21 1:00:00 PM**

**Page 53: [50] Formatted  Wiggins, Elizabeth B. (LARC-E3)[UNIVERSITIES SPACE RESEARCH ASSOCIATION]                    2/3/21 1:00:00 PM**

Left

**Page 53: [51] Moved from page 54 (Move #36)  Wiggins, Elizabeth B. (LARC-E3)[UNIVERSITIES SPACE RESEARCH ASSOCIATION]                    2/3/21 1:00:00 PM**

**Table 1**. Comparison of CO emission ratio and modified combustion efficiency (MCE) from previous studies that sampled emissions from boreal forest fires. The studies are organized according to wildfire domain (North America or Siberia), management practice (wildfire or management fire), and sampling approach (aircraft, laboratory, or surface tower). Siberian studies are indicated as aircraft studies (A), surface based studies (S), or a combination of the two (A & S). The CO emission ratio column has units of ppmv ppmv$^{-1}$ and uses $CO_2$ as the reference gas.

**Page 53: [51] Moved from page 54 (Move #36)  Wiggins, Elizabeth B. (LARC-E3)[UNIVERSITIES SPACE RESEARCH ASSOCIATION]                    2/3/21 1:00:00 PM**

**Page 53: [52] Formatted  Wiggins, Elizabeth B. (LARC-E3)[UNIVERSITIES SPACE RESEARCH ASSOCIATION]                    2/3/21 1:00:00 PM**

Font: Bold

**Page 53: [53] Deleted  Wiggins, Elizabeth B. (LARC-E3)[UNIVERSITIES SPACE RESEARCH ASSOCIATION]                    2/3/21 1:00:00 PM**

**Page 53: [54] Formatted  Wiggins, Elizabeth B. (LARC-E3)[UNIVERSITIES SPACE RESEARCH ASSOCIATION]                    2/3/21 1:00:00 PM**

Position: Horizontal: Left, Relative to: Column, Vertical: In line, Relative to: Margin, Horizontal:  0", Wrap Around

**Page 53: [55] Formatted  Wiggins, Elizabeth B. (LARC-E3)[UNIVERSITIES SPACE RESEARCH ASSOCIATION]                    2/3/21 1:00:00 PM**

Font color: Auto

**Page 53: [56] Formatted Table  Wiggins, Elizabeth B. (LARC-E3)[UNIVERSITIES SPACE RESEARCH ASSOCIATION]                    2/3/21 1:00:00 PM**

Formatted Table

**Page 53: [57] Formatted  Wiggins, Elizabeth B. (LARC-E3)[UNIVERSITIES SPACE RESEARCH ASSOCIATION]                    2/3/21 1:00:00 PM**

Font color: Black

**Page 53: [58] Formatted  Wiggins, Elizabeth B. (LARC-E3)[UNIVERSITIES SPACE RESEARCH ASSOCIATION]                    2/3/21 1:00:00 PM**

Font color: Auto

**Page 53: [59] Formatted  Wiggins, Elizabeth B. (LARC-E3)[UNIVERSITIES SPACE RESEARCH ASSOCIATION]                    2/3/21 1:00:00 PM**

Position: Horizontal: Left, Relative to: Column, Vertical: In line, Relative to: Margin, Horizontal:  0", Wrap Around

**Page 53: [60] Formatted  Wiggins, Elizabeth B. (LARC-E3)[UNIVERSITIES SPACE RESEARCH ASSOCIATION]                    2/3/21 1:00:00 PM**

Font color: Black

**Page 53: [61] Formatted  Wiggins, Elizabeth B. (LARC-E3)[UNIVERSITIES SPACE RESEARCH ASSOCIATION]                    2/3/21 1:00:00 PM**

Font color: Auto

**Page 53: [61] Formatted  Wiggins, Elizabeth B. (LARC-E3)[UNIVERSITIES SPACE RESEARCH ASSOCIATION]                    2/3/21 1:00:00 PM**

Font color: Auto

**Page 53: [62] Formatted  Wiggins, Elizabeth B. (LARC-E3)[UNIVERSITIES SPACE RESEARCH ASSOCIATION]                    2/3/21 1:00:00 PM**

Font color: Auto

**Page 53: [62] Formatted  Wiggins, Elizabeth B. (LARC-E3)[UNIVERSITIES SPACE RESEARCH ASSOCIATION]                    2/3/21 1:00:00 PM**

Font color: Auto

**Page 53: [63] Formatted  Wiggins, Elizabeth B. (LARC-E3)[UNIVERSITIES SPACE RESEARCH ASSOCIATION]                    2/3/21 1:00:00 PM**

Font color: Auto

**Page 53: [63] Formatted  Wiggins, Elizabeth B. (LARC-E3)[UNIVERSITIES SPACE RESEARCH ASSOCIATION]                    2/3/21 1:00:00 PM**

Font color: Auto

**Page 53: [64] Formatted  Wiggins, Elizabeth B. (LARC-E3)[UNIVERSITIES SPACE RESEARCH ASSOCIATION]                    2/3/21 1:00:00 PM**

Font color: Auto

**Page 53: [65] Formatted  Wiggins, Elizabeth B. (LARC-E3)[UNIVERSITIES SPACE RESEARCH ASSOCIATION]                    2/3/21 1:00:00 PM**

Position: Horizontal: Left, Relative to: Column, Vertical: In line, Relative to: Margin, Horizontal:  0", Wrap Around

**Page 53: [66] Formatted  Wiggins, Elizabeth B. (LARC-E3)[UNIVERSITIES SPACE RESEARCH ASSOCIATION]                    2/3/21 1:00:00 PM**

Font color: Black

**Page 53: [67] Formatted  Wiggins, Elizabeth B. (LARC-E3)[UNIVERSITIES SPACE RESEARCH ASSOCIATION]                    2/3/21 1:00:00 PM**

Font color: Auto

**Page 53: [67] Formatted  Wiggins, Elizabeth B. (LARC-E3)[UNIVERSITIES SPACE RESEARCH ASSOCIATION]                2/3/21 1:00:00 PM**

Font color: Auto

**Page 53: [68] Formatted  Wiggins, Elizabeth B. (LARC-E3)[UNIVERSITIES SPACE RESEARCH ASSOCIATION]                2/3/21 1:00:00 PM**

Font color: Auto

**Page 53: [68] Formatted  Wiggins, Elizabeth B. (LARC-E3)[UNIVERSITIES SPACE RESEARCH ASSOCIATION]                2/3/21 1:00:00 PM**

Font color: Auto

**Page 53: [69] Formatted  Wiggins, Elizabeth B. (LARC-E3)[UNIVERSITIES SPACE RESEARCH ASSOCIATION]                2/3/21 1:00:00 PM**

Font color: Auto

**Page 53: [69] Formatted  Wiggins, Elizabeth B. (LARC-E3)[UNIVERSITIES SPACE RESEARCH ASSOCIATION]                2/3/21 1:00:00 PM**

Font color: Auto

**Page 53: [70] Formatted  Wiggins, Elizabeth B. (LARC-E3)[UNIVERSITIES SPACE RESEARCH ASSOCIATION]                2/3/21 1:00:00 PM**

Font color: Auto

**Page 53: [71] Formatted  Wiggins, Elizabeth B. (LARC-E3)[UNIVERSITIES SPACE RESEARCH ASSOCIATION]                2/3/21 1:00:00 PM**

Position: Horizontal: Left, Relative to: Column, Vertical: In line, Relative to: Margin, Horizontal:  0", Wrap Around

**Page 53: [72] Formatted  Wiggins, Elizabeth B. (LARC-E3)[UNIVERSITIES SPACE RESEARCH ASSOCIATION]                2/3/21 1:00:00 PM**

Font color: Black

**Page 53: [73] Formatted  Wiggins, Elizabeth B. (LARC-E3)[UNIVERSITIES SPACE RESEARCH ASSOCIATION]                2/3/21 1:00:00 PM**

Font color: Auto

**Page 53: [73] Formatted  Wiggins, Elizabeth B. (LARC-E3)[UNIVERSITIES SPACE RESEARCH ASSOCIATION]                2/3/21 1:00:00 PM**

Font color: Auto

**Page 53: [74] Formatted  Wiggins, Elizabeth B. (LARC-E3)[UNIVERSITIES SPACE RESEARCH ASSOCIATION]                2/3/21 1:00:00 PM**

Font color: Auto

**Page 53: [74] Formatted  Wiggins, Elizabeth B. (LARC-E3)[UNIVERSITIES SPACE RESEARCH ASSOCIATION]                2/3/21 1:00:00 PM**

Font color: Auto

**Page 53: [75] Formatted  Wiggins, Elizabeth B. (LARC-E3)[UNIVERSITIES SPACE RESEARCH ASSOCIATION]                2/3/21 1:00:00 PM**

Font color: Auto

**Page 53: [75] Formatted  Wiggins, Elizabeth B. (LARC-E3)[UNIVERSITIES SPACE RESEARCH ASSOCIATION]                2/3/21 1:00:00 PM**

Font color: Auto

**Page 53: [76] Formatted   Wiggins, Elizabeth B. (LARC-E3)[UNIVERSITIES SPACE RESEARCH ASSOCIATION]                    2/3/21 1:00:00 PM**

Font color: Auto

**Page 53: [77] Formatted   Wiggins, Elizabeth B. (LARC-E3)[UNIVERSITIES SPACE RESEARCH ASSOCIATION]                    2/3/21 1:00:00 PM**

Position: Horizontal: Left, Relative to: Column, Vertical: In line, Relative to: Margin, Horizontal:  0", Wrap Around

**Page 53: [78] Formatted   Wiggins, Elizabeth B. (LARC-E3)[UNIVERSITIES SPACE RESEARCH ASSOCIATION]                    2/3/21 1:00:00 PM**

Font color: Black

**Page 53: [79] Formatted   Wiggins, Elizabeth B. (LARC-E3)[UNIVERSITIES SPACE RESEARCH ASSOCIATION]                    2/3/21 1:00:00 PM**

Font color: Auto

**Page 53: [79] Formatted   Wiggins, Elizabeth B. (LARC-E3)[UNIVERSITIES SPACE RESEARCH ASSOCIATION]                    2/3/21 1:00:00 PM**

Font color: Auto

**Page 53: [80] Formatted   Wiggins, Elizabeth B. (LARC-E3)[UNIVERSITIES SPACE RESEARCH ASSOCIATION]                    2/3/21 1:00:00 PM**

Font color: Auto

**Page 53: [80] Formatted   Wiggins, Elizabeth B. (LARC-E3)[UNIVERSITIES SPACE RESEARCH ASSOCIATION]                    2/3/21 1:00:00 PM**

Font color: Auto

**Page 53: [81] Formatted   Wiggins, Elizabeth B. (LARC-E3)[UNIVERSITIES SPACE RESEARCH ASSOCIATION]                    2/3/21 1:00:00 PM**

Font color: Auto

**Page 53: [81] Formatted   Wiggins, Elizabeth B. (LARC-E3)[UNIVERSITIES SPACE RESEARCH ASSOCIATION]                    2/3/21 1:00:00 PM**

Font color: Auto

**Page 53: [82] Formatted   Wiggins, Elizabeth B. (LARC-E3)[UNIVERSITIES SPACE RESEARCH ASSOCIATION]                    2/3/21 1:00:00 PM**

Font color: Auto

**Page 53: [83] Formatted   Wiggins, Elizabeth B. (LARC-E3)[UNIVERSITIES SPACE RESEARCH ASSOCIATION]                    2/3/21 1:00:00 PM**

Position: Horizontal: Left, Relative to: Column, Vertical: In line, Relative to: Margin, Horizontal:  0", Wrap Around

**Page 53: [84] Formatted   Wiggins, Elizabeth B. (LARC-E3)[UNIVERSITIES SPACE RESEARCH ASSOCIATION]                    2/3/21 1:00:00 PM**

Font color: Black

**Page 53: [85] Formatted   Wiggins, Elizabeth B. (LARC-E3)[UNIVERSITIES SPACE RESEARCH ASSOCIATION]                    2/3/21 1:00:00 PM**

Font color: Auto

**Page 53: [85] Formatted   Wiggins, Elizabeth B. (LARC-E3)[UNIVERSITIES SPACE RESEARCH ASSOCIATION]                    2/3/21 1:00:00 PM**

Font color: Auto

**Page 53: [86] Formatted   Wiggins, Elizabeth B. (LARC-E3)[UNIVERSITIES SPACE RESEARCH ASSOCIATION]                          2/3/21 1:00:00 PM**

Font color: Auto

**Page 53: [86] Formatted   Wiggins, Elizabeth B. (LARC-E3)[UNIVERSITIES SPACE RESEARCH ASSOCIATION]                          2/3/21 1:00:00 PM**

Font color: Auto

**Page 53: [87] Formatted   Wiggins, Elizabeth B. (LARC-E3)[UNIVERSITIES SPACE RESEARCH ASSOCIATION]                          2/3/21 1:00:00 PM**

Font color: Auto

**Page 53: [87] Formatted   Wiggins, Elizabeth B. (LARC-E3)[UNIVERSITIES SPACE RESEARCH ASSOCIATION]                          2/3/21 1:00:00 PM**

Font color: Auto

**Page 53: [88] Formatted   Wiggins, Elizabeth B. (LARC-E3)[UNIVERSITIES SPACE RESEARCH ASSOCIATION]                          2/3/21 1:00:00 PM**

Font color: Auto

**Page 53: [89] Formatted   Wiggins, Elizabeth B. (LARC-E3)[UNIVERSITIES SPACE RESEARCH ASSOCIATION]                          2/3/21 1:00:00 PM**

Position: Horizontal: Left, Relative to: Column, Vertical: In line, Relative to: Margin, Horizontal:  0", Wrap Around

**Page 53: [90] Formatted   Wiggins, Elizabeth B. (LARC-E3)[UNIVERSITIES SPACE RESEARCH ASSOCIATION]                          2/3/21 1:00:00 PM**

Font color: Black

**Page 53: [91] Formatted   Wiggins, Elizabeth B. (LARC-E3)[UNIVERSITIES SPACE RESEARCH ASSOCIATION]                          2/3/21 1:00:00 PM**

Font color: Auto

**Page 53: [91] Formatted   Wiggins, Elizabeth B. (LARC-E3)[UNIVERSITIES SPACE RESEARCH ASSOCIATION]                          2/3/21 1:00:00 PM**

Font color: Auto

**Page 53: [92] Formatted   Wiggins, Elizabeth B. (LARC-E3)[UNIVERSITIES SPACE RESEARCH ASSOCIATION]                          2/3/21 1:00:00 PM**

Font color: Auto

**Page 53: [92] Formatted   Wiggins, Elizabeth B. (LARC-E3)[UNIVERSITIES SPACE RESEARCH ASSOCIATION]                          2/3/21 1:00:00 PM**

Font color: Auto

**Page 53: [93] Formatted   Wiggins, Elizabeth B. (LARC-E3)[UNIVERSITIES SPACE RESEARCH ASSOCIATION]                          2/3/21 1:00:00 PM**

Font color: Auto

**Page 53: [93] Formatted   Wiggins, Elizabeth B. (LARC-E3)[UNIVERSITIES SPACE RESEARCH ASSOCIATION]                          2/3/21 1:00:00 PM**

Font color: Auto

**Page 53: [94] Formatted   Wiggins, Elizabeth B. (LARC-E3)[UNIVERSITIES SPACE RESEARCH ASSOCIATION]                          2/3/21 1:00:00 PM**

Font color: Auto

**Page 53: [95] Formatted  Wiggins, Elizabeth B. (LARC-E3)[UNIVERSITIES SPACE RESEARCH ASSOCIATION]                    2/3/21 1:00:00 PM**

Position: Horizontal: Left, Relative to: Column, Vertical: In line, Relative to: Margin, Horizontal:  0", Wrap Around

**Page 53: [96] Formatted  Wiggins, Elizabeth B. (LARC-E3)[UNIVERSITIES SPACE RESEARCH ASSOCIATION]                    2/3/21 1:00:00 PM**

Font color: Black

**Page 53: [97] Formatted  Wiggins, Elizabeth B. (LARC-E3)[UNIVERSITIES SPACE RESEARCH ASSOCIATION]                    2/3/21 1:00:00 PM**

Font color: Auto

**Page 53: [97] Formatted  Wiggins, Elizabeth B. (LARC-E3)[UNIVERSITIES SPACE RESEARCH ASSOCIATION]                    2/3/21 1:00:00 PM**

Font color: Auto

**Page 53: [98] Formatted  Wiggins, Elizabeth B. (LARC-E3)[UNIVERSITIES SPACE RESEARCH ASSOCIATION]                    2/3/21 1:00:00 PM**

Font color: Auto

**Page 53: [98] Formatted  Wiggins, Elizabeth B. (LARC-E3)[UNIVERSITIES SPACE RESEARCH ASSOCIATION]                    2/3/21 1:00:00 PM**

Font color: Auto

**Page 53: [99] Formatted  Wiggins, Elizabeth B. (LARC-E3)[UNIVERSITIES SPACE RESEARCH ASSOCIATION]                    2/3/21 1:00:00 PM**

Font color: Auto

**Page 53: [99] Formatted  Wiggins, Elizabeth B. (LARC-E3)[UNIVERSITIES SPACE RESEARCH ASSOCIATION]                    2/3/21 1:00:00 PM**

Font color: Auto

**Page 53: [100] Formatted  Wiggins, Elizabeth B. (LARC-E3)[UNIVERSITIES SPACE RESEARCH ASSOCIATION]                    2/3/21 1:00:00 PM**

Font color: Auto

**Page 53: [101] Formatted  Wiggins, Elizabeth B. (LARC-E3)[UNIVERSITIES SPACE RESEARCH ASSOCIATION]                    2/3/21 1:00:00 PM**

Position: Horizontal: Left, Relative to: Column, Vertical: In line, Relative to: Margin, Horizontal:  0", Wrap Around

**Page 53: [102] Formatted  Wiggins, Elizabeth B. (LARC-E3)[UNIVERSITIES SPACE RESEARCH ASSOCIATION]                    2/3/21 1:00:00 PM**

Font color: Black

**Page 53: [103] Formatted  Wiggins, Elizabeth B. (LARC-E3)[UNIVERSITIES SPACE RESEARCH ASSOCIATION]                    2/3/21 1:00:00 PM**

Font color: Auto

**Page 53: [103] Formatted  Wiggins, Elizabeth B. (LARC-E3)[UNIVERSITIES SPACE RESEARCH ASSOCIATION]                    2/3/21 1:00:00 PM**

Font color: Auto

**Page 53: [104] Formatted  Wiggins, Elizabeth B. (LARC-E3)[UNIVERSITIES SPACE RESEARCH ASSOCIATION]                    2/3/21 1:00:00 PM**

Font color: Auto

**Page 53: [104] Formatted  Wiggins, Elizabeth B. (LARC-E3)[UNIVERSITIES SPACE RESEARCH ASSOCIATION]                        2/3/21 1:00:00 PM**

Font color: Auto

**Page 53: [105] Formatted  Wiggins, Elizabeth B. (LARC-E3)[UNIVERSITIES SPACE RESEARCH ASSOCIATION]                        2/3/21 1:00:00 PM**

Font color: Auto

**Page 53: [105] Formatted  Wiggins, Elizabeth B. (LARC-E3)[UNIVERSITIES SPACE RESEARCH ASSOCIATION]                        2/3/21 1:00:00 PM**

Font color: Auto

**Page 53: [106] Formatted  Wiggins, Elizabeth B. (LARC-E3)[UNIVERSITIES SPACE RESEARCH ASSOCIATION]                        2/3/21 1:00:00 PM**

Font color: Auto

**Page 53: [107] Formatted  Wiggins, Elizabeth B. (LARC-E3)[UNIVERSITIES SPACE RESEARCH ASSOCIATION]                        2/3/21 1:00:00 PM**

Position: Horizontal: Left, Relative to: Column, Vertical: In line, Relative to: Margin, Horizontal:  0", Wrap Around

**Page 53: [108] Formatted  Wiggins, Elizabeth B. (LARC-E3)[UNIVERSITIES SPACE RESEARCH ASSOCIATION]                        2/3/21 1:00:00 PM**

Font color: Black

**Page 53: [109] Formatted  Wiggins, Elizabeth B. (LARC-E3)[UNIVERSITIES SPACE RESEARCH ASSOCIATION]                        2/3/21 1:00:00 PM**

Font color: Auto

**Page 53: [109] Formatted  Wiggins, Elizabeth B. (LARC-E3)[UNIVERSITIES SPACE RESEARCH ASSOCIATION]                        2/3/21 1:00:00 PM**

Font color: Auto

**Page 53: [110] Formatted  Wiggins, Elizabeth B. (LARC-E3)[UNIVERSITIES SPACE RESEARCH ASSOCIATION]                        2/3/21 1:00:00 PM**

Font color: Auto

**Page 53: [110] Formatted  Wiggins, Elizabeth B. (LARC-E3)[UNIVERSITIES SPACE RESEARCH ASSOCIATION]                        2/3/21 1:00:00 PM**

Font color: Auto

**Page 53: [111] Formatted  Wiggins, Elizabeth B. (LARC-E3)[UNIVERSITIES SPACE RESEARCH ASSOCIATION]                        2/3/21 1:00:00 PM**

Font color: Auto

**Page 53: [111] Formatted  Wiggins, Elizabeth B. (LARC-E3)[UNIVERSITIES SPACE RESEARCH ASSOCIATION]                        2/3/21 1:00:00 PM**

Font color: Auto

**Page 53: [112] Formatted  Wiggins, Elizabeth B. (LARC-E3)[UNIVERSITIES SPACE RESEARCH ASSOCIATION]                        2/3/21 1:00:00 PM**

Font color: Auto

**Page 53: [113] Formatted  Wiggins, Elizabeth B. (LARC-E3)[UNIVERSITIES SPACE RESEARCH ASSOCIATION]                        2/3/21 1:00:00 PM**

Position: Horizontal: Left, Relative to: Column, Vertical: In line, Relative to: Margin, Horizontal:  0", Wrap Around

**Page 53: [114] Formatted  Wiggins, Elizabeth B. (LARC-E3)[UNIVERSITIES SPACE RESEARCH ASSOCIATION]                    2/3/21 1:00:00 PM**

Font color: Black

**Page 53: [115] Formatted  Wiggins, Elizabeth B. (LARC-E3)[UNIVERSITIES SPACE RESEARCH ASSOCIATION]                    2/3/21 1:00:00 PM**

Font color: Auto

**Page 53: [115] Formatted  Wiggins, Elizabeth B. (LARC-E3)[UNIVERSITIES SPACE RESEARCH ASSOCIATION]                    2/3/21 1:00:00 PM**

Font color: Auto

**Page 53: [116] Formatted  Wiggins, Elizabeth B. (LARC-E3)[UNIVERSITIES SPACE RESEARCH ASSOCIATION]                    2/3/21 1:00:00 PM**

Font color: Auto

**Page 53: [116] Formatted  Wiggins, Elizabeth B. (LARC-E3)[UNIVERSITIES SPACE RESEARCH ASSOCIATION]                    2/3/21 1:00:00 PM**

Font color: Auto

**Page 53: [117] Formatted  Wiggins, Elizabeth B. (LARC-E3)[UNIVERSITIES SPACE RESEARCH ASSOCIATION]                    2/3/21 1:00:00 PM**

Font color: Auto

**Page 53: [117] Formatted  Wiggins, Elizabeth B. (LARC-E3)[UNIVERSITIES SPACE RESEARCH ASSOCIATION]                    2/3/21 1:00:00 PM**

Font color: Auto

**Page 53: [118] Formatted  Wiggins, Elizabeth B. (LARC-E3)[UNIVERSITIES SPACE RESEARCH ASSOCIATION]                    2/3/21 1:00:00 PM**

Font color: Auto

**Page 53: [119] Formatted  Wiggins, Elizabeth B. (LARC-E3)[UNIVERSITIES SPACE RESEARCH ASSOCIATION]                    2/3/21 1:00:00 PM**

Position: Horizontal: Left, Relative to: Column, Vertical: In line, Relative to: Margin, Horizontal:  0", Wrap Around

**Page 53: [120] Formatted  Wiggins, Elizabeth B. (LARC-E3)[UNIVERSITIES SPACE RESEARCH ASSOCIATION]                    2/3/21 1:00:00 PM**

Font color: Black

**Page 53: [121] Formatted  Wiggins, Elizabeth B. (LARC-E3)[UNIVERSITIES SPACE RESEARCH ASSOCIATION]                    2/3/21 1:00:00 PM**

Font color: Auto

**Page 53: [122] Formatted  Wiggins, Elizabeth B. (LARC-E3)[UNIVERSITIES SPACE RESEARCH ASSOCIATION]                    2/3/21 1:00:00 PM**

Position: Horizontal: Left, Relative to: Column, Vertical: In line, Relative to: Margin, Horizontal:  0", Wrap Around

**Page 53: [123] Formatted  Wiggins, Elizabeth B. (LARC-E3)[UNIVERSITIES SPACE RESEARCH ASSOCIATION]                    2/3/21 1:00:00 PM**

Font color: Black

**Page 53: [124] Formatted  Wiggins, Elizabeth B. (LARC-E3)[UNIVERSITIES SPACE RESEARCH ASSOCIATION]                    2/3/21 1:00:00 PM**

Font color: Auto

**Page 53: [124] Formatted  Wiggins, Elizabeth B. (LARC–E3)[UNIVERSITIES SPACE RESEARCH ASSOCIATION]                2/3/21 1:00:00 PM**

Font color: Auto

**Page 53: [125] Formatted  Wiggins, Elizabeth B. (LARC–E3)[UNIVERSITIES SPACE RESEARCH ASSOCIATION]                2/3/21 1:00:00 PM**

Font color: Auto

**Page 53: [125] Formatted  Wiggins, Elizabeth B. (LARC–E3)[UNIVERSITIES SPACE RESEARCH ASSOCIATION]                2/3/21 1:00:00 PM**

Font color: Auto

**Page 53: [126] Formatted  Wiggins, Elizabeth B. (LARC–E3)[UNIVERSITIES SPACE RESEARCH ASSOCIATION]                2/3/21 1:00:00 PM**

Font color: Auto

**Page 53: [126] Formatted  Wiggins, Elizabeth B. (LARC–E3)[UNIVERSITIES SPACE RESEARCH ASSOCIATION]                2/3/21 1:00:00 PM**

Font color: Auto

**Page 53: [127] Formatted  Wiggins, Elizabeth B. (LARC–E3)[UNIVERSITIES SPACE RESEARCH ASSOCIATION]                2/3/21 1:00:00 PM**

Font color: Auto

**Page 53: [128] Formatted  Wiggins, Elizabeth B. (LARC–E3)[UNIVERSITIES SPACE RESEARCH ASSOCIATION]                2/3/21 1:00:00 PM**

Position: Horizontal: Left, Relative to: Column, Vertical: In line, Relative to: Margin, Horizontal:  0", Wrap Around

**Page 53: [129] Formatted  Wiggins, Elizabeth B. (LARC–E3)[UNIVERSITIES SPACE RESEARCH ASSOCIATION]                2/3/21 1:00:00 PM**

Font color: Black

**Page 53: [130] Formatted  Wiggins, Elizabeth B. (LARC–E3)[UNIVERSITIES SPACE RESEARCH ASSOCIATION]                2/3/21 1:00:00 PM**

Font color: Auto

**Page 53: [130] Formatted  Wiggins, Elizabeth B. (LARC–E3)[UNIVERSITIES SPACE RESEARCH ASSOCIATION]                2/3/21 1:00:00 PM**

Font color: Auto

**Page 53: [131] Formatted  Wiggins, Elizabeth B. (LARC–E3)[UNIVERSITIES SPACE RESEARCH ASSOCIATION]                2/3/21 1:00:00 PM**

Font color: Auto

**Page 53: [131] Formatted  Wiggins, Elizabeth B. (LARC–E3)[UNIVERSITIES SPACE RESEARCH ASSOCIATION]                2/3/21 1:00:00 PM**

Font color: Auto

**Page 53: [132] Formatted  Wiggins, Elizabeth B. (LARC–E3)[UNIVERSITIES SPACE RESEARCH ASSOCIATION]                2/3/21 1:00:00 PM**

Font color: Auto

**Page 53: [132] Formatted  Wiggins, Elizabeth B. (LARC–E3)[UNIVERSITIES SPACE RESEARCH ASSOCIATION]                2/3/21 1:00:00 PM**

Font color: Auto

**Page 53: [133] Formatted  Wiggins, Elizabeth B. (LARC-E3)[UNIVERSITIES SPACE RESEARCH ASSOCIATION]**                2/3/21 1:00:00 PM

Font color: Auto

**Page 53: [134] Formatted  Wiggins, Elizabeth B. (LARC-E3)[UNIVERSITIES SPACE RESEARCH ASSOCIATION]**                2/3/21 1:00:00 PM

Position: Horizontal: Left, Relative to: Column, Vertical: In line, Relative to: Margin, Horizontal:  0", Wrap Around

**Page 53: [135] Formatted  Wiggins, Elizabeth B. (LARC-E3)[UNIVERSITIES SPACE RESEARCH ASSOCIATION]**                2/3/21 1:00:00 PM

Font color: Black

**Page 53: [136] Formatted  Wiggins, Elizabeth B. (LARC-E3)[UNIVERSITIES SPACE RESEARCH ASSOCIATION]**                2/3/21 1:00:00 PM

Font color: Auto

**Page 53: [136] Formatted  Wiggins, Elizabeth B. (LARC-E3)[UNIVERSITIES SPACE RESEARCH ASSOCIATION]**                2/3/21 1:00:00 PM

Font color: Auto

**Page 53: [137] Formatted  Wiggins, Elizabeth B. (LARC-E3)[UNIVERSITIES SPACE RESEARCH ASSOCIATION]**                2/3/21 1:00:00 PM

Font color: Auto

**Page 53: [137] Formatted  Wiggins, Elizabeth B. (LARC-E3)[UNIVERSITIES SPACE RESEARCH ASSOCIATION]**                2/3/21 1:00:00 PM

Font color: Auto

**Page 53: [138] Formatted  Wiggins, Elizabeth B. (LARC-E3)[UNIVERSITIES SPACE RESEARCH ASSOCIATION]**                2/3/21 1:00:00 PM

Font color: Auto

**Page 53: [138] Formatted  Wiggins, Elizabeth B. (LARC-E3)[UNIVERSITIES SPACE RESEARCH ASSOCIATION]**                2/3/21 1:00:00 PM

Font color: Auto

**Page 53: [139] Formatted  Wiggins, Elizabeth B. (LARC-E3)[UNIVERSITIES SPACE RESEARCH ASSOCIATION]**                2/3/21 1:00:00 PM

Font color: Auto

**Page 53: [140] Formatted  Wiggins, Elizabeth B. (LARC-E3)[UNIVERSITIES SPACE RESEARCH ASSOCIATION]**                2/3/21 1:00:00 PM

Position: Horizontal: Left, Relative to: Column, Vertical: In line, Relative to: Margin, Horizontal:  0", Wrap Around

**Page 53: [141] Formatted  Wiggins, Elizabeth B. (LARC-E3)[UNIVERSITIES SPACE RESEARCH ASSOCIATION]**                2/3/21 1:00:00 PM

Font color: Black

**Page 53: [142] Formatted  Wiggins, Elizabeth B. (LARC-E3)[UNIVERSITIES SPACE RESEARCH ASSOCIATION]**                2/3/21 1:00:00 PM

Font color: Auto

**Page 53: [142] Formatted  Wiggins, Elizabeth B. (LARC-E3)[UNIVERSITIES SPACE RESEARCH ASSOCIATION]**                2/3/21 1:00:00 PM

Font color: Auto

**Page 53: [143] Formatted  Wiggins, Elizabeth B. (LARC-E3)[UNIVERSITIES SPACE RESEARCH ASSOCIATION]                    2/3/21 1:00:00 PM**

Font color: Auto

**Page 53: [143] Formatted  Wiggins, Elizabeth B. (LARC-E3)[UNIVERSITIES SPACE RESEARCH ASSOCIATION]                    2/3/21 1:00:00 PM**

Font color: Auto

**Page 53: [144] Formatted  Wiggins, Elizabeth B. (LARC-E3)[UNIVERSITIES SPACE RESEARCH ASSOCIATION]                    2/3/21 1:00:00 PM**

Font color: Auto

**Page 53: [144] Formatted  Wiggins, Elizabeth B. (LARC-E3)[UNIVERSITIES SPACE RESEARCH ASSOCIATION]                    2/3/21 1:00:00 PM**

Font color: Auto

**Page 53: [145] Formatted  Wiggins, Elizabeth B. (LARC-E3)[UNIVERSITIES SPACE RESEARCH ASSOCIATION]                    2/3/21 1:00:00 PM**

Font color: Auto

**Page 53: [146] Formatted  Wiggins, Elizabeth B. (LARC-E3)[UNIVERSITIES SPACE RESEARCH ASSOCIATION]                    2/3/21 1:00:00 PM**

Position: Horizontal: Left, Relative to: Column, Vertical: In line, Relative to: Margin, Horizontal:  0", Wrap Around

**Page 53: [147] Formatted  Wiggins, Elizabeth B. (LARC-E3)[UNIVERSITIES SPACE RESEARCH ASSOCIATION]                    2/3/21 1:00:00 PM**

Font color: Black

**Page 53: [148] Formatted  Wiggins, Elizabeth B. (LARC-E3)[UNIVERSITIES SPACE RESEARCH ASSOCIATION]                    2/3/21 1:00:00 PM**

Font color: Auto

**Page 53: [148] Formatted  Wiggins, Elizabeth B. (LARC-E3)[UNIVERSITIES SPACE RESEARCH ASSOCIATION]                    2/3/21 1:00:00 PM**

Font color: Auto

**Page 53: [149] Formatted  Wiggins, Elizabeth B. (LARC-E3)[UNIVERSITIES SPACE RESEARCH ASSOCIATION]                    2/3/21 1:00:00 PM**

Font color: Auto

**Page 53: [149] Formatted  Wiggins, Elizabeth B. (LARC-E3)[UNIVERSITIES SPACE RESEARCH ASSOCIATION]                    2/3/21 1:00:00 PM**

Font color: Auto

**Page 53: [150] Formatted  Wiggins, Elizabeth B. (LARC-E3)[UNIVERSITIES SPACE RESEARCH ASSOCIATION]                    2/3/21 1:00:00 PM**

Font color: Auto

**Page 53: [150] Formatted  Wiggins, Elizabeth B. (LARC-E3)[UNIVERSITIES SPACE RESEARCH ASSOCIATION]                    2/3/21 1:00:00 PM**

Font color: Auto

**Page 53: [151] Formatted  Wiggins, Elizabeth B. (LARC-E3)[UNIVERSITIES SPACE RESEARCH ASSOCIATION]                    2/3/21 1:00:00 PM**

Font color: Auto

**Page 53: [152] Formatted  Wiggins, Elizabeth B. (LARC-E3)[UNIVERSITIES SPACE RESEARCH ASSOCIATION]                  2/3/21 1:00:00 PM**

Position: Horizontal: Left, Relative to: Column, Vertical: In line, Relative to: Margin, Horizontal:  0", Wrap Around

**Page 53: [153] Formatted  Wiggins, Elizabeth B. (LARC-E3)[UNIVERSITIES SPACE RESEARCH ASSOCIATION]                  2/3/21 1:00:00 PM**

Font color: Black

**Page 53: [154] Formatted  Wiggins, Elizabeth B. (LARC-E3)[UNIVERSITIES SPACE RESEARCH ASSOCIATION]                  2/3/21 1:00:00 PM**

Position: Horizontal: Left, Relative to: Column, Vertical: In line, Relative to: Margin, Horizontal:  0", Wrap Around

**Page 53: [155] Formatted  Wiggins, Elizabeth B. (LARC-E3)[UNIVERSITIES SPACE RESEARCH ASSOCIATION]                  2/3/21 1:00:00 PM**

Font color: Auto

**Page 53: [156] Formatted  Wiggins, Elizabeth B. (LARC-E3)[UNIVERSITIES SPACE RESEARCH ASSOCIATION]                  2/3/21 1:00:00 PM**

Font: Not Bold, Font color: Black

**Page 53: [157] Formatted  Wiggins, Elizabeth B. (LARC-E3)[UNIVERSITIES SPACE RESEARCH ASSOCIATION]                  2/3/21 1:00:00 PM**

Font color: Auto

**Page 53: [158] Formatted  Wiggins, Elizabeth B. (LARC-E3)[UNIVERSITIES SPACE RESEARCH ASSOCIATION]                  2/3/21 1:00:00 PM**

Font color: Auto

**Page 53: [159] Formatted  Wiggins, Elizabeth B. (LARC-E3)[UNIVERSITIES SPACE RESEARCH ASSOCIATION]                  2/3/21 1:00:00 PM**

Font color: Black

**Page 53: [160] Formatted  Wiggins, Elizabeth B. (LARC-E3)[UNIVERSITIES SPACE RESEARCH ASSOCIATION]                  2/3/21 1:00:00 PM**

Font color: Auto

**Page 53: [161] Formatted  Wiggins, Elizabeth B. (LARC-E3)[UNIVERSITIES SPACE RESEARCH ASSOCIATION]                  2/3/21 1:00:00 PM**

Font color: Auto

**Page 53: [162] Formatted  Wiggins, Elizabeth B. (LARC-E3)[UNIVERSITIES SPACE RESEARCH ASSOCIATION]                  2/3/21 1:00:00 PM**

Font color: Black

**Page 53: [163] Formatted  Wiggins, Elizabeth B. (LARC-E3)[UNIVERSITIES SPACE RESEARCH ASSOCIATION]                  2/3/21 1:00:00 PM**

Font color: Auto

**Page 53: [163] Formatted  Wiggins, Elizabeth B. (LARC-E3)[UNIVERSITIES SPACE RESEARCH ASSOCIATION]                  2/3/21 1:00:00 PM**

Font color: Auto

**Page 53: [164] Formatted  Wiggins, Elizabeth B. (LARC-E3)[UNIVERSITIES SPACE RESEARCH ASSOCIATION]                  2/3/21 1:00:00 PM**

Position: Horizontal: Left, Relative to: Column, Vertical: In line, Relative to: Margin, Horizontal:  0", Wrap Around

**Page 53: [165] Formatted  Wiggins, Elizabeth B. (LARC-E3)[UNIVERSITIES SPACE RESEARCH ASSOCIATION]                    2/3/21 1:00:00 PM**

Font color: Auto

**Page 53: [166] Formatted  Wiggins, Elizabeth B. (LARC-E3)[UNIVERSITIES SPACE RESEARCH ASSOCIATION]                    2/3/21 1:00:00 PM**

Font: Not Bold, Font color: Black

**Page 53: [167] Formatted  Wiggins, Elizabeth B. (LARC-E3)[UNIVERSITIES SPACE RESEARCH ASSOCIATION]                    2/3/21 1:00:00 PM**

Font color: Auto

**Page 53: [168] Formatted  Wiggins, Elizabeth B. (LARC-E3)[UNIVERSITIES SPACE RESEARCH ASSOCIATION]                    2/3/21 1:00:00 PM**

Font color: Black

**Page 53: [169] Formatted  Wiggins, Elizabeth B. (LARC-E3)[UNIVERSITIES SPACE RESEARCH ASSOCIATION]                    2/3/21 1:00:00 PM**

Font color: Auto

**Page 53: [170] Formatted  Wiggins, Elizabeth B. (LARC-E3)[UNIVERSITIES SPACE RESEARCH ASSOCIATION]                    2/3/21 1:00:00 PM**

Font color: Black

**Page 53: [171] Formatted  Wiggins, Elizabeth B. (LARC-E3)[UNIVERSITIES SPACE RESEARCH ASSOCIATION]                    2/3/21 1:00:00 PM**

Font color: Auto

**Page 53: [171] Formatted  Wiggins, Elizabeth B. (LARC-E3)[UNIVERSITIES SPACE RESEARCH ASSOCIATION]                    2/3/21 1:00:00 PM**

Font color: Auto

**Page 53: [172] Formatted  Wiggins, Elizabeth B. (LARC-E3)[UNIVERSITIES SPACE RESEARCH ASSOCIATION]                    2/3/21 1:00:00 PM**

Position: Horizontal: Left, Relative to: Column, Vertical: In line, Relative to: Margin, Horizontal:  0", Wrap Around

**Page 53: [173] Formatted Table  Wiggins, Elizabeth B. (LARC-E3)[UNIVERSITIES SPACE RESEARCH ASSOCIATION]                    2/3/21 1:00:00 PM**

Formatted Table

**Page 53: [174] Formatted  Wiggins, Elizabeth B. (LARC-E3)[UNIVERSITIES SPACE RESEARCH ASSOCIATION]                    2/3/21 1:00:00 PM**

Font color: Auto

**Page 53: [175] Formatted  Wiggins, Elizabeth B. (LARC-E3)[UNIVERSITIES SPACE RESEARCH ASSOCIATION]                    2/3/21 1:00:00 PM**

Font: Not Bold, Font color: Black

**Page 53: [176] Formatted  Wiggins, Elizabeth B. (LARC-E3)[UNIVERSITIES SPACE RESEARCH ASSOCIATION]                    2/3/21 1:00:00 PM**

Font color: Auto

**Page 53: [177] Formatted  Wiggins, Elizabeth B. (LARC-E3)[UNIVERSITIES SPACE RESEARCH ASSOCIATION]                    2/3/21 1:00:00 PM**

Font color: Auto

**Page 53: [178] Formatted  Wiggins, Elizabeth B. (LARC-E3)[UNIVERSITIES SPACE RESEARCH ASSOCIATION]                              2/3/21 1:00:00 PM**

Font color: Black

**Page 53: [179] Formatted  Wiggins, Elizabeth B. (LARC-E3)[UNIVERSITIES SPACE RESEARCH ASSOCIATION]                              2/3/21 1:00:00 PM**

Font color: Auto

**Page 53: [180] Formatted  Wiggins, Elizabeth B. (LARC-E3)[UNIVERSITIES SPACE RESEARCH ASSOCIATION]                              2/3/21 1:00:00 PM**

Font color: Auto

**Page 53: [181] Formatted  Wiggins, Elizabeth B. (LARC-E3)[UNIVERSITIES SPACE RESEARCH ASSOCIATION]                              2/3/21 1:00:00 PM**

Font color: Black

**Page 53: [182] Formatted  Wiggins, Elizabeth B. (LARC-E3)[UNIVERSITIES SPACE RESEARCH ASSOCIATION]                              2/3/21 1:00:00 PM**

Font color: Auto

**Page 53: [182] Formatted  Wiggins, Elizabeth B. (LARC-E3)[UNIVERSITIES SPACE RESEARCH ASSOCIATION]                              2/3/21 1:00:00 PM**

Font color: Auto

**Page 53: [183] Formatted  Wiggins, Elizabeth B. (LARC-E3)[UNIVERSITIES SPACE RESEARCH ASSOCIATION]                              2/3/21 1:00:00 PM**

Position: Horizontal: Left, Relative to: Column, Vertical: In line, Relative to: Margin, Horizontal:  0", Wrap Around

**Page 53: [184] Formatted Table  Wiggins, Elizabeth B. (LARC-E3)[UNIVERSITIES SPACE RESEARCH ASSOCIATION]                              2/3/21 1:00:00 PM**

Formatted Table

**Page 53: [185] Formatted  Wiggins, Elizabeth B. (LARC-E3)[UNIVERSITIES SPACE RESEARCH ASSOCIATION]                              2/3/21 1:00:00 PM**

Font: Not Bold, Font color: Black

**Page 53: [186] Formatted  Wiggins, Elizabeth B. (LARC-E3)[UNIVERSITIES SPACE RESEARCH ASSOCIATION]                              2/3/21 1:00:00 PM**

Font: Not Bold, Font color: Auto

**Page 53: [187] Formatted  Wiggins, Elizabeth B. (LARC-E3)[UNIVERSITIES SPACE RESEARCH ASSOCIATION]                              2/3/21 1:00:00 PM**

Font: Not Bold, Font color: Auto

**Page 53: [188] Formatted  Wiggins, Elizabeth B. (LARC-E3)[UNIVERSITIES SPACE RESEARCH ASSOCIATION]                              2/3/21 1:00:00 PM**

Font: Not Bold, Font color: Black

**Page 53: [189] Formatted  Wiggins, Elizabeth B. (LARC-E3)[UNIVERSITIES SPACE RESEARCH ASSOCIATION]                              2/3/21 1:00:00 PM**

Font: Not Bold, Font color: Auto

**Page 53: [190] Formatted  Wiggins, Elizabeth B. (LARC-E3)[UNIVERSITIES SPACE RESEARCH ASSOCIATION]                              2/3/21 1:00:00 PM**

Font: Not Bold, Font color: Auto

**Page 53: [191] Formatted  Wiggins, Elizabeth B. (LARC-E3)[UNIVERSITIES SPACE RESEARCH ASSOCIATION]                    2/3/21 1:00:00 PM**

Font: Not Bold, Font color: Black

**Page 53: [192] Formatted  Wiggins, Elizabeth B. (LARC-E3)[UNIVERSITIES SPACE RESEARCH ASSOCIATION]                    2/3/21 1:00:00 PM**

Font: Not Bold, Font color: Black

**Page 53: [193] Formatted  Wiggins, Elizabeth B. (LARC-E3)[UNIVERSITIES SPACE RESEARCH ASSOCIATION]                    2/3/21 1:00:00 PM**

Font color: Auto

**Page 53: [194] Formatted  Wiggins, Elizabeth B. (LARC-E3)[UNIVERSITIES SPACE RESEARCH ASSOCIATION]                    2/3/21 1:00:00 PM**

Position: Horizontal: Left, Relative to: Column, Vertical: In line, Relative to: Margin, Horizontal:  0", Wrap Around

**Page 53: [195] Formatted Table  Wiggins, Elizabeth B. (LARC-E3)[UNIVERSITIES SPACE RESEARCH ASSOCIATION]                    2/3/21 1:00:00 PM**

Formatted Table

**Page 53: [196] Formatted  Wiggins, Elizabeth B. (LARC-E3)[UNIVERSITIES SPACE RESEARCH ASSOCIATION]                    2/3/21 1:00:00 PM**

Font color: Black

**Page 53: [197] Formatted  Wiggins, Elizabeth B. (LARC-E3)[UNIVERSITIES SPACE RESEARCH ASSOCIATION]                    2/3/21 1:00:00 PM**

Font color: Auto

**Page 53: [198] Formatted  Wiggins, Elizabeth B. (LARC-E3)[UNIVERSITIES SPACE RESEARCH ASSOCIATION]                    2/3/21 1:00:00 PM**

Position: Horizontal: Left, Relative to: Column, Vertical: In line, Relative to: Margin, Horizontal:  0", Wrap Around

**Page 53: [199] Formatted  Wiggins, Elizabeth B. (LARC-E3)[UNIVERSITIES SPACE RESEARCH ASSOCIATION]                    2/3/21 1:00:00 PM**

Font color: Black

**Page 53: [200] Formatted  Wiggins, Elizabeth B. (LARC-E3)[UNIVERSITIES SPACE RESEARCH ASSOCIATION]                    2/3/21 1:00:00 PM**

Font color: Auto

**Page 53: [200] Formatted  Wiggins, Elizabeth B. (LARC-E3)[UNIVERSITIES SPACE RESEARCH ASSOCIATION]                    2/3/21 1:00:00 PM**

Font color: Auto

**Page 53: [201] Formatted  Wiggins, Elizabeth B. (LARC-E3)[UNIVERSITIES SPACE RESEARCH ASSOCIATION]                    2/3/21 1:00:00 PM**

Font color: Auto

**Page 53: [201] Formatted  Wiggins, Elizabeth B. (LARC-E3)[UNIVERSITIES SPACE RESEARCH ASSOCIATION]                    2/3/21 1:00:00 PM**

Font color: Auto

**Page 53: [202] Formatted  Wiggins, Elizabeth B. (LARC-E3)[UNIVERSITIES SPACE RESEARCH ASSOCIATION]                    2/3/21 1:00:00 PM**

Font color: Auto

**Page 53: [202] Formatted   Wiggins, Elizabeth B. (LARC‑E3)[UNIVERSITIES SPACE RESEARCH ASSOCIATION]                    2/3/21 1:00:00 PM**

Font color: Auto

**Page 53: [203] Formatted   Wiggins, Elizabeth B. (LARC‑E3)[UNIVERSITIES SPACE RESEARCH ASSOCIATION]                    2/3/21 1:00:00 PM**

Position: Horizontal: Left, Relative to: Column, Vertical: In line, Relative to: Margin, Horizontal:  0", Wrap Around

**Page 53: [204] Formatted   Wiggins, Elizabeth B. (LARC‑E3)[UNIVERSITIES SPACE RESEARCH ASSOCIATION]                    2/3/21 1:00:00 PM**

Font color: Auto

**Page 53: [204] Formatted   Wiggins, Elizabeth B. (LARC‑E3)[UNIVERSITIES SPACE RESEARCH ASSOCIATION]                    2/3/21 1:00:00 PM**

Font color: Auto

**Page 53: [205] Formatted   Wiggins, Elizabeth B. (LARC‑E3)[UNIVERSITIES SPACE RESEARCH ASSOCIATION]                    2/3/21 1:00:00 PM**

Font color: Auto

**Page 53: [205] Formatted   Wiggins, Elizabeth B. (LARC‑E3)[UNIVERSITIES SPACE RESEARCH ASSOCIATION]                    2/3/21 1:00:00 PM**

Font color: Auto

**Page 53: [206] Formatted   Wiggins, Elizabeth B. (LARC‑E3)[UNIVERSITIES SPACE RESEARCH ASSOCIATION]                    2/3/21 1:00:00 PM**

Font color: Auto

**Page 53: [206] Formatted   Wiggins, Elizabeth B. (LARC‑E3)[UNIVERSITIES SPACE RESEARCH ASSOCIATION]                    2/3/21 1:00:00 PM**

Font color: Auto

**Page 53: [207] Formatted   Wiggins, Elizabeth B. (LARC‑E3)[UNIVERSITIES SPACE RESEARCH ASSOCIATION]                    2/3/21 1:00:00 PM**

Font color: Auto

**Page 53: [207] Formatted   Wiggins, Elizabeth B. (LARC‑E3)[UNIVERSITIES SPACE RESEARCH ASSOCIATION]                    2/3/21 1:00:00 PM**

Font color: Auto

**Page 53: [208] Formatted   Wiggins, Elizabeth B. (LARC‑E3)[UNIVERSITIES SPACE RESEARCH ASSOCIATION]                    2/3/21 1:00:00 PM**

Font color: Auto

**Page 53: [209] Formatted   Wiggins, Elizabeth B. (LARC‑E3)[UNIVERSITIES SPACE RESEARCH ASSOCIATION]                    2/3/21 1:00:00 PM**

Position: Horizontal: Left, Relative to: Column, Vertical: In line, Relative to: Margin, Horizontal:  0", Wrap Around

**Page 53: [210] Formatted   Wiggins, Elizabeth B. (LARC‑E3)[UNIVERSITIES SPACE RESEARCH ASSOCIATION]                    2/3/21 1:00:00 PM**

Font color: Black

**Page 53: [211] Formatted   Wiggins, Elizabeth B. (LARC‑E3)[UNIVERSITIES SPACE RESEARCH ASSOCIATION]                    2/3/21 1:00:00 PM**

Font color: Auto

**Page 53: [211] Formatted   Wiggins, Elizabeth B. (LARC‑E3)[UNIVERSITIES SPACE RESEARCH ASSOCIATION]                2/3/21 1:00:00 PM**

Font color: Auto

**Page 53: [212] Formatted   Wiggins, Elizabeth B. (LARC‑E3)[UNIVERSITIES SPACE RESEARCH ASSOCIATION]                2/3/21 1:00:00 PM**

Font color: Auto

**Page 53: [212] Formatted   Wiggins, Elizabeth B. (LARC‑E3)[UNIVERSITIES SPACE RESEARCH ASSOCIATION]                2/3/21 1:00:00 PM**

Font color: Auto

**Page 53: [213] Formatted   Wiggins, Elizabeth B. (LARC‑E3)[UNIVERSITIES SPACE RESEARCH ASSOCIATION]                2/3/21 1:00:00 PM**

Font color: Auto

**Page 53: [213] Formatted   Wiggins, Elizabeth B. (LARC‑E3)[UNIVERSITIES SPACE RESEARCH ASSOCIATION]                2/3/21 1:00:00 PM**

Font color: Auto

**Page 53: [214] Formatted   Wiggins, Elizabeth B. (LARC‑E3)[UNIVERSITIES SPACE RESEARCH ASSOCIATION]                2/3/21 1:00:00 PM**

Font color: Auto

**Page 53: [215] Formatted   Wiggins, Elizabeth B. (LARC‑E3)[UNIVERSITIES SPACE RESEARCH ASSOCIATION]                2/3/21 1:00:00 PM**

Position: Horizontal: Left, Relative to: Column, Vertical: In line, Relative to: Margin, Horizontal:  0", Wrap Around

**Page 53: [216] Formatted   Wiggins, Elizabeth B. (LARC‑E3)[UNIVERSITIES SPACE RESEARCH ASSOCIATION]                2/3/21 1:00:00 PM**

Font color: Black

**Page 53: [217] Formatted   Wiggins, Elizabeth B. (LARC‑E3)[UNIVERSITIES SPACE RESEARCH ASSOCIATION]                2/3/21 1:00:00 PM**

Font color: Auto

**Page 53: [217] Formatted   Wiggins, Elizabeth B. (LARC‑E3)[UNIVERSITIES SPACE RESEARCH ASSOCIATION]                2/3/21 1:00:00 PM**

Font color: Auto

**Page 53: [218] Formatted   Wiggins, Elizabeth B. (LARC‑E3)[UNIVERSITIES SPACE RESEARCH ASSOCIATION]                2/3/21 1:00:00 PM**

Font color: Auto

**Page 53: [218] Formatted   Wiggins, Elizabeth B. (LARC‑E3)[UNIVERSITIES SPACE RESEARCH ASSOCIATION]                2/3/21 1:00:00 PM**

Font color: Auto

**Page 53: [219] Formatted   Wiggins, Elizabeth B. (LARC‑E3)[UNIVERSITIES SPACE RESEARCH ASSOCIATION]                2/3/21 1:00:00 PM**

Font color: Auto

**Page 53: [219] Formatted   Wiggins, Elizabeth B. (LARC‑E3)[UNIVERSITIES SPACE RESEARCH ASSOCIATION]                2/3/21 1:00:00 PM**

Font color: Auto

**Page 53: [220] Formatted  Wiggins, Elizabeth B. (LARC–E3)[UNIVERSITIES SPACE RESEARCH ASSOCIATION]                        2/3/21 1:00:00 PM**

Font color: Auto

**Page 53: [221] Formatted  Wiggins, Elizabeth B. (LARC–E3)[UNIVERSITIES SPACE RESEARCH ASSOCIATION]                        2/3/21 1:00:00 PM**

Position: Horizontal: Left, Relative to: Column, Vertical: In line, Relative to: Margin, Horizontal:  0", Wrap Around

**Page 53: [222] Formatted  Wiggins, Elizabeth B. (LARC–E3)[UNIVERSITIES SPACE RESEARCH ASSOCIATION]                        2/3/21 1:00:00 PM**

Font color: Black

**Page 53: [223] Formatted  Wiggins, Elizabeth B. (LARC–E3)[UNIVERSITIES SPACE RESEARCH ASSOCIATION]                        2/3/21 1:00:00 PM**

Font color: Auto

**Page 53: [224] Formatted  Wiggins, Elizabeth B. (LARC–E3)[UNIVERSITIES SPACE RESEARCH ASSOCIATION]                        2/3/21 1:00:00 PM**

Position: Horizontal: Left, Relative to: Column, Vertical: In line, Relative to: Margin, Horizontal:  0", Wrap Around

**Page 53: [225] Formatted  Wiggins, Elizabeth B. (LARC–E3)[UNIVERSITIES SPACE RESEARCH ASSOCIATION]                        2/3/21 1:00:00 PM**

Font color: Black

**Page 53: [226] Formatted  Wiggins, Elizabeth B. (LARC–E3)[UNIVERSITIES SPACE RESEARCH ASSOCIATION]                        2/3/21 1:00:00 PM**

Font color: Auto

**Page 53: [226] Formatted  Wiggins, Elizabeth B. (LARC–E3)[UNIVERSITIES SPACE RESEARCH ASSOCIATION]                        2/3/21 1:00:00 PM**

Font color: Auto

**Page 53: [227] Formatted  Wiggins, Elizabeth B. (LARC–E3)[UNIVERSITIES SPACE RESEARCH ASSOCIATION]                        2/3/21 1:00:00 PM**

Font color: Auto

**Page 53: [227] Formatted  Wiggins, Elizabeth B. (LARC–E3)[UNIVERSITIES SPACE RESEARCH ASSOCIATION]                        2/3/21 1:00:00 PM**

Font color: Auto

**Page 53: [228] Formatted  Wiggins, Elizabeth B. (LARC–E3)[UNIVERSITIES SPACE RESEARCH ASSOCIATION]                        2/3/21 1:00:00 PM**

Font color: Auto

**Page 53: [228] Formatted  Wiggins, Elizabeth B. (LARC–E3)[UNIVERSITIES SPACE RESEARCH ASSOCIATION]                        2/3/21 1:00:00 PM**

Font color: Auto

**Page 53: [229] Formatted  Wiggins, Elizabeth B. (LARC–E3)[UNIVERSITIES SPACE RESEARCH ASSOCIATION]                        2/3/21 1:00:00 PM**

Font color: Auto

**Page 53: [230] Formatted  Wiggins, Elizabeth B. (LARC–E3)[UNIVERSITIES SPACE RESEARCH ASSOCIATION]                        2/3/21 1:00:00 PM**

Position: Horizontal: Left, Relative to: Column, Vertical: In line, Relative to: Margin, Horizontal:  0", Wrap Around

**Page 53: [231] Formatted  Wiggins, Elizabeth B. (LARC-E3)[UNIVERSITIES SPACE RESEARCH ASSOCIATION]                    2/3/21 1:00:00 PM**

Font color: Black

**Page 53: [232] Formatted  Wiggins, Elizabeth B. (LARC-E3)[UNIVERSITIES SPACE RESEARCH ASSOCIATION]                    2/3/21 1:00:00 PM**

Font color: Auto

**Page 53: [232] Formatted  Wiggins, Elizabeth B. (LARC-E3)[UNIVERSITIES SPACE RESEARCH ASSOCIATION]                    2/3/21 1:00:00 PM**

Font color: Auto

**Page 53: [233] Formatted  Wiggins, Elizabeth B. (LARC-E3)[UNIVERSITIES SPACE RESEARCH ASSOCIATION]                    2/3/21 1:00:00 PM**

Font color: Auto

**Page 53: [233] Formatted  Wiggins, Elizabeth B. (LARC-E3)[UNIVERSITIES SPACE RESEARCH ASSOCIATION]                    2/3/21 1:00:00 PM**

Font color: Auto

**Page 53: [234] Formatted  Wiggins, Elizabeth B. (LARC-E3)[UNIVERSITIES SPACE RESEARCH ASSOCIATION]                    2/3/21 1:00:00 PM**

Font color: Black

**Page 55: [235] Deleted  Wiggins, Elizabeth B. (LARC-E3)[UNIVERSITIES SPACE RESEARCH ASSOCIATION]                    2/3/21 1:00:00 PM**

**Page 56: [236] Formatted  Wiggins, Elizabeth B. (LARC-E3)[UNIVERSITIES SPACE RESEARCH ASSOCIATION]                    2/3/21 1:00:00 PM**

Position: Horizontal: Left, Relative to: Column, Vertical: In line, Relative to: Margin, Horizontal:  0", Wrap Around

**Page 56: [237] Formatted  Wiggins, Elizabeth B. (LARC-E3)[UNIVERSITIES SPACE RESEARCH ASSOCIATION]                    2/3/21 1:00:00 PM**

Position: Horizontal: Left, Relative to: Column, Vertical: In line, Relative to: Margin, Horizontal:  0", Wrap Around

**Page 56: [238] Formatted  Wiggins, Elizabeth B. (LARC-E3)[UNIVERSITIES SPACE RESEARCH ASSOCIATION]                    2/3/21 1:00:00 PM**

Position: Horizontal: Left, Relative to: Column, Vertical: In line, Relative to: Margin, Horizontal:  0", Wrap Around

**Page 56: [239] Formatted  Wiggins, Elizabeth B. (LARC-E3)[UNIVERSITIES SPACE RESEARCH ASSOCIATION]                    2/3/21 1:00:00 PM**

Position: Horizontal: Left, Relative to: Column, Vertical: In line, Relative to: Margin, Horizontal:  0", Wrap Around

**Page 56: [240] Formatted  Wiggins, Elizabeth B. (LARC-E3)[UNIVERSITIES SPACE RESEARCH ASSOCIATION]                    2/3/21 1:00:00 PM**

Position: Horizontal: Left, Relative to: Column, Vertical: In line, Relative to: Margin, Horizontal:  0", Wrap Around

**Page 56: [241] Formatted  Wiggins, Elizabeth B. (LARC-E3)[UNIVERSITIES SPACE RESEARCH ASSOCIATION]                    2/3/21 1:00:00 PM**

Position: Horizontal: Left, Relative to: Column, Vertical: In line, Relative to: Margin, Horizontal: 0", Wrap Around

**Page 56: [242] Formatted  Wiggins, Elizabeth B. (LARC-E3)[UNIVERSITIES SPACE RESEARCH ASSOCIATION]                    2/3/21 1:00:00 PM**

Position: Horizontal: Left, Relative to: Column, Vertical: In line, Relative to: Margin, Horizontal: 0", Wrap Around

**Page 56: [243] Formatted  Wiggins, Elizabeth B. (LARC-E3)[UNIVERSITIES SPACE RESEARCH ASSOCIATION]                    2/3/21 1:00:00 PM**

Position: Horizontal: Left, Relative to: Column, Vertical: In line, Relative to: Margin, Horizontal: 0", Wrap Around

**Page 56: [244] Formatted  Wiggins, Elizabeth B. (LARC-E3)[UNIVERSITIES SPACE RESEARCH ASSOCIATION]                    2/3/21 1:00:00 PM**

Position: Horizontal: Left, Relative to: Column, Vertical: In line, Relative to: Margin, Horizontal: 0", Wrap Around

**Page 56: [245] Formatted  Wiggins, Elizabeth B. (LARC-E3)[UNIVERSITIES SPACE RESEARCH ASSOCIATION]                    2/3/21 1:00:00 PM**

Position: Horizontal: Left, Relative to: Column, Vertical: In line, Relative to: Margin, Horizontal: 0", Wrap Around

**Page 56: [246] Formatted  Wiggins, Elizabeth B. (LARC-E3)[UNIVERSITIES SPACE RESEARCH ASSOCIATION]                    2/3/21 1:00:00 PM**

Position: Horizontal: Left, Relative to: Column, Vertical: In line, Relative to: Margin, Horizontal: 0", Wrap Around

**Page 56: [247] Formatted  Wiggins, Elizabeth B. (LARC-E3)[UNIVERSITIES SPACE RESEARCH ASSOCIATION]                    2/3/21 1:00:00 PM**

Position: Horizontal: Left, Relative to: Column, Vertical: In line, Relative to: Margin, Horizontal: 0", Wrap Around

**Page 56: [248] Formatted  Wiggins, Elizabeth B. (LARC-E3)[UNIVERSITIES SPACE RESEARCH ASSOCIATION]                    2/3/21 1:00:00 PM**

Position: Horizontal: Left, Relative to: Column, Vertical: In line, Relative to: Margin, Horizontal: 0", Wrap Around

**Page 56: [249] Formatted  Wiggins, Elizabeth B. (LARC-E3)[UNIVERSITIES SPACE RESEARCH ASSOCIATION]                    2/3/21 1:00:00 PM**

Position: Horizontal: Left, Relative to: Column, Vertical: In line, Relative to: Margin, Horizontal: 0", Wrap Around

**Page 56: [250] Formatted  Wiggins, Elizabeth B. (LARC-E3)[UNIVERSITIES SPACE RESEARCH ASSOCIATION]                    2/3/21 1:00:00 PM**

Position: Horizontal: Left, Relative to: Column, Vertical: In line, Relative to: Margin, Horizontal: 0", Wrap Around

**Page 56: [251] Formatted  Wiggins, Elizabeth B. (LARC-E3)[UNIVERSITIES SPACE RESEARCH ASSOCIATION]                    2/3/21 1:00:00 PM**

Position: Horizontal: Left, Relative to: Column, Vertical: In line, Relative to: Margin, Horizontal: 0", Wrap Around

**Page 56: [252] Formatted  Wiggins, Elizabeth B. (LARC-E3)[UNIVERSITIES SPACE RESEARCH ASSOCIATION]                    2/3/21 1:00:00 PM**

Position: Horizontal: Left, Relative to: Column, Vertical: In line, Relative to: Margin, Horizontal:  0", Wrap Around

**Page 56: [253] Formatted  Wiggins, Elizabeth B. (LARC-E3)[UNIVERSITIES SPACE RESEARCH ASSOCIATION]                    2/3/21 1:00:00 PM**

Position: Horizontal: Left, Relative to: Column, Vertical: In line, Relative to: Margin, Horizontal:  0", Wrap Around

**Page 56: [254] Formatted  Wiggins, Elizabeth B. (LARC-E3)[UNIVERSITIES SPACE RESEARCH ASSOCIATION]                    2/3/21 1:00:00 PM**

Position: Horizontal: Left, Relative to: Column, Vertical: In line, Relative to: Margin, Horizontal:  0", Wrap Around

**Page 56: [255] Formatted  Wiggins, Elizabeth B. (LARC-E3)[UNIVERSITIES SPACE RESEARCH ASSOCIATION]                    2/3/21 1:00:00 PM**

Position: Horizontal: Left, Relative to: Column, Vertical: In line, Relative to: Margin, Horizontal:  0", Wrap Around

**Page 56: [256] Formatted  Wiggins, Elizabeth B. (LARC-E3)[UNIVERSITIES SPACE RESEARCH ASSOCIATION]                    2/3/21 1:00:00 PM**

Position: Horizontal: Left, Relative to: Column, Vertical: In line, Relative to: Margin, Horizontal:  0", Wrap Around

**Page 56: [257] Formatted  Wiggins, Elizabeth B. (LARC-E3)[UNIVERSITIES SPACE RESEARCH ASSOCIATION]                    2/3/21 1:00:00 PM**

Position: Horizontal: Left, Relative to: Column, Vertical: In line, Relative to: Margin, Horizontal:  0", Wrap Around

**Page 56: [258] Formatted  Wiggins, Elizabeth B. (LARC-E3)[UNIVERSITIES SPACE RESEARCH ASSOCIATION]                    2/3/21 1:00:00 PM**

Position: Horizontal: Left, Relative to: Column, Vertical: In line, Relative to: Margin, Horizontal:  0", Wrap Around

**Page 56: [259] Formatted  Wiggins, Elizabeth B. (LARC-E3)[UNIVERSITIES SPACE RESEARCH ASSOCIATION]                    2/3/21 1:00:00 PM**

Position: Horizontal: Left, Relative to: Column, Vertical: In line, Relative to: Margin, Horizontal:  0", Wrap Around

**Page 56: [260] Formatted  Wiggins, Elizabeth B. (LARC-E3)[UNIVERSITIES SPACE RESEARCH ASSOCIATION]                    2/3/21 1:00:00 PM**

Position: Horizontal: Left, Relative to: Column, Vertical: In line, Relative to: Margin, Horizontal:  0", Wrap Around

**Page 56: [261] Formatted  Wiggins, Elizabeth B. (LARC-E3)[UNIVERSITIES SPACE RESEARCH ASSOCIATION]                    2/3/21 1:00:00 PM**

Position: Horizontal: Left, Relative to: Column, Vertical: In line, Relative to: Margin, Horizontal:  0", Wrap Around

**Page 56: [262] Formatted  Wiggins, Elizabeth B. (LARC-E3)[UNIVERSITIES SPACE RESEARCH ASSOCIATION]                    2/3/21 1:00:00 PM**

Position: Horizontal: Left, Relative to: Column, Vertical: In line, Relative to: Margin, Horizontal:  0", Wrap Around

**Page 56: [263] Formatted  Wiggins, Elizabeth B. (LARC-E3)[UNIVERSITIES SPACE RESEARCH ASSOCIATION]                    2/3/21 1:00:00 PM**

Position: Horizontal: Left, Relative to: Column, Vertical: In line, Relative to: Margin, Horizontal:  0", Wrap Around

**Page 56: [264] Formatted  Wiggins, Elizabeth B. (LARC–E3)[UNIVERSITIES SPACE RESEARCH ASSOCIATION]                           2/3/21 1:00:00 PM**

Position: Horizontal: Left, Relative to: Column, Vertical: In line, Relative to: Margin, Horizontal:  0", Wrap Around

**Page 56: [265] Formatted  Wiggins, Elizabeth B. (LARC–E3)[UNIVERSITIES SPACE RESEARCH ASSOCIATION]                           2/3/21 1:00:00 PM**

Position: Horizontal: Left, Relative to: Column, Vertical: In line, Relative to: Margin, Horizontal:  0", Wrap Around

**Page 56: [266] Formatted  Wiggins, Elizabeth B. (LARC–E3)[UNIVERSITIES SPACE RESEARCH ASSOCIATION]                           2/3/21 1:00:00 PM**

Position: Horizontal: Left, Relative to: Column, Vertical: In line, Relative to: Margin, Horizontal:  0", Wrap Around

**Page 56: [267] Formatted  Wiggins, Elizabeth B. (LARC–E3)[UNIVERSITIES SPACE RESEARCH ASSOCIATION]                           2/3/21 1:00:00 PM**

Position: Horizontal: Left, Relative to: Column, Vertical: In line, Relative to: Margin, Horizontal:  0", Wrap Around

**Page 56: [268] Formatted  Wiggins, Elizabeth B. (LARC–E3)[UNIVERSITIES SPACE RESEARCH ASSOCIATION]                           2/3/21 1:00:00 PM**

Position: Horizontal: Left, Relative to: Column, Vertical: In line, Relative to: Margin, Horizontal:  0", Wrap Around

**Page 56: [269] Formatted  Wiggins, Elizabeth B. (LARC–E3)[UNIVERSITIES SPACE RESEARCH ASSOCIATION]                           2/3/21 1:00:00 PM**

Position: Horizontal: Left, Relative to: Column, Vertical: In line, Relative to: Margin, Horizontal:  0", Wrap Around

**Page 56: [270] Formatted  Wiggins, Elizabeth B. (LARC–E3)[UNIVERSITIES SPACE RESEARCH ASSOCIATION]                           2/3/21 1:00:00 PM**

Position: Horizontal: Left, Relative to: Column, Vertical: In line, Relative to: Margin, Horizontal:  0", Wrap Around

**Page 56: [271] Deleted  Wiggins, Elizabeth B. (LARC–E3)[UNIVERSITIES SPACE RESEARCH ASSOCIATION]                           2/3/21 1:00:00 PM**

**Page 56: [272] Deleted  Wiggins, Elizabeth B. (LARC–E3)[UNIVERSITIES SPACE RESEARCH ASSOCIATION]                           2/3/21 1:00:00 PM**